# Long-range GABAergic projections contribute to cortical feedback control of sensory processing

Camille Mazo [1,2] ✉, Antoine Nissant [1], Soham Saha[1], Enzo Peroni [1], Pierre-Marie Lledo [1,3] ✉ & Gabriel Lepousez [1,3] ✉

In the olfactory system, the olfactory cortex sends glutamatergic projections back to the first stage of olfactory processing, the olfactory bulb (OB). Such corticofugal excitatory circuits − a canonical circuit motif described in all sensory systems− dynamically adjust early sensory processing. Here, we uncover a corticofugal inhibitory feedback to OB, originating from a sub-population of GABAergic neurons in the anterior olfactory cortex and inner-vating both local and output OB neurons. In vivo imaging and network modeling showed that optogenetic activation of cortical GABAergic projec-tions drives a net subtractive inhibition of both spontaneous and odor-evoked activity in local as well as output neurons. In output neurons, stimulation of cortical GABAergic feedback enhances separation of population odor responses in tufted cells, but not mitral cells. Targeted pharmacogenetic silencing of cortical GABAergic axon terminals impaired discrimination of similar odor mixtures. Thus, corticofugal GABAergic projections represent an additional circuit motif in cortical feedback control of sensory processing.

Recent advances in genetic tools applied to cell- and circuit-tracing has allowed for the discovery of an increasing number of long-range GABAergic projection neurons in the cortex – where they may con-stitute 1–10% of the total GABAergic neurons in mice, rats, cats and monkeys[1–5]. Long- range projecting GABAergic neurons express a variety of classical markers for interneurons[6–8], sometimes forming intermingled populations within a single structure, where they exhibit distinct connectivity and exert various functions. For instance, bidir-ectional GABAergic projections between the hippocampus and entorhinal cortex synchronize the rhythmic network activity and gate spike-timing plasticity[6,7], cortico-striatal and cortico-amygdala GABAergic projections regulate spike generation and excitability of their postsynaptic target and influence locomotion as well as reward coding[9–11].

In mammalian sensory systems, external stimuli trigger a feed-forward flow of information from the sensory organ to the primary and higher-order sensory cortices via a set of subcortical structures, thereby defining a hierarchy between sensory brain regions. In parallel, higher-order cortical sensory areas send top-down information to lower-order areas, constantly shaping early information processing. Such feedback is thought to convey contextual information and pre-dictions to lower areas, not only playing a decisive role in selective attention and object expectation, but also in the encoding and recall of learned information[12–14]. Top-down cortical feedback projections are thought to be exclusively mediated by glutamatergic neurons, while GABAergic neurons are in turn frequently referred to as exclusively mediating local information processing[15]. In the present study, we challenge that view by investigating whether sensory cortical circuits can also parallelize excitatory and inhibitory top-down projections.

In the olfactory system, olfactory sensory neurons project to the olfactory bulb (OB), in the glomerular layer (GL) where they form synapses with apical dendrites of mitral and tufted cells (MCs and TCs,

[1]Institut Pasteur, Université Paris Cité, Centre National de la Recherche Scientifique, Unité Mixte de Recherche 3571, Perception and Memory Unit, F-75015 Paris, France. [2]Present address: Champalimaud Foundation, Lisbon, Portugal. [3]These authors contributed equally: Pierre-Marie Lledo, Gabriel Lepousez. ✉e-mail: camille.mazo@gmail.com; pmlledo@pasteur.fr; gabriel.lepousez@pasteur.fr

respectively), the output projection neurons of the OB. MC and TC activity is shaped by a large population of local GABAergic interneurons which synapse onto their apical or lateral dendrites. The anterior olfactory nucleus (AON) and the anterior piriform cortex (APC) − forming the anterior olfactory cortex (AOC) − is the primary recipient of OB outputs. Like the cortico-thalamic feedback pathway, the AOC send extensive glutamatergic projections back to the OB[16–26]. Glutamatergic feedback projections from the AOC target virtually all types of neurons in the OB and induce robust disynaptic inhibition onto MCs and TCs[16,23,27–29]. These reciprocal connections between the OB network and the AOC are important for proper oscillations in the OB[30,31], decorrelation of OB output activity[32], inter-hemispheric coordination[33] and modulate odor perception threshold[34] and odor-association learning[35] in a context-dependent manner[36]. The OB additionally receives external inputs from neuromodulatory systems. Specifically, the basal forebrain sends GABAergic axons to the OB[37,38] where they form synapses exclusively onto inhibitory neurons[39–41]. Optogenetic stimulation of basal forebrain GABAergic axons in the OB results in a bidirectional modulation of MCs, switching from an inhibitory to disinhibitory net effect in the presence of odor input[40,42]. Basal forebrain GABAergic projections have also been shown to influence OB oscillations, MC spike synchronization[40] and olfactory discrimination[43].

Here we reveal that in addition to the cortical glutamatergic feedback, the AOC sends GABAergic projections back to the OB. Specifically, the AON *pars posterioralis* (AONp) form a particularly dense cluster of OB-projecting GABAergic neurons. Similar to their glutamatergic counterpart[16,23,27] and in contrast to basal forebrain

GABAergic inputs, we demonstrate that cortical GABAergic feedback forms synapses with MCs and TCs as well as deep-layer GABAergic interneurons, but spares GL GABAergic neurons. In awake mice, long-range GABAergic projection stimulation entrained beta oscillations in the OB. Cortical GABAergic feedback drives a net inhibition of both spontaneous and odor-evoked activity in local and output neurons, as predicted by network modeling. Further, cortical GABAergic feedback stimulation separated population odor responses in TCs, but not MCs. At the behavioral level, silencing of cortical GABAergic projections impaired fine odor discrimination of close binary mixture of enantiomers. Lastly, cortico-subcortical GABAergic projections are also observed between the primary somatosensory cortex (S1) and its respective lower-order thalamic nuclei.

## Results

### Anterior olfactory cortex sends GABAergic projections to the OB

To determine whether the AOC sends GABAergic projections back to the OB, in parallel to the well-described glutamatergic projections, we expressed different fluorescent reporters in the GABAergic and glutamatergic populations of the AOC. Using transgenic mice expressing the Cre recombinase under the vesicular GABA transporter VGAT (VGAT-Cre), we employed a conditional genetic approach to restrict expression of eYFP in GABAergic neurons while expressing mCherry in excitatory neurons using the CaMKIIa promoter (Fig. 1a). Both GABAergic and glutamatergic axons were found in the OB but showed different innervation profiles. While GABAergic axons accumulated preferentially in the superficial granule cell layer (GCL) to the mitral

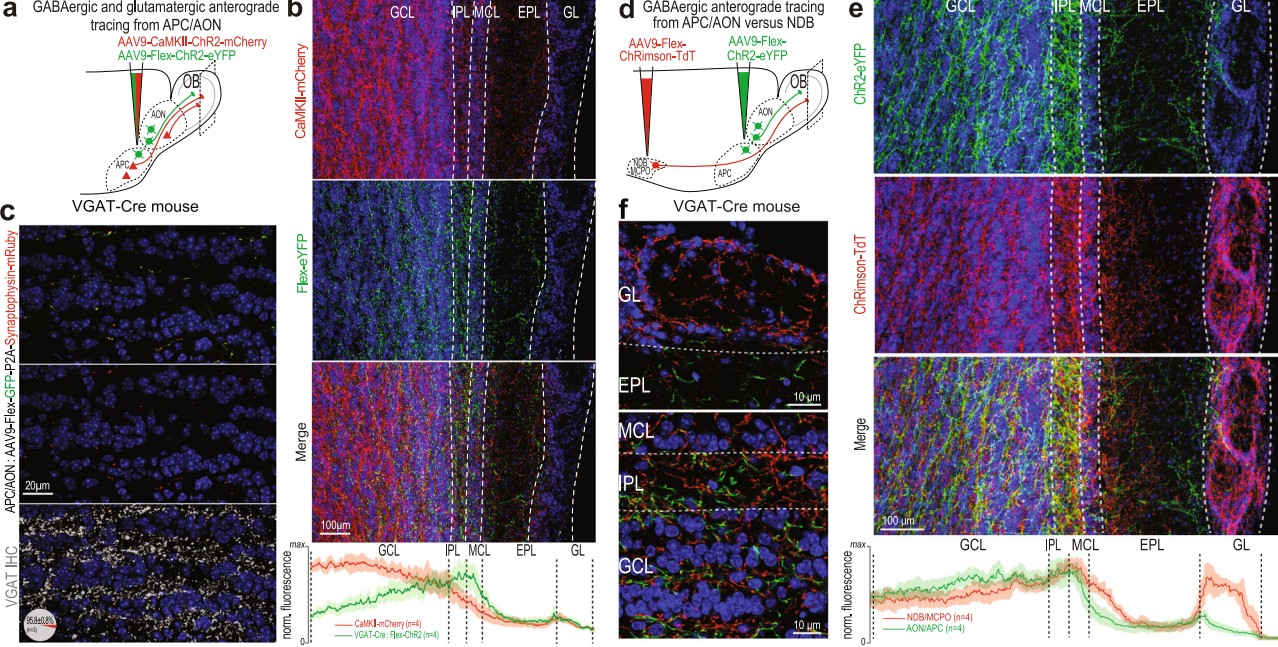

**Fig. 1 | The olfactory cortex sends GABAergic projections back to the OB. a** Viral strategy for comparative anterograde labeling of glutamatergic (CaMKII-ChR2-mCherry) and GABAergic (Flex-ChR2-eYFP) axons in the OB from the AON/APC in a VGAT-Cre mice. **b** Confocal images exhibiting the laminar profile of OB innervation by glutamatergic (top, red) or GABAergic (middle, green) axons from the AON/APC. Bottom, merge. Bottom plot, normalized fluorescence intensity from glutamatergic (red) versus GABAergic (green) axons across OB layers. Mean (solid line) ± sem (shaded). *n* = 4 mice. **c** High magnification of the GCL of VGAT-Cre mice injected with AAV9-Flex-GFP-P2A-Synaptophysin-mRuby in the AON/APC. Top, GABAergic axon shafts (green) and their presynaptic mRuby-positive boutons (red). Middle, Synaptophysin-mRuby channel only. Bottom, immunohistochemistry (IHC) against VGAT is overlayed with the Synaptophysin-mRuby signal. 98 ± 0.8% of the putative

boutons colocalize with VGAT staining (*n* = 3 mice). **d** Comparative anterograde labeling of AON/APC (ChR2-eYFP) versus NDB/MCPO (ChRimson-TdTomato) GABAergic axons in the OB using Cre-dependent AAV injection in VGAT-Cre mice. **e** Confocal images exhibiting the laminar profile of OB innervation by GABAergic axons from the AON/APC (top, green) or NDB/MCPO (middle, red). Bottom, merge. Bottom plot, normalized axon fluorescence intensity from AON/APC (green) versus NDB/MCPO (red) across OB layers. Mean (solid line) ± sem (shaded). *n* = 4 mice. **f** Higher magnification of **e** in the different OB layers. NDB/MCPO and AON/APC GABAergic axons are intermingled but distinct in the OB. Blue, DAPI. GL glomerular Layer, EPL external plexiform layer, MCL mitral cell layer, IPL internal plexiform layer, GCL granule cell layer. Source data are provided as a Source Data file.

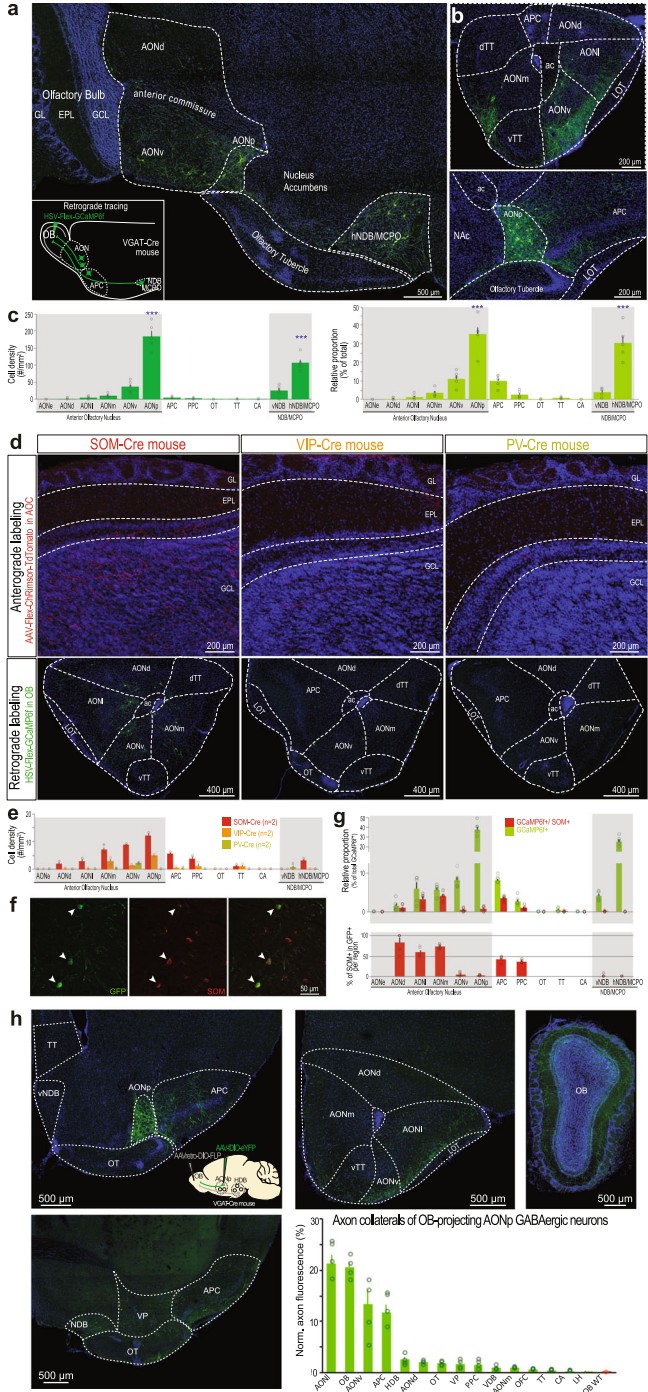

**Fig. 2 | Anatomical and neurochemical identification of the OB-projecting GABAergic cells in the anterior olfactory cortex. a** OB-projecting GABAergic cells in sagittal slice. Inset: schematic of the injection. **b** Coronal slices through the AON (top) and APC/AONp (bottom). **c** Cell density (left) and relative proportion (right) of the OB-projecting GABAergic cells ($n = 2134$ cells from 5 mice). AONp and hNDB/MCPO contain the largest density and proportion of cells (density: one-way ANOVA, $F(12,52) = 75.45$, $p = 10^{-28}$, Tukey multiple comparison test, AON or hNDB/MCPO vs each of the other areas, $p < 10^{-7}$; proportions: one-way ANOVA, $F(12,52) = 40.86$, $p = 10^{-22}$, Tukey multiple comparison test, AON or hNDB/MCPO vs each of the other areas, $p < 10^{-7}$). **d** Anterograde (top, sagittal sections) and retrograde labeling (bottom, coronal sections) of OB-projecting cells in SOM-, VIP- and PV-Cre mice. **e** Density of retrogradely-labeled cells for SOM-Cre ($n = 2$ mice), VIP-Cre ($n = 2$ mice) and PV-Cre mice ($n = 2$ mice). **f** Example SOM labeling of OB-projecting GABAergic neurons in the AON (arrowheads). **g** Top, quantification of co-labeled cells across cortical olfactory regions (1075 cells from 5 mice). Bottom, percentage of SOM+ cells among all the GCaMP+ cells in each region. **h** Selective expression of eYFP in OB-projecting GABAergic neurons of the AOC using a double-conditional strategy. Somatic labeling was largely restricted to the AONp ($85.1 \pm 2.5\%$ of labeled neurons in the AONp, $9.9 \pm 1.4$ in the APC, $3.2 \pm 0.4\%$ in AONv and $1.6 \pm 0.6$ in AONl, $n = 4$ mice). Axons were found in various anterior olfactory cortical areas. OB WT, specificity control (Supplementary Fig. 3d, $n = 2$). Data is mean ± sem across mice. Circles, individual mice. AONl AON lateralis, AONd AON dorsalis, AONv AON ventralis, AONp AON posterioralis, AONm AON medialis, OT olfactory tubercle, dTT dorsal Tenia Tecta, vTT ventral Tenia Tecta, LOT lateral olfactory tract, hNDB/MCPO horizontal limb of the nucleus of the diagonal band/magnocellular preoptic nucleus, vNDB vertical limb of the NDB, CA cortical amygdala, NAc nucleus accumbens, ac anterior commissure. Source data are provided as a Source Data file.

Another identified source of GABAergic inputs to the OB originates from the nucleus of the diagonal band and magnocellular preoptic area (NDB/MCPO)[37,38,45,46]. We directly compared the OB innervation patterns of GABAergic axons from the olfactory cortex vs. NDB/MCPO using conditional virus expression in both brain regions (Fig. 1d). NDB/MCPO and AOC axon innervation patterns were strikingly different in the GL (Fig. 1e, f). While NDB/MCPO profusely innervated the GL[39–43], AON/APC projections were restricted to the internal part of the GL. NDB/MCPO axons also appeared to innervate more the inner part of the external plexiform layer[42] (EPL).

Taken together, these results show that the AOC sends GABAergic axons back to the OB, and these projections are distinguishable from the well-established cortico-bulbar glutamatergic projections and from the basal forebrain GABAergic projections.

### Identity of the long-range GABAergic projections to the OB

To identify the source(s) of cortical GABAergic feedback to the OB, we employed a conditional retrograde labeling approach. A Herpes Simplex Virus (HSV) expressing GCaMP6f in a Cre-dependent manner was injected in the OB of VGAT-Cre mice (Fig. 2a). Retrogradely-labeled cells were found mainly in the AON, APC and NDB/MCPO, and occasionally in the posterior piriform cortex (PPC) and tenia tecta (TT), but not in the olfactory tubercle (OT) − a large striatal GABAergic structure of the olfactory system (Fig. 2a–c). Following unilateral injection, retrogradely labeled cells were found only in the ipsi-lateral, but not contra-lateral side of the injection (Supplementary Fig. 2a). Retrogradely-labeled cells were not uniformly distributed within the AOC. A large proportion of the labeled cells were concentrated in the AON *pars posterioralis* (AONp) − the most caudal part of the AON, located in between the APC and the OT (Fig. 2c). An appreciable fiber tract was often observed between the AONp and the NDB/MCPO (Fig. 2a), yet the former cluster of GABAergic projection neurons was not a rostral extension of striatal or pallidal territory as it was not intermingled with neurons expressing the acetylcholine-synthesizing enzyme ChAT − in contrast to OB-projecting GABAergic neurons of the NDB/MCPO[40,41] (Supplementary Fig. 2b). To confirm these observations based on retrograde viral vectors, we combined conventional retrograde labeling of OB-projecting neurons using cholera toxin

cell layer (MCL), glutamatergic axons were more concentrated in the deep GCL and their density progressively decreased towards the MCL (Fig. 1b). To control for the specific expression of the conditional vectors, we injected the same virus mix in the AOC of wild-type mice and found no expression of the conditional fluorophore eYFP in the AOC (Supplementary Fig. 1a). In VGAT-Cre mice, viral injection in the AOC infected virtually no somata in the GCL ($0.21 \pm 0.11$ neurons/mm², $n = 6$ mice), ruling out any significant AAV transduction of migrating adult-born neurons[23,44] (Supplementary Fig. 1b–c). Further, to confirm the GABAergic nature of the labeled cortical axons in the OB, we injected a conditional virus expressing GFP and Synaptophysin fused with mRuby in the AOC of VGAT-Cre mice. In the OB, synaptohysin-mRuby+ presynaptic boutons colocalized extensively with VGAT immunostaining (Fig. 1c).

subunit-B conjugated to a red fluorophore (CTB) with somatic viral labeling of GABAergic neurons of the AON/APC. Likewise, dually-labeled cells were found scattered in the AOC, with a higher density in the AONp (Supplementary Fig. 2c).

We found both spiny and aspiny neurons in the cortical OB-projecting GABAergic neurons (Supplementary Fig. 3a). We thus set out to substantiate these observations and precise their neurochemical nature. Somatostatin (SOM), parvalbumin (PV) and the vasoactive intestinal peptide (VIP) characterize the vast majority of GABAergic neurons in the cortex and have been reported in largely non-overlapping populations in the AON[47] and APC[48]. To identify the marker preferentially expressed by OB-projecting cortical GABAergic neurons, we first injected an AAV-Flex-ChR2-TdTomato in the AON/APC of SOM-Cre, VIP-Cre or PV-Cre mice. Substantial axonal innervation in the OB was observed in SOM-Cre mice, while very sparse fibers were detected in PV-Cre and VIP-Cre mice (Fig. 2d). Reciprocally, injection of a Cre-dependent retrograde vector in the OB led to a denser number of neurons in SOM-Cre than in VIP-Cre or PV-Cre mice (Fig. 2d, e). However, we did not observe a predominant labeling in the AONp, in contrast to results obtained in VGAT-Cre mice. To confirm this observation, we performed immunostaining against the protein SOM in retrogradely-labeled GABAergic neurons (VGAT-Cre mice; Fig. 2f). In the olfactory cortex, we found that a substantial fraction of OB-projecting GABAergic neurons was co-labeled with SOM in the AONd, AONl, AONm, APC and PPC, but not in the AONv or AONp (Fig. 2g). In SOM-Cre mice, $95.5 \pm 0.5\%$ of the genetically-labeled neurons were dually-labeled with SOM immunostaining, confirming the efficiency of our SOM immuno-histological labeling (Supplementary Fig. 1c). Focusing on the AONp, the densest source of projection neurons, we performed further immuno-histological characterization for GABAergic neuron markers and found that calbindin colocalized more than any other marker tested, yet still to a modest degree (Supplementary Fig. 3b, c).

We also wondered whether OB-projecting GABAergic neurons send axon collaterals elsewhere in the brain. Using a double-conditional strategy based on FLP and Cre recombinase, we specifically restricted the expression of eYFP to GABAergic OB-projecting neurons of the AONp (Fig. 2h, Supplementary Fig. 3d). In addition to the OB, these AONp GABAergic neurons innervate the AONl, AONv and APC, but no sizeable axonal arborization was found in the NDB/MCPO or lateral hypothalamus. In conclusion, inhibitory projection neurons were found scattered in the AOC, with a substantial cluster located in the AONp. These GABAergic neurons innervate preferentially the olfactory system and have limited projections in non-olfactory areas.

### The primary somatosensory cortex also sends GABAergic projections to the somatosensory thalamus

We wondered whether such inhibitory cortical feedback motifs down the hierarchy existed in other sensory systems. In the primary somatosensory cortex (S1), we performed a similar dual ante-rograde labeling of deep-layers GABAergic and glutamatergic neurons (Supplementary Fig. 4a). GABAergic axons were found alongside glutamatergic axons in the lower-order (ventroposterior medial and lateral, VPM and VPL) and higher-order (posteriomedial, POm) somatosensory thalamic nuclei (Supplementary Fig. 4b). GABAergic cortico-thalamic projections intermingled with gluta-matergic projections and did not seem to project to other thalamic territories. These results indicate that corticofugal inhibitory projections might be a more common motif than previously thought in sensory pathways.

### Cortical GABAergic projections target both OB principal cells and interneurons

In the olfactory system, we tested whether cortical GABAergic inputs form functional synapses onto OB neurons and whether GABAergic inputs exhibit target selectivity. Channelrhodopsin-2 (ChR2) was expressed selectively in the GABAergic cells of the AON and APC − the 2 regions consisting of ~95% of the OB-projecting cortical neurons (Fig. 2c) − and whole-cell recordings were obtained in acute OB slices (Fig. 3a). In responsive neurons (74/177), light stimulation of GABAergic axons evoked short-latency post-synaptic currents (PSCs; Supplementary Table 1), consistent with monosynaptic events. PSCs were characterized by a current-voltage linear relationship reversing at the reversal potential for chloride (~-75 mV; Fig. 3b) and were unchanged in presence of the AMPA receptor antagonist NBQX (10 μM) but were completely abolished by the GABA$_A$ receptor antagonist Gabazine (SR95531, 10 μM; Fig. 3c, d), confirming the GABAergic nature of the PSCs.

We next investigated the target specificity of these cortical GABAergic inputs. OB neurons were classified according to their intrinsic properties, morphology, soma size, and laminar position in an OB slice (Fig. 3e; see Methods). Inhibitory PSCs (IPSCs) were detected in roughly half of the excitatory neurons tested (MCs, TCs, and eTCs). Among GCL inhibitory neurons, we found a stronger connectivity and larger current amplitude in dSACs compared to GCs (Fig. 3f, g), reminiscent of previous observations with glutamatergic feedback[16,23]. In contrast, glomerular inhibitory neurons were spared, consistent with the lower density of axons observed in the glomerular region (Fig. 1b, e). Importantly, none of the recorded GCs exhibited ChR2-mediated inward currents (0/64) and IPSC kinetics in MCs and TCs were not consistent with GC-mediated inhibition[49]. To confirm a direct synaptic connection from cortical GABAergic neurons, we performed rabies-based retrograde monosynaptic tracing from MCs and TCs[50] (Supplementary Fig. 5a). Monosynaptic retrogradely labeled neurons were observed in the AOC and some colocalized with the GABA syn-thesizing enzymes glutamic acid decarboxylase 67 (GAD67; Supplementary Fig. 5a). We also found monosynaptic retrogradely labeled GABAergic cells in the AOC with GCL GABAergic interneurons as the starter cell population (Supplementary Fig. 5b). Thus, cortical GABAergic feedback provides direct functional inputs to a variety of OB neurons, both inhibitory and excitatory.

### Cortical GABAergic inputs influence OB network oscillations

Long-range cortical GABAergic neurons have been repeatedly pro-posed to play a role in tuning network oscillations and synchronizing distant brain areas[7]. To explore the functional role of cortical GABAergic projections to the OB, we first investigated to what extent this cortico-bulbar GABAergic pathway can influence oscillatory regimes in the OB. Network oscillations are prominent in the OB and can be subdivided into different frequency bands, theta (1–12 Hz), beta (15–40 Hz) and gamma oscillations (40–100 Hz). We coupled pat-terned optogenetic stimulation of cortical GABAergic axons at differ-ent frequencies with local field potential (LFP) recordings in the OB of awake VGAT-Cre mice to investigate if the OB network respond maximally, or resonate, at specific driving frequencies (Supplementary Fig. 6a). Spontaneous beta oscillations, but not theta or gamma, were amplified upon light stimulation of GABAergic cortical axons. Specifi-cally, light stimulation at 33 Hz increased beta band frequencies, while stimulation at 10 or 66 Hz had no significant effect (Supplementary Fig. 6b, c). Thus, long-range cortical GABAergic projections tune OB oscillations and specifically enhance beta oscillations when entrained at beta frequencies.

### Optogenetic activation of cortical GABAergic inputs inhibits OB GCL interneurons in vivo

To assess the functional impact of the cortical GABAergic feed-back on its main target layer (GCL), we employed fiber photo-metry in freely moving mice. The volume fluorescence of GCaMP6f-expressing GCL GABAergic neurons was continuously

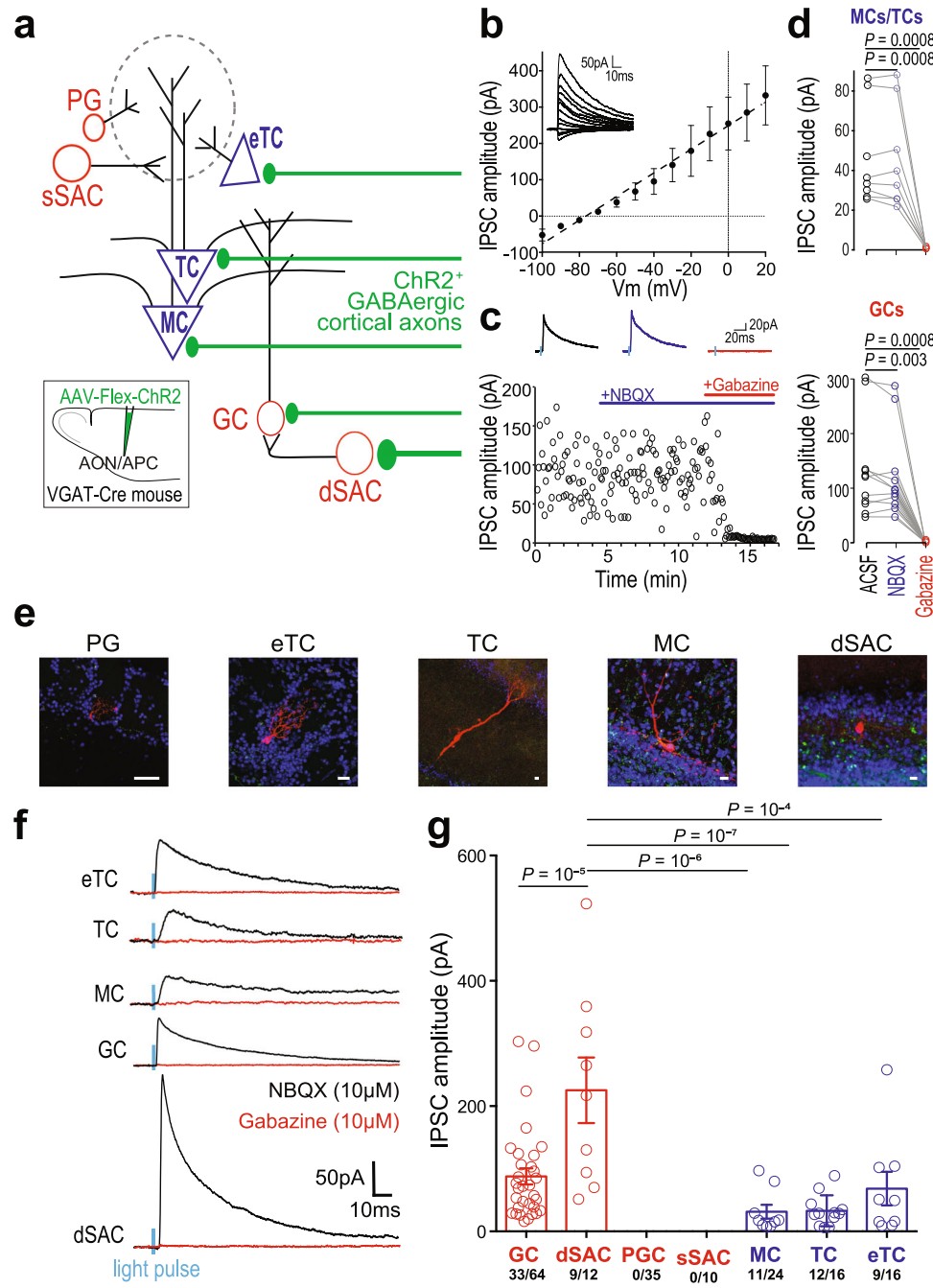

**Fig. 3 | Cortico-bulbar GABAergic axons form functional synapses with inhibitory and excitatory neurons in the OB. a** Recording schematic. Periglomerular (PG) cells, external tufted cells (eTCs), superficial short-axon cells (sSACs), tufted cells (TCs), mitral cells (MCs), granule cells (GCs) and deep short-axon cells (dSACs) were patched and GABAergic feedback axons expressing ChR2 (inset) were light-stimulated (2 ms). Width of the axon shafts indicates connection probability. **b** PSCs recorded in GCs (*n* = 4 cells) at different holding potentials, in the presence of NBQX (10 μm). Responses reversed at -–75 mV, consistent with GABAergic receptor activation. In the rest of the figure, $V_c$ = 0 mV. **c** IPSC amplitudes in an example GC in the presence of ACSF, NBQX or Gabazine. **d** PSCs were resistant to NBQX application (10 μM) but completely abolished by Gabazine (SR95531, 10 μM) in MCs/TCs (top) and GCs (bottom; GCs: One-way ANOVA, $F_{(2,33)}$ = 9.82, $p = 10^{-4}$,

with Tukey's post-hoc test, *n* = 11; MCs/TCs: One-way ANOVA, $F_{(2,21)}$ = 12.56, $p$ = 0.0003 with Tukey's post-hoc test, *n* = 8). **e** Representative examples of the patched neurons analyzed in **f**. Neurons were first visualized and identified online. GCs and sSACs morphology were not reconstructed *post hoc* because pipette withdrawal after recording did not preserve the integrity of the cells. Scale bars are 10 μm. **f** Representative trial-average IPSCs in cells recorded at $V_c$ = 0 mV. Responses were systematically blocked in Gabazine. Blue, light-pulse; black, recordings in NBQX (10 μM); red, recordings in Gabazine (SR95531, 10 μM). **g** IPSC amplitudes across the cell tested (One-way ANOVA, $F_{(4,68)}$ = 11.23, $p = 10^{-7}$, with Tukey's post-hoc test). Blue bars, excitatory neurons; Red bars, inhibitory neurons; circle, individual cell. Data presented as mean ± sem. Source data are provided as a Source Data file.

recorded using an optic fiber implanted above the GCaMP6f injection site, while AOC GABAergic projections were light-stimulated in the ventral OB (Fig. 4a). Using the red-shifted opsin ChRimson, we could independently control GABAergic

axons and avoid cross-excitation of GCaMP6f[27,51]. ChRimson light stimulation at 10 Hz, 33 Hz or with a continuous light step (CL) produced a global reduction of spontaneous activity in GCL GABAergic neurons while red light stimulation per se did not alter

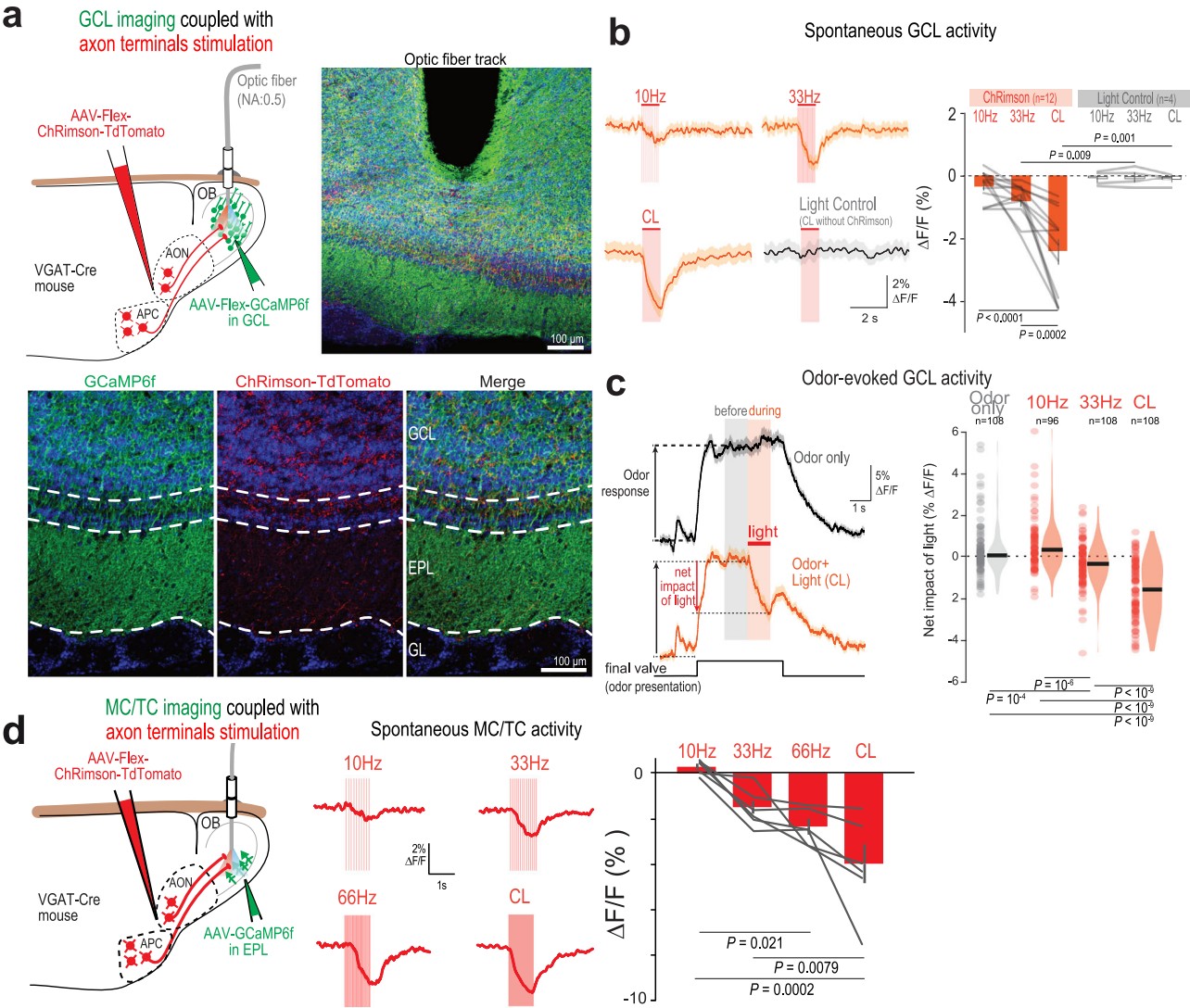

**Fig. 4 | Cortico-bulbar GABAergic axons induce net inhibition onto GCL neurons and MC/TC populations in vivo. a** Cortical GABAergic axons were stimulated in the OB of freely moving mice. Top, Optic fiber for cortical GABAergic axon optogenetic stimulation (1 s) and GABAergic interneuron calcium imaging in the ventral GCL. Bottom, GCaMP6f expression across OB layers and ChRimson in AON/APC GABAergic axons. Blue, DAPI. **b** Stimulation of cortical GABAergic axons decreased GCL interneuron spontaneous population activity. Left, example trial-averaged responses to different light stimulation patterns. CL continuous light. "Light control" is CL illumination in mice lacking ChRimson expression. Right, Mean fluorescence change in response to light stimulation (RM-One-way-ANOVA with Tukey's post-hoc test, $F(2,22) = 21.28$, $P < 0.0001$). Data presented as mean ± sem; gray lines, individual mice. **c** Stimulation of cortical GABAergic axons decreased GCL interneuron odor-evoked population activity. Left, fluorescence signals during odor presentation only (black) and odor presentation coupled with stimulation of cortical GABAergic axons (orange). Traces are trial-averaged example responses. Mean ± sem are represented. Right, Net impact of light on odor responses across

light stimulation protocols (10 Hz, 33 Hz, CL) compared to no light stimulation ("odor only"). The net impact of light was measured as the difference between the mean odor-evoked response in the 1 s window before versus during light stimulation. Population odor response were inhibited by light stimulation of cortical GABAergic axons and inhibition magnitude scaled with the stimulation strength (One-way-ANOVA with Tukey's post-hoc test, $F(3,416) = 62.98$, $p = 10^{-3}$). Violin plots are ks density estimates; black bar, median; circle, individual odor-recording site pair. **d** MC/TC population responses to cortical GABAergic axon optogenetic stimulation in the OB of freely moving mice utilizing fiber photometry (left). Middle: Representative trial-averaged traces of light-evoked inhibitory responses in MCs/TCs using different stimulation patterns. Right: Stimulation of cortical GABAergic axons produced increasing inhibition with increasing stimulation frequency (RM-One-way ANOVA with Tukey's post-hoc test, $F(3,15) = 12.20$, $P = 0.0003$, $n = 6$ recording sites in 4 mice). Data presented as mean ± sem; Gray, individual recording sites. Source data are provided as a Source Data file.

spontaneous activity in control animals (expressing GCaMP6f in the GCL, but not ChRimson in GABAergic feedback; Fig. 4b). Increasing light stimulation frequency up to a continuous light pulse induced increasing inhibition of GCL neuron activity (Fig. 4b). This feature was still observed 1 s after light stimulation offset (Supplementary Fig. 7a).

We next investigated the impact of GABAergic feedback stimulation on odor-evoked activity in the GCL. Odor stimulation induced a strong population response in GCL neurons[35] (Fig. 4c). We quantified the net decrease in Ca²⁺ activity relative to the period before light

stimulation, within the same odor response (Fig. 4c). When compared with odor response dynamics without light stimulation ("odor only"), GABAergic feedback light stimulation effectively dampened odor responses with 33 Hz and CL, but not 10 Hz, stimulation patterns (Fig. 4c). CL light inhibition of odor responses outlasted the light stimulation period: 1 s after light stimulation offset, CL still caused a sustained inhibition of the odor-evoked activity (Supplementary Fig. 7a). Thus, cortical GABAergic axon stimulation efficiently drives inhibition of both spontaneous and odor-evoked activity in GCL GABAergic neurons.

### Cortical GABAergic inhibition to OB output neurons scales with the frequency of stimulation

Given that cortical GABAergic feedback synapses both on GCL interneurons and OB principal neurons, we wondered how GABAergic axon stimulation impacts MC/TC activity in vivo. We targeted our fiber photometry recordings to MCs/TCs and found that light stimulation of cortical GABAergic axons inhibited MC/TC activity (Fig. 4d). Increasing light stimulation frequency up to a continuous light pulse induced increasing inhibition of MC/TC activity (Fig. 4d). In addition to GABAergic feedback-mediated inhibition, the AOC inhibits MCs and TCs through a disynaptic pathway: glutamatergic feedback drives GCs and PG cells which in turn inhibit MCs and TCs[16,23,27] (Supplementary Fig. 7b,c). We thus wished to compare the differential frequency recruitment of inhibition resulting from stimulation of the monosynaptic cortical GABAergic vs. disynaptic cortical glutamatergic pathway. While GABAergic projection stimulation drove increasing MC/TC inhibition with increasing stimulation frequency, the MC/TC inhibition evoked by light stimulation of glutamatergic projection peaked at 33 Hz, implementing a bell-shaped, low-pass filtering of the excitatory drive (Supplementary Fig. 7c).

### Computational modeling of cortical inhibitory feedback on OB network

Optogenetic stimulation of GABAergic feedback axons results in a net inhibition of the two recurrently connected excitatory (MCs and TCs) and inhibitory neuron populations (GCs). To tackle this apparent paradoxical effect and further explore the outcomes of GABAergic feedback on OB neurons, we built a population model of the OB based on our experimental results. Our model consisted of reciprocally connected excitatory (MCs/TCs) and inhibitory subnetworks (GCs). GCs additionally receive inhibitory inputs from an additional inhibitory population (dSACs; Fig. 5a). We computed the steady state of the network without and with GABAergic feedback over a range of GC-MC/TC synapse and feedback strength. The relative strength of the feedback on MCs/TCs and GCs was derived from our slice recording data (Fig. 3). This parsimonious model showed that GABAergic feedback resulted in a net inhibition on both MCs/TCs and GCs (Fig. 5c, d). Importantly, this effect was observed for a range of values consistent with GC and MC/TC firing rate observed in vivo (Fig. 5b). We additionally observed that increasing GABAergic feedback stimulation strength produces stronger inhibition on MCs/TCs and GCs, which mirrors our in vivo data with increasing stimulation frequency (Fig. 4).

### Activation of cortical GABAergic inputs enhances the distance in TC population odor responses

We next investigate whether MCs and TCs are similarly inhibited by GABAergic feedback. To resolve individual MCs and TCs, we performed two-photon Ca$^{2+}$ recordings in awake, head-fixed mice. Based on our fiber photometry data, we used 33 Hz and CL stimulation to probe optimal regimes of cortical inhibitory drive. We therefore switched to the opsin ChIEF because it yields stronger currents upon long light pulses and a more naturalistic drive of GABAergic axons (Fig. 6a)[52]. The axon terminals were light-stimulated through the microscope's objective, while the photomultiplier tube (PMT) shutter was closed and reopened 50 ms before and after light onset and offset. Due to the slow kinetics of GCaMP6s, we could capture Ca$^{2+}$ events following the light offset and reopening of the shutter (Fig. 6c). MCs and TCs were identified based on the recording depth and the cytoarchitecture of each OB layer[32,53–56].

Light stimulation of cortical GABAergic axons induced a significant reduction of spontaneous activity in the large majority of the MCs and TCs, both at 33 Hz and with CL (Fig. 6d; Supplementary Fig. 8a, b). In MCs, CL stimulation significantly reduced activity in a larger fraction of cells compared to 33 Hz stimulation, and inhibitory response magnitudes were larger. These differences were not

observed in TCs. The observed inhibition was not an artifact of closing and reopening the PMT shutter. Indeed, in 'shutter control' trials, the number of cells exhibiting a significant change in activity was at statistical chance level, and the change in activity was significantly smaller, by an order of magnitude, than for light-stimulation trials ('shutter control' trials: trials with shutter closing, but no light presented, Fig. 6d; Supplementary Fig. 8b). We additionally controlled for an effect of blue light illumination per se in control animals that did not express ChIEF ('light control'). A small, yet above chance proportion of MCs and TCs showed significant reduction of activity, consistent with previous reports[57], and that proportion increased with CL (Supplementary Fig. 8b). However, the magnitude of the light-induced inhibitory responses was 10-fold bigger in ChIEF-expressing animals compared to control animals not expressing ChIEF and therefore cannot significantly contribute to the reported effect (Fig. 6d).

GABAergic feedback inputs reduced spontaneous activity in the OB, but how does it influence incoming sensory feedforward information? Odor stimulation induced both inhibitory and excitatory responses in MCs and TCs, but with different relative proportions, as previously reported (Fig. 6e)[55,58,59]. In odor-responsive cells, stimulation of GABAergic cortical axons induced a reduction of excitatory odor responses and a greater inhibition of inhibitory odor responses. This was true across both cell types and light stimulation patterns. The magnitude of the light-evoked inhibition and the odor responses were not correlated, resulting in linear subtraction of the odor-evoked activity (Fig. 6f). We also compared the impact of cortical GABAergic stimulation on spontaneous and odor-evoked activity at the individual neuron level. We found little correlation between the magnitude of the light-driven responses in spontaneous versus odor-evoked activity in both MCs and TCs (Supplementary Fig. 8c). For both cell types, light-driven inhibition was slightly stronger during spontaneous activity (Supplementary Fig. 8c).

To evaluate the effect of cortical GABAergic axon stimulation on the separation of odor representation within MC and TC populations, we calculated the Euclidean distance between population responses to either different ('Between odors') or the same odor ('Within odor'; Fig. 6g). Consistent with a linear subtraction of the odor responses in MCs and TCs, light did not significantly alter the pairwise distance between population responses to a given odor ('Within odor' design; Fig. 6g). In contrast, in the 'between odors' design, light stimulation increased the distance in population odor representation of TCs, but not MCs (Fig. 6g). This shows that stimulation of GABAergic axons specifically increases the difference in the representation of two different odors in TCs.

Another recipient of GABAergic projections in the OB is the internal part of the GL. We thus targeted our recordings to juxtaglomerular cells (JG cells, at the transition between the GL and EPL). As for MCs and TCs, CL stimulation of cortical GABAergic axons inhibited spontaneous activity of JG cells (Supplementary Fig. 8d). Odor stimulation drove mainly excitatory responses in JG cells, as reported previously[60] (Supplementary Fig. 8e). In the odor-responsive population, light stimulation of cortical GABAergic axons induced a linear reduction of the odor-evoked activity (Supplementary Fig. 8f). As seen in MCs and TCs, inhibition of spontaneous and odor-evoked activities was only weakly correlated (Supplementary Fig. 8f).

### Silencing cortical GABAergic outputs to the OB affects fine odor discrimination

Since GABAergic feedback modulates odor responses in OB output neurons, we next examined whether it could contribute to olfactory perception. To specifically inhibit cortical GABAergic feedback to the OB during the extent of an olfactory-guided task, we employed a pharmacogenetic approach. We expressed the inhibitory designer receptors exclusively activated by designer drugs (DREADD) hM4Di specifically in GABAergic AON/APC neurons. The axon terminals in the

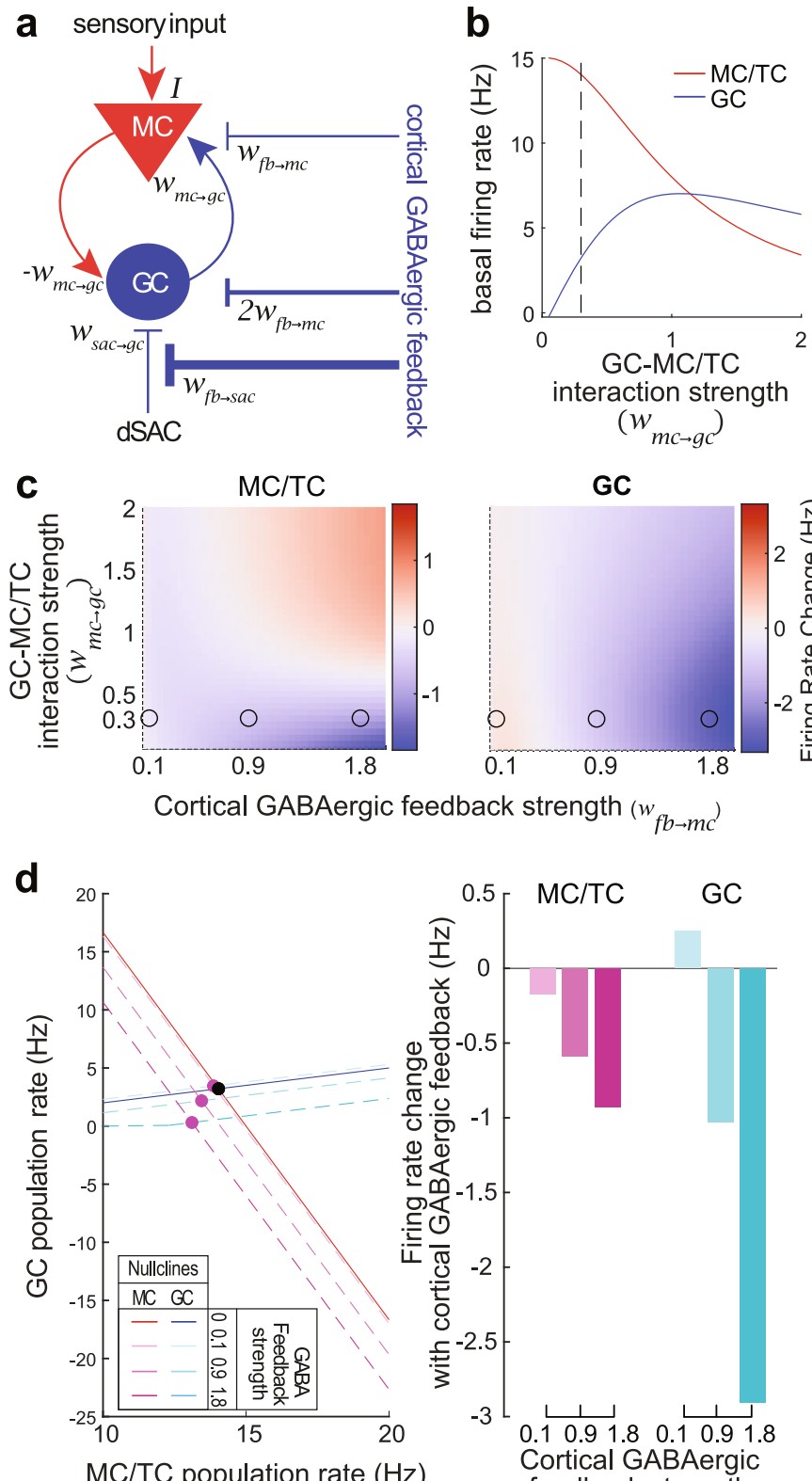

OB were selectively silenced by the local application of exogenous ligand clozapine-*N*-oxide (CNO), thereby sparing activity to other targets of the projections (Fig. 7a). The mice odor detection and discrimination thresholds were evaluated in a go/no-go task using carvone and limonene enantiomers (Fig. 7b). Detection threshold was assessed by diluting each day by a factor of 10 the two enantiomers to detect (from 1% to 0.0001% dilution). Discrimination threshold was assessed by presenting binary mixtures of the enantiomers, with a progressive and symmetric increase of the proportion of one into the other each day (from pure enantiomers discrimination, i.e., 100:0 vs. 0:100, to discrimination of mixtures with 55:45 vs. 45:55 enantiomer ratios).

Selective blocking of cortical GABAergic feedback had no effect on the detection of carvone or limonene enantiomers, even for very

**Fig. 5 | A parsimonious population model recapitulated the inhibition of both MC/TC and GC populations upon cortical GABAergic feedback stimulation.** **a** Schematic of a population model with excitatory (MCs/TCs, red) and inhibitory (GCs, blue) subnetworks. MC/TC population additionally receives external excitatory inputs (odor stimulation) and GC population inhibitory inputs from dSACs. Cortical GABAergic feedback directly inhibits MC/TC and GC populations and is twice stronger on GCs. It also shunts the inhibitory input from the dSAC to GC population. **b** Basal firing rate of MC/TC and GC populations when the system is at the equilibrium (fixed point). Dashed line is the value $W_{mc/gc}$ chosen in **d**. **c** Change in the firing rate upon cortical GABAergic feedback stimulation for a range of GC-MC interaction and feedback strength. Dots are the strength of the MC/TC-GC and

GABAergic feedback used in **d**. **d** Fixed point analysis of the system without and with different strength of cortical GABAergic feedback stimulation. Left, steady state in MC/TC and GC population is represented by the nullclines, (MC, solid red line without stimulation, dashed lines in shades of purple with feedback stimulation; GC, solid blue line without stimulation, dashed lines in shades of blue with feedback stimulation). Equilibrium of the system is obtained by the crossing of the nullclines (fixed point: without stimulation, black; with stimulation, purple). GC firing rate is rectified to avoid negative firing rates. Right, quantification of the change in firing rate for the MC/TC and GC populations for the different strength of the GABAergic feedback stimulation. Source data are provided as a Source Data file.

low odor concentration (Fig. 7c). In contrast, the discrimination of very similar binary mixtures of enantiomers was impaired. Indeed, a significant decrease in discrimination was observed for limonene enantiomers. A similar reduction in performance was observed for carvone enantiomers, although it did not reach statistical significance (Fig. 7c). When analyzing discrimination performances collectively for both pairs of enantiomers, blocking cortical GABAergic feedback did reduce fine odor discrimination performances (Fig. 7c). No significant difference in odor sampling time was observed (Supplementary Fig. 9). Altogether, the behavior data shows that silencing cortical GABAergic axon outputs to the OB impairs fine odor discrimination.

## Discussion

This study reveals the presence of GABAergic feedback projections from the primary olfactory cortex to the OB, with the AONp particularly densely packed with projecting neurons. We showed that cortical GABAergic feedback to the OB forms functional synapses with both GCL interneurons and principal output neurons – MCs and TCs. In awake mice, stimulation of this inhibitory feedback diminished both spontaneous and odor-evoked activities in GCL interneurons as well as in MCs and TCs. This global inhibition of the OB network was also captured by a computational model based on our experimental results. Interestingly, cortical GABAergic feedback separated odor population responses specifically in TCs, but not in MCs. Silencing of cortico-bulbar inhibitory axons altered performances during a fine odor discrimination task. Finally, we reported an analogous cortico-fugal inhibitory projection in the somatosensory system, suggesting a possible extension of our observations to other sensory systems.

As a first step to investigate the function of cortical GABAergic feedback in sensory systems, we manipulated these inputs collectively by labeling both the AON and APC, the two regions consisting of ~95% of the cortical GABAergic OB-projecting neurons. Using genetics, immunolabeling and pharmacological tools, we showed that AOC GABAergic neurons contribute to the cortico-bulbar pathway. Using anterograde tracing, we show that GABAergic cortico-bulbar axons terminals 1) express the VGAT marker, 2) have a distinct laminar OB innervation profile compared to cortical glutamatergic or basal forebrain GABAergic projections, 3) do not result from a 'leak' in conditional viral expression or from a direct viral transduction of OB interneurons. We confirmed the existence of OB-projecting GABAergic neurons using four retrograde tracing methods (conditional HSVs, a double-conditional AAV approach, monosynaptic rabies tracing and conventional CTB-based retrograde). Electrophysiological recordings coupled with pharmacology confirmed that GABAergic AOC neurons form monosynaptic GABAergic synapse with OB neurons. The slightly longer IPSC latencies and slower kinetics in MCs and TCs (Fig. 3f, Supplementary Table 1) are consistent with input on electrotonically remote dendrites, presumably apical dendrites in the glomerular layer which are innervated by cortical GABAergic axons (Fig. 1b)[49]. Lastly, data from our photometry and electrophysiology experiments – lack of GCL interneuron excitation and the different frequency recruitment of MC/TC population inhibition upon light-stimulation of cortical GABAergic versus glutamatergic axons – prove the specificity

of our experimental approach and exclude a possible cross-reactivity with cortical glutamatergic axons.

OB-projecting GABAergic neurons originate from various olfactory cortical areas and express different neurochemical markers. We identified a dense cluster of GABAergic projection neurons in the AONp (at the border between the AONv, APC and OT). Early non-specific retrograde labeling studies had already identified a cluster of OB-projecting cells in the AONp in hamsters[19,61], rats[20,25] and mice[18,38,62,63], yet their neurochemical content had not been specified. Recently, a study identified a cluster of lateral hypothalamus-projecting GABAergic neurons, presumably from the same region (coined ventral olfactory nucleus)[64], suggesting that the AONp could be a hub for broadcasting inhibition to olfactory and non-olfactory brain regions. Our data indicates that OB- and lateral hypothalamus-projecting GABAergic neurons are separate populations (Fig. 2h) and further work should decipher the interplay between these two populations and whether they fulfill different functions.

Stimulation of cortical GABAergic feedback produced a net inhibition in both GCL interneuron and MC/TC populations. This observation – a phenomena akin to a "paradoxical" effect observed in cortical networks[65]– is supported by a modeling approach where the reciprocally connected excitatory (MC/TC) and inhibitory subnetworks (GC) are both inhibited by GABAergic feedback stimulation. Our parsimonious network model also reproduces the frequency-dependent inhibition magnitude observed in our GCL and MC/TC fiber photometry data. We reasoned that upon weak cortical GABAergic feedback stimulation the reduction of the inhibitory drive from dSACs might counteract the direct inhibition from cortical feedback onto GCs. This interpretation is corroborated by work from Labarerra et al.[66] showing that GCs are under tonic GABAergic inhibition in the awake state. Importantly, our model reproduces our experimental data with MC/TC-GC connectivity weight values producing physiological firing rates in both the excitatory and inhibitory subnetworks (low firing rate in GCs[66–69]; ~15 Hz firing of MCs/TCs[27,70–72]; Fig. 5d).

At the functional level, we showed that silencing cortical GABAergic feedback axons disrupted fine sensory discrimination of similar odor mixtures, adding evidence for a role of corticofugal projections in sensory detection and discrimination[73]. Several non-exclusive mechanisms could account for the functional impact of cortical GABAergic feedback. First, GABAergic feedback facilitates beta band oscillations when stimulated at beta frequency. Beta oscillations have been shown to emerge during odor discrimination learning and require intact communication between the OB and the olfactory cortex[71,74]. Moreover, precise spike timing of MCs and TCs relative to OB oscillations is critical for coding of odor intensity[70,75–77], odor identity[78] and increases during olfactory learning[79]. Thus, altering the tightly regulated spike-field coherence could be a mechanism through which cortical GABAergic feedback directly shape odor discrimination. In the future, it would be interesting to address whether stimulating cortical GABAergic feedback at different phases of the sniff cycle differently impacts MC/TC activity and behavior. Second, cortical GABAergic feedback can modulate the time-window for integrating cortical excitatory inputs. Cortical glutamatergic axons drive

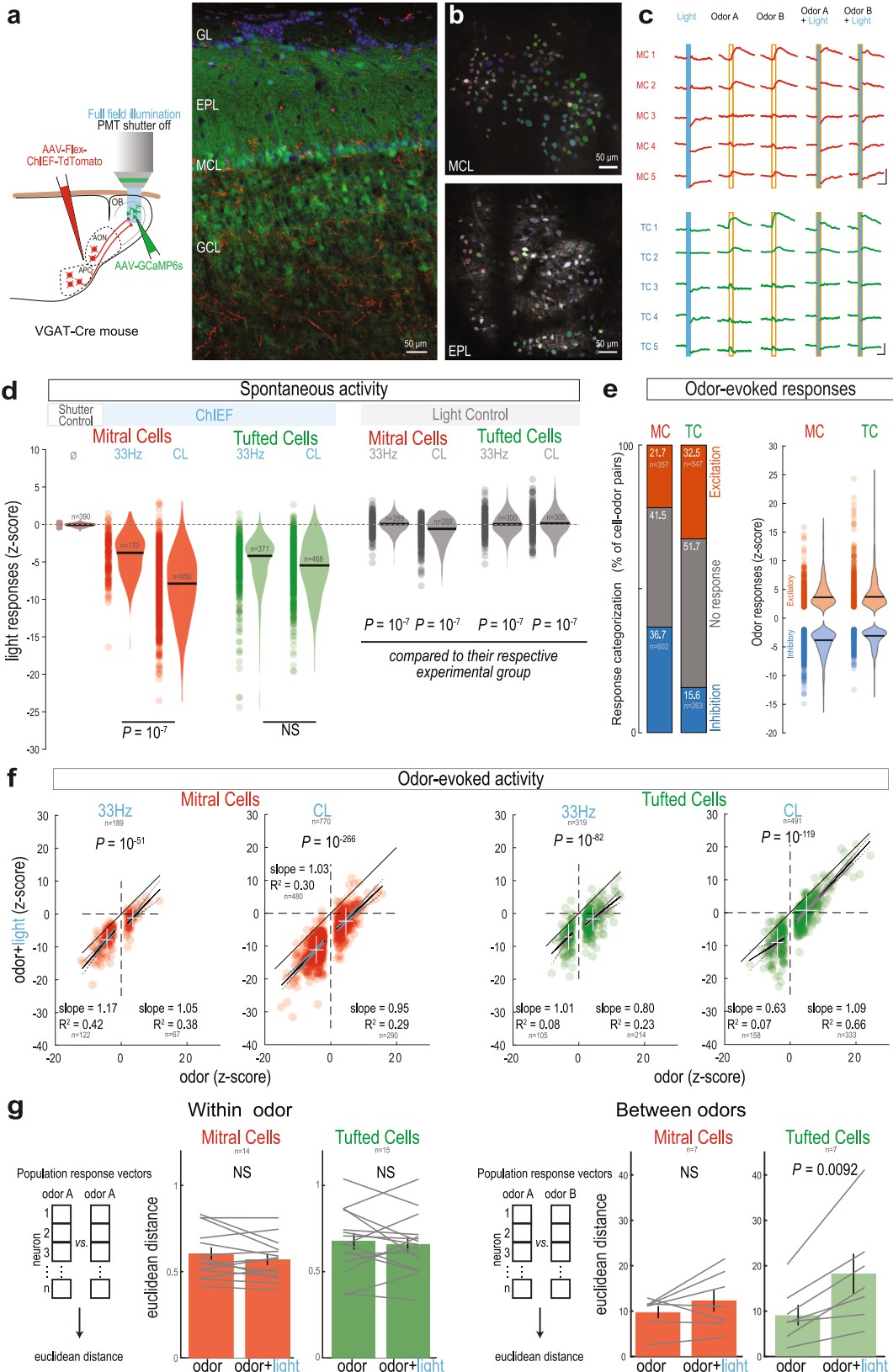

excitation in MCs/TCs shortly followed by disynaptic inhibition[16,23], and GABA_B receptors activation specifically at cortical glutamatergic axon-to-GCs terminals can relax the temporal window for integration of excitatory inputs by MCs/TCs[27]. Cortical GABAergic axons could be the source of GABA activating GABA_B receptors. We reason that this heterosynaptic modulation of glutamatergic inputs could benefit from

a fine temporal regulation in the cortex. Cortical GABAergic feedback is thus ideally positioned to modulate the timing for integrating cortical excitatory inputs[33]. Third, we reported that GABAergic feedback stimulation separated odor population responses specifically in TCs. Since MCs and TCs showed similar connectivity with cortical GABAergic inputs (Fig. 3), differential impact on MCs and TCs could

**Fig. 6 | GABAergic feedback axon stimulation inhibits TC and MC activity in vivo. a** Two-photon imaging of TCs and MCs coupled with optogenetic stimulation of GABAergic cortical axons in awake mice. Right, representative examples of post-hoc confocal image showing the GCaMP6s (green) expression in TCs and MCs along with ChIEF-TdTomato[+] GABAergic axons (red) and further analyzed in **d**. Blue, DAPI. **b** Representative examples of pseudo-colored masks from MCs and TCs obtained by imaging in the MCL (top) and EPL (bottom) and further analyzed in **d**. **c** Example traces of MCs (red, top) and TCs (green, bottom) during light (blue shaded box), odors (orange contoured box), and odor and light simultaneous stimulation (blue shaded and orange contoured box). Scale bars, 5% ΔF/F and 2 s. **d** Light impact on MCs (red) and TCs (green) spontaneous activity in the presence (left) or absence (control, right) of ChIEF. Light illumination was either pulsed (33 Hz) or continuous (CL). 'Shutter control': closing/reopening the shutter without light stimulation. Stimulation of ChIEF[+] GABAergic cortical axons produced significantly greater inhibition than the respective control stimulation without ChIEF (One-Way ANOVA, $F(10,3865) = 456.7$, $p = 0$, Tukey's post-hoc test). In MCs, but not

TCs, ChIEF CL stimulation produced greater inhibition than 33 Hz stimulation (two-sided t-test). Note that CL control in MCs is significantly greater than 'shutter control' (two-sided t-test, $P = 0.02$; $P > 0.05$ for all other comparisons). **e** Categorization (left) and magnitude (right) of the odor-evoked responses in MCs and TCs. In **d** and **e**, violin plots are estimated ks-density from the data, black line is the median and circles represent individual cells. **f** MC (left) and TC (right) responses to simultaneous light and odor stimulation versus odor stimulation only (two-sided paired t-test). Circle, individual responsive cell-odor pair; white cross, mean ± s.d. for excitatory and inhibitory odor responses separately; solid line ± dashed lines, linear fit and 95% confidence interval. **g** Intra-odor Euclidean distance (distance between the population responses to the same odor; "Within odor) and inter-odors Euclidean distance (distance between the population responses to the two odors, "Between odors"; two sided paired t-test) in odor responsive neurons. Data presented as mean ± sem. Gray lines, paired measurements from same recording site. Source data are provided as a Source Data file.

arise from differential intrinsic properties, local connectivity[80], or odor response properties[70,81]. Alternatively, MCs and TCs, or the respective GC populations they connect to (superficial vs. deep GC), could be connected distinctly to GABAergic cortical neurons (different cortical regions preferentially targeting MCs versus TCs and/or different types of GCL GABAergic neurons) and could engage distinct functional loops with their respective cortical targets[82]. Either way, by separating TC population representation of different odors, GABAergic feedback stimulation can possibly enhance the discriminability capacity of a downstream decoder. Interestingly, in contrast to GABAergic feedback, manipulating glutamatergic feedback has been reported to alter the similarity of odor representation in MC, and not TC populations[32]. This suggests that GABAergic and glutamatergic cortico-bulbar projections may have distinct network effects and roles in olfactory behavior. Cortico-bulbar GABAergic projections also differ from basal forebrain GABAergic projections, the latter targeting specifically local GL and GCL interneurons[39–41], resulting in a bidirectional modulation of MCs —switching from an inhibitory to disinhibitory net effect in the presence of odors— as well as a reduction of gamma oscillations[40,42]. Altogether, controlling the proper establishment of sensory-evoked network oscillations, modulating the time-window for cortically-driven excitation and separating representation of odor responses are three mechanisms through which GABAergic feedback can directly shape early sensory processing and odor perception.

Our observations highlight the advantage of direct cortical inhibitory projections. In cortico-thalamic and cortico-bulbar circuits, corticofugal glutamatergic projections produce disynaptic inhibition onto glutamatergic neurons through a GABAergic relay. In the paleocortex (olfactory system), this relay is mediated by local interneurons[16,23], while in the neocortex GABAergic relay neurons are located in the reticular thalamic nucleus[83]. In both paleo and neocortices, GABAergic relay neurons seem to implement band-pass filtering of the cortical glutamatergic drive. In the OB, stimulating cortical glutamatergic projections yields optimal inhibition of MCs and TCs in the beta range (20–40 Hz) and decreases with faster stimulation regimes[27]. Similar frequency-dependent effect also takes place in thalamic nuclei: low-frequency cortico-thalamic axon stimulation suppresses thalamic activity while high-frequency stimulation enhances it[83,84]. In contrast, the strength of the inhibition driven by direct cortical GABAergic axons in MCs and TCs increased with increasing stimulation frequency and was faithful to even high frequency regimes (Fig. 4d). Similarly, in the thalamus, direct extrathalamic GABAergic innervation from subcortical nuclei display a high fidelity to fast stimulation regimes[85]. We also report cortico-thalamic GABAergic projection in the somatosensory system, suggesting that corticofugal GABAergic projections might be a common motif in sensory systems (Supplementary Fig. 4). Further studies will investigate whether this projection is also able to follow fast stimulation frequencies. In

addition, our experimental data and computational model showed that cortico-bulbar inhibitory inputs and their broad connectivity are in position to drive a global inhibition of the downstream network, limiting and preventing disinhibitory events on output neurons. Such a global network control triggers a synchronized reset of the network and may be engaged during specific brain states such as sleep[85].

## Methods

### Animals

Adult (8–10 weeks at the time of injection) male and female VGAT-Cre (heterozygotes, Slc32a1[tm(cre)Lowl], MGI ID: 5141270), SOM-Cre (Ssttm2.1(cre)Zjh, MGI ID: 4838416), VIP-Cre (Viptm1(cre)Zjh, MGI ID: 4431361), PV-Cre (Pvalbtm1(cre)Arbr, MGI ID: 3590684), Tbet-Cre (Tg(Tbx21-cre)1Dlc, MGI ID: J203355)[86] and C57BL/6JRj mice (Janvier Labs) were used in this study. Animal were kept in individually ventilated cages, in a 12:12 hour light:dark cycle at room temperature (20–22 °C) and 40-60% humidity. This work was performed in compliance with the French application of the European Communities Council Directive of 22 September 2010 (2010/63/EEC) and approved by the Institut Pasteur ethical committee (CETEA #89, project #01126.02, #2013-0086 and #DAP200025).

### Stereotaxic injections

Adeno-associated viruses (AAV) were generated by the Penn Vector Core, University of North Carolina Vector core, Addgene vector core or produced by the Vector core of the Gene Therapy Laboratory of Nantes (INSERM UMR1089, https://umr1089.univ-nantes.fr/en/facilities-cores/cpv; Table 1). Herpes simplex viruses (HSV) were produced by the MIT gene transfer core (Table 1). CTB conjugated to Alexa Fluor 555 (C34776) was obtained from Molecular probes.

For viral injections, mice were deeply anesthetized using ketamine and xylazine mixture (150 mg/kg Imalgene and 5 mg/kg Rompun, respectively; i.p.) and placed in a stereotaxic apparatus (Bregma and Lambda aligned on the same horizontal plane). A small craniotomy was performed, and a viral (see Table 1 for details about viral constructs, titers and production) or CTB (A555-conjugated CTB; C34776, Molecular Probes) solution was injected into the brain through a glass micropipette attached to a Nanoinjector system (Nanoject II, Drummond). The coordinates and volumes used for injections were as follows: AON: 2.3 mm anterior and 1.1 mm lateral from Bregma, 3.3 and 3.6 mm deep from the brain surface, 100 nL/site; APC: 1.9 mm anterior and 2.25 mm lateral from Bregma, and 3.8 and 4.2 mm deep from the brain surface, 150-200 nL/site; NDB/MCPO: 0.1 mm anterior and 1.5 mm lateral from bregma, 5.5 deep from brain surface, 100 nL/site; Somatosensory cortex S1, barrel field: 1 mm anterior and 3 mm lateral from Bregma, 1.2 deep from brain surface, 200 nl/site; AONp: 1.9 mm anterior and 1.7 mm lateral, 4 mm deep from brain surface, 100 nl/site; APC/NAc : 1.2 anterior and 2.2 lateral to Bregma, 3 and 4.2 deep

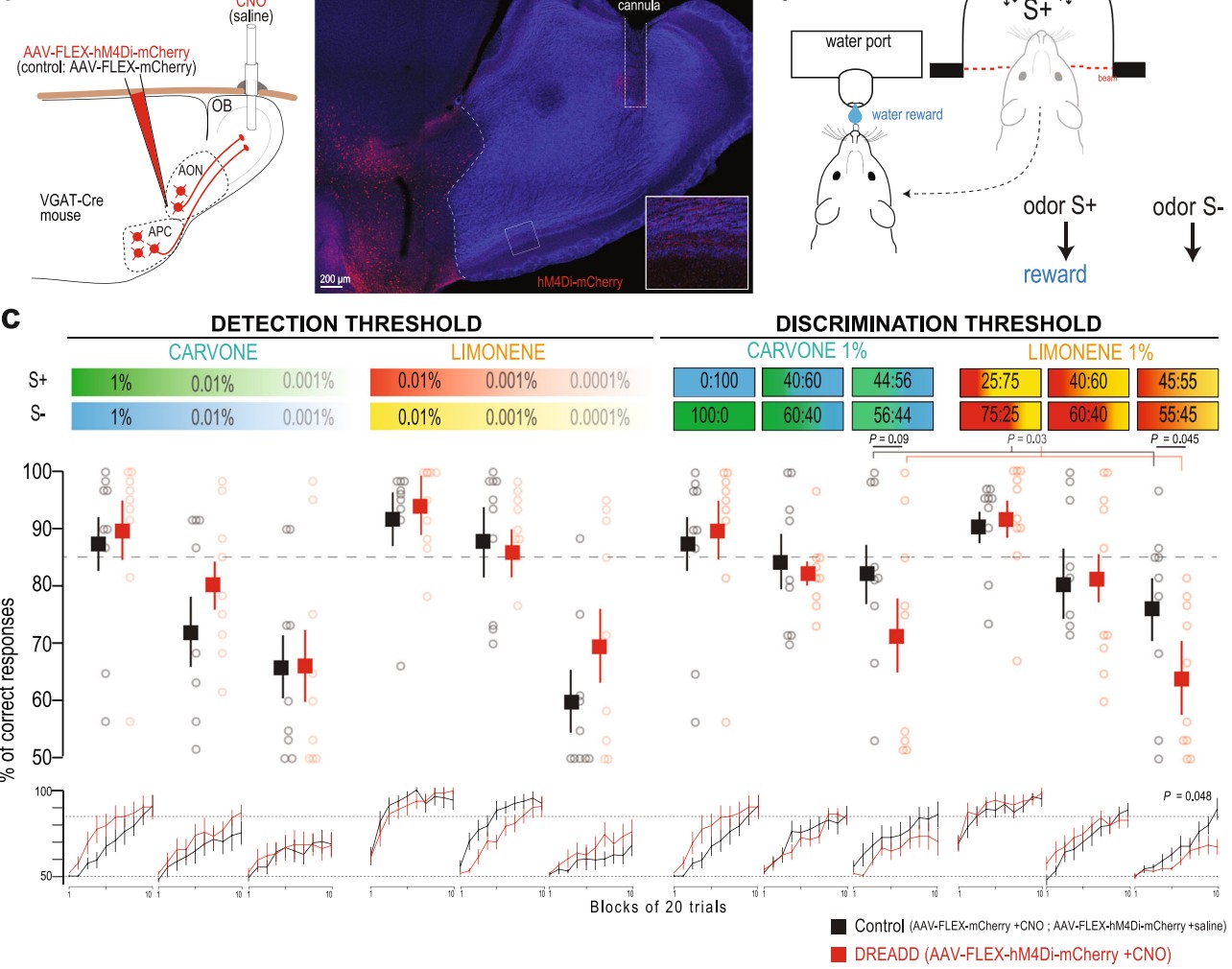

**Fig. 7 | Targeted pharmacogenetic inhibition of cortico-bulbar GABAergic axons impairs fine odor discrimination. a** Specific silencing of axonal outputs of AOC GABAergic neuron expressing hM4Di-mCherry by locally injecting CNO in the OB (0.1 mg/mL, 1 μL per hemisphere). Control animal expressed the protein mCherry and received bilateral CNO injection. Right, Coronal section showing hM4Di-mCherry⁺ GABAergic cells in the AON and their axonal projections in the OB, together with the track of the injection cannula targeted to the core of the OB. Inset: high magnification of the boxed region with increased red fluorescence gain. **b** Odor-reward association task. After a nose poke into the odor port, mice had to lick on the rewarded odor (S⁺) to obtain a water reward and refrained licking on the non-rewarded odor (S⁻). **c** Performance (percentage of correct responses) for the discrimination of the carvone (green/blue) and limonene (red/yellow) enantiomers.

Mean final performance (top; 3 last blocks, i.e., 60 last trials) and mean performance per block of 20 trials (bottom; 10 blocks per session, i.e., 200 trials). CNO reduced the performance for fine limonene mixture discrimination (enantiomers mixtures, 55:45 vs 45:55; top, two-sided Mann–Whitney test; bottom, repeated measure two-way ANOVA, $F_{(9,153)} = 1.955$, $P = 0.0483$; hM4Di, $n = 10$ mice; Control, $n = 9$ mice. A similar trend was observed on carvone enantiomers, although it did not reach significance ($n = 9$ mice in each group, two-sided Mann–Whitney test). The effects of CNO on fine discrimination performance was still significant when analyzing together carvone and limonene performances (55:45 & 56:44, last three blocks; two-way ANOVA, $F_{(1,33)} = 5.115$, $P = 0.0304$). Data presented as mean ± sem. Circle, individual mouse. Note that the data for carvone 1% and for carvone 100:0 are the same data. Source data are provided as a Source Data file.

from brain surface; OB: 1 mm anterior and 1 mm lateral from junction of inferior cerebral vein and superior sagittal sinus, 1, 1.5 and 2 mm deep from the brain surface, 100 nL/site. OB Lateral GL: 1 mm anterior and 2 mm lateral from junction of inferior cerebral vein and superior sagittal sinus, 1.5 mm from brain surface, 50 nL/site. Smaller volumes were used for sparse labeling and quantification of Synaptophysin-mRuby⁺ presynaptic boutons (Fig. 1c): 50nL/site in the AON and 50 nL/site in the APC. Injection coordinates in the AON/APC were optimized to prevent any direct diffusion of the AAV virus in the OB (Supplementary Fig. 1). Animals in which post-hoc histological examination showed that viral injections were not in the correct location were excluded from analysis. In the correctly injected animals, stereotaxic injection of AAV lead to virtually no somatas in the GCL (0.21 ± 0.11 neurons/mm², $n = 6$ mice, 8 sections per mouse), as reported previously in wild-type mice[44].

## Histology

**Tissue preparation.** Animals were intracardially perfused (4% paraformaldehyde (PFA) in 0.1 M phosphate buffer) and the brains were removed and post-fixed in the same fixative overnight. Following cryoprotection (30% sucrose in PBS for 48 h), brain sections were then cut with a freezing microtome (Leica). For post-hoc analyses of recording sites and viral expression, 60-μm-thick sections were sliced. OB sections were inspected to check for proper axonal expression, absence of virus diffusion into the OB, and for the absence of significant somatic labeling in the OB. 60-μm-thick sections were used for anatomical/histological analyzes. For immunodetection of synaptic protein, animals were intracardially perfused (freshly prepared 4% paraformaldehyde (PFA) in 0.1 M phosphate buffer), the brains were quickly removed, rinced in PBS and directly cryoprotected in (30% sucrose in PBS

**Table 1 | Details of the viruses used**

| Virus | Injection site | Titer | Source |
|---|---|---|---|
| AAV2/9-EF1a-DIO-hChR2(H134R)-eYFP | AON/APC (Figs. 1 and 3) APC/NAc (Supplementary Fig. 1) | $1.8 \times 10^{13}$ | Plasmid: Addgene#20298 Production: INSERM U1089 Vector Core |
| AAV2/5-hSyn-DIO-ChrimsonR-TdTomato | NDB/MCPO (Fig. 1) AON/APC (Supplementary Figs. 1 and 7, Figs. 2 and 4) S1 (Supplementary Fig. 4) | $4.5 \times 10^{12}$ | Addgene#62723 |
| AAVretro-EF1a-DIO-FLPo-WPRE-hGHpA | OB (Fig. 2, Supplementary Fig. 3) | $2 \times 10^{13}$ | Addgene#87306 |
| AAV5-EF1a-fDIO-hChR2(H134R)-eYFP | AONp (Fig. 2, Supplementary Fig. 3) | $3 \times 10^{12}$ | UNC Vector Core |
| AAV2/9-hSyn-ChrimsonR-TdTomato | AON/APC (Supplementary Fig. 7) | $5 \times 10^{12}$ | Penn Vector Core |
| HSV-hEF1-Flex-GCaMP6f (LT) | OB (Fig. 2, Supplementary Figs. 1, 2, and 3) | $1 \times 10^{8}$ | MIT gene transfer core |
| AAV2/9-CaMKIIa-hChR2(H134R)-eYFP | AON/APC (Fig. 1) APC/NAc (Supplementary Fig. 1) S1 (Supplementary Fig. 4) | $2 \times 10^{13}$ | Plasmid: Addgene#26969 Production: INSERM U1089 Vector Core |
| AAV2/1-hSyn-DIO-GCaMP6f | OB (Fig. 4, Supplementary Figs. 2 and 7) | $1.7 \times 10^{13}$ | Penn Vector Core |
| AAV1-synP-FLEX-splitTVA-EGFP-B19G | OB (Supplementary Fig. 5) | | Addgene #52473 |
| (EnvA)SAD-ΔG-mCherry | OB (Supplementary Fig. 5) | | Provided by Karl-Klaus Conzelmann |
| AAV2/9-DIO-GFP-IRES-Synaptophysin-mRuby | AON/APC (Fig. 1) | $5.7 \times 10^{12}$ | Plasmid: Addgene#71760 Production: INSERM U1089 Vector Core |
| AAV2/1-hSyn-GCaMP6s | OB (Figs. 4 and 6, Supplementary Fig. 8) | $2.1 \times 10^{13}$ | Penn Vector Core |
| AAV2/1-hSyn-GCaMP6f | OB (Supplementary Fig. 7) | $1.2 \times 10^{13}$ | Penn Vector Core |
| AAV2/5-CAG-DIO-ChIEF-TdTomato | AON/APC (Fig. 6, Supplementary Figs. 6 and 8) | $5.5 \times 10^{12}$ | Plasmid: Addgene#30541 Production: INSERM U1089 Vector Core |
| AAV2/5-hSyn-DIO-hM4Di-mCherry | AON/APC (Fig. 7, Supplementary Fig. 9) | $6.5 \times 10^{12}$ | UNC Vector Core |
| AAV2/5-hSyn-DIO-mCherry | AON/APC (Fig. 7, Supplementary Fig. 9) | $7 \times 10^{12}$ | UNC Vector Core |

for 24 h) without any post-fixation, before sectioning the OB on a freezing microtome (40 μm).

**Immunohistochemistry.** Primary and secondary antibodies used in this study are summarized in Tables 2 and 3, respectively. Immunochemistry labeling was performed as follows: slices were rinsed, permeabilized and blocked in 10% Normal Goat Serum and PBS containing 0.25% Triton-X100 (PBST) for 2 h. Primary antibodies were then incubated for up to 48 h at 4 °C in PBST containing 1% serum and 0.01% azide, washed three times and secondary antibodies were finally added for 2 h in PBST containing 2% serum. Slices were then rinsed and counterstained with DAPI, mounted and imaged with a confocal microscope (LSM 700, Zeiss Zen Black software) or epifluorescence microscope (Axiovert 200, Zeiss) equipped with an Apotome system (Zeiss, Zeiss Zen 2.6 Blue software). A secondary antibody control was included by omitting the primary antibodies in the staining protocol. For immunodetection of synaptic protein, the same IHC protocol was used except the fact that PBST only contained 0.1% of Triton-X100 and primary antibodies were incubated for 72 h. Confocal images (LSM 700, Zeiss) of the GCL were obtained with a 40X immersion objective (Zeiss) from the first 4 μm from the slice surface, given the limited penetration of VGAT antibodies in such a GABAergic region.

**Cell counting of retrogradely-labeled cells.** Coronal slices were serially collected and analyzed from the OB (+4 mm from Bregma) to the cortical amygdala (−0.5 mm from bregma). To evaluate cell density for each imaged slice, immunopositive somatas were manually counted for each subdivision and the surface of subdivision was measured (Zeiss Zen Black, Zeiss). Counting was blind to the genotype in SOM-Cre, PV-Cre and VIP-Cre mice. Values for each subregion are averaged across sections for each mouse and used to calculate the mean cell density (averaged number of cells per mm$^2$) and mean proportion of cells (% of cells counted in each subdivision relative to the total number of cells labeled in the respective brain). One out of every four slices were

used for GFP counting in Fig. 2c. One out of every six slices were used for GFP/SOM colocalization in Fig. 2g.

**Delineation of divisions and subdivisions of relevant brain regions.** Brain regions were manually delineated using morphological parameters, DAPI staining, immunohistochemistry labeling, and the Allen Mouse Brain Reference Atlas.

The *piriform cortex* is located in the ventrolateral forebrain, with a typical three-layered cortex and a layer 2 containing densely packed neurons. The piriform cortex was subdivided into anterior (APC) and posterior (PPC) regions, with the boundary at the caudal end of the lateral olfactory tract (LOT), as in ref. 24.

The *OT* is "readily identifiable as a large, pronounced, elliptical bulge nested between the LOT, the optic chiasm and the hemispheric midline ridge". It is a "trilaminar region which contains a peculiar gyrating structure with anatomically defined 'hills' (gyri and sulci) and 'islands'"[87]. The OT stains heavily for choline acetyltransferase and this staining was used in some slices (Supplementary Fig. 2b).

The *AON* is mainly located in the olfactory peduncle and consists of most of it. Yet, as detailed below, it extends caudally to the piriform cortex. The AON is a bilaminar region. Subdivisions of the AON were defined according to well-documented anatomical and cytoarchitectural landmarks[21,88,89]. The AON can be divided into two basic zone, the *pars externa* which is a "thin ring of cells that encircles the rostral end of the olfactory peduncle"[88] and the *pars principalis*. The latter is further subdivided in five regions, four of which are defined as a quadrant emerging from the anterior commissure: *pars dorsalis* (AONd), *pars ventralis* (AONv), *pars lateralis* (AONl) and *pars medialis* (AONm). A fifth region extends caudally to the piriform cortex (*pars posterioralis*, AONp). AONl: area that lies directly under the LOT. AONl has the highest density of cells in the *pars principalis*, forming almost a visible layer. AONm, anterior section: AONm is lying below the OB. Posterior: AONm is delimited by the dorsal and ventral Tenia tecta. The ventral part of AONm also exhibits a cell-free gap which marks the border with AONv. AONd: facing orbito-frontal cortex, with no contact with LOT. The AONd is delimited on the medial border by the dTT. AONv: diffuse layer 2, with lower density of cells and no visible layer-

**Table 2 | Details of the primary antibodies used**

| Primary antibodies | | | |
|---|---|---|---|
| Raised against | Host species | Dilution | Source |
| Calbindin D-28k | Mouse (monoclonal) | 1:2,000 | Swant 300 Clone Name: CB300 |
| Calretinin | Rabbit (polyclonal) | 1:2,000 | Swant 7697 |
| ChAT | Goat (polyclonal) | 1:200 | Millipore AB144P |
| GAD67 | Mouse (monoclonal) | 1:1,000 | Merck Millipore MAB5406 Clone name: 1G10.2 |
| GFP | Chicken (polyclonal) | 1:4,000 | Abcam ab13970 |
| Parvalbumin | Rabbit (polyclonal) | 1:2,000 | Swant PV27 |
| Somatostatin | Goat (polyclonal) Rabbit (polyclonal) | 1:500 1:4,000 | Santa Cruz D20 Immunostar #20067 |
| VGAT | Rabbit (polyclonal) | 1:1,000 | Synaptic Systems 131-002 |
| Vasoactive intestinal peptide | Rabbit (polyclonal) | 1:1,000 | Immunostar 20077 |
| RFP | Rabbit (polyclonal) | 1:4,000 | Rockland Inc. 600-401-379 |

**Table 3 | Details of the secondary antibody used**

| Secondary antibodies | | | |
|---|---|---|---|
| Raised against | Host species | Dilution | Source |
| Alexa 488 | | | |
| anti-Chicken | Goat | 1:1,000 | Molecular Probes A-11039 |
| anti-Rabbit | Goat | 1:1,000 | Molecular Probes A-11034 |
| anti-Rabbit | Donkey | 1:500 | Jackson 711-546-152 |
| anti-Goat | Donkey | 1:500 | Jackson 705-546-147 |
| Alexa 568 | | | |
| anti-Rabbit | Goat | 1:1,000 | Molecular Probes A-11036 |
| Alexa 647 or Cy5 | | | |
| anti-Mouse | Goat | 1:1,000 | Jackson 115-175-166 |
| anti-Rabbit | Goat | 1:1,000 | Jackson 111-175-144 |
| Alexa647-Streptavidin | | 1:1,000 | Molecular Probes, S21374 |
| Biotin-conjugated | | | |
| anti-Rabbit | Donkey | 1:1,000 | Jackson 711-065-152 |
| anti-Goat | Donkey | 1:200 | Santa Cruz SC-2042 |

like compared to AONl, which marks the border. AONv has limited contact with the LOT. The OT appears over the AONv in posterior sections.

AONp: caudal to AONv, buried between OT and APC, outside the olfactory peduncle per se. In contact with the anterior commissure, but with limited contact with LOT. The AONp starts when the dorsal and ventral TT fused and when the OT emerges clearly with 3 layers visible, isolating the AONp from the LOT. AONp has a group of large, loosely aggregated neurons (further refs. [61,90,91]).

The *NDB/MCPO* is a more caudal structure, located in the basal forebrain that runs rostro-caudally from the septum to the anterior amygdala area[25,38].

**Density of fluorescent axons.** For the density profiles, OB coronal slices were imaged, and the immunoreactivity profile was determined using ImageJ 1.53t. Measurements were performed in matching slices and averaged across 4 sections per animal, normalized to the maximum intensity and then averaging between animals. For axonal density in Fig. 2h, each section was imaged with the same parameter, and a binary image was calculated on ImageJ to then calculate the proportion of immunopositive pixels in each region. This value was then normalized to the density of immunopositive pixels in the injected region (AONp).

**Quantification of synaptic punctas.** The colocalization pattern of VGAT and mRuby immunoreactive clusters was determined in the GCL from 8–10 confocal sections per animal ($n = 3$ mice) using Imaris software (Bitplane, Zurich, Switzerland). For each section, each single-labeled VGAT$^+$ or mRuby$^+$ punctas was identified by Imaris surface segmentation algorithm (intensity threshold: 15–25% of the highest intensity value, surface threshold: 0.1 μm$^2$) and the number and size of VGAT$^+$ or mRuby$^+$ punctas was independently quantified. The colocalization algorithm was then used to identify the proportions of VGAT/mRuby colocalized punctas as well as the proportion of surface shared by the two markers. We considered a mRuby$^+$ puncta co-labeled with VGAT$^+$ punctas using a conservative estimate corresponding to a surface threshold of 0.1 μm$^2$, i.e., 50% of the average mRuby surface.

### Slice electrophysiology

**Slicing procedure.** Three weeks post injections, mice were deeply anesthetized with intraperitoneal injection of ketamine (100 mg/kg) and xylazine (10 mg/kg) and swiftly decapitated. The OB and frontal cortices were rapidly dissected and placed in ice-cold artificial cerebrospinal fluid (ACSF) containing 124 mM NaCl, 3 mM KCl, 1.3 mM MgSO4, 26 mM NaHCO3, 1.25 mM NaHPO4, 20 mM glucose, 2 mM CaCl [~310 mOsm, pH 7.3 when bubbled with a mixture of 95% O2 and 5% (vol/vol) CO2; all chemicals from Sigma-Aldrich]. Horizontal slices (300-μm thick) of the OB were placed in bubbling ACSF in a warming bath at 35 °C for 30 min and then at room temperature (i.e., $22 \pm 1$ °C). For whole-cell recordings, individual slices were placed in a chamber mounted on a Zeiss Axioskop upright microscope, and continuously perfused (1.5 mL/min) with 30 °C ACSF (Warner Instrument inline heater). Slices were visualized using a 40× water immersion objective. Recordings were performed 3 weeks post-injection to avoid any possible contamination from adult-born GCs[49].

**Identification of neuronal subtypes.** We obtained whole-cell patch-clamp recordings from visually targeted GCs, MCs, dSACs, PG cells, sSACs, eTCs and TCs. Neurons were filled with fluorescent dye (Alexa 488, 40 μM) and classified based on their somata laminar location, morphological, electrophysiological criteria[16,92,93]. Some patched cells were also filled with biocytin. For post-hoc revelation of biocytin-filled cells, slices were immediately fixed in PFA 4% for 24 h, then rinced in PBS and incubated with PBS containing Triton (0.5%), DAPI (1:10000, Molecular Probes) and Alexa568-conjugated streptavidin (Molecular probes) for 2 h. Slices were finally rinced, mounted (Fluoromount) and imaged with epifluorescence microscope (Axiovert 200, Zeiss) equipped with an Apotome system (Zeiss) and the Zeiss Zen 2.6 Blue software.

*eTCs*: large (~20 μm diameter) somata in the inner part of the GL, a single dendrite and tuft ramifying within one glomerulus, an axon extending into the EPL and a relatively low input resistance (~200 MΩ).

*PG cells*: smaller somata (~8–10 μm diameter) residing in the GL, with high input resistance (~500–1000 MΩ). Highly ramified dendrite arbor in only one glomerulus.

*sSACs*: larger soma in the GL (>10 μm diameter), low input resistance (~200–300 MΩ), unique dendritic arbors that are exclusively periglomerular, span multiple glomeruli, lack tufts, and are poorly branched.

*TCs*: large soma (10–20 μm diameter) in the inner part of the EPL (20–150 μm above the MCL). Large apical dendrite innervating a single glomerulus, lateral dendrites in the EPL. Very low input resistance (~50–100 MΩ).

*MCs*: very large soma in the MCL (20–30 μm diameter). Large apical dendrite innervating a single glomerulus. Lateral dendrites in the EPL. Very low input resistance (~50–100 MΩ).

*dSACs*: soma size of >10 μm diameter in the IPL or immediate surrounding GCL, with unique multipolar dendritic morphology (compared to GCs or MCs) and multiple neurites in the IPL. Low input resistance (~200–300 MΩ).

*GCs*: small soma (8–10 μm diameter) in the GCL with one apical dendrite arborizing in the EPL and small basal dendrites, high input resistance (~500–1000 MΩ). Patched GCs were preferentially located in the superficial GCL (100–150 μm below the IPL).

Averaged input resistances of patched cells are presented in Supplementary Table 1.

**Recordings.** Patch pipettes, pulled from borosilicate glass (OD: 1.5 mm, ID: 0.86 mm; Sutter instrument; P-87 Flaming/Brown micropipette puller, Sutter Instruments), had resistances of 6–10 MΩ for GCs and PG cells recordings and of 3–5 MΩ and were filled with a cesium gluconate-based solution: 126 mM Cs-gluconate, 6 mM CsCl, 2 mM NaCl, 10 mM Na-Hepes, 10 mM D-glucose, 0.2 mM Cs-EGTA, 0.3 mM GTP, 4 mM Mg-ATP, 280–290 mOsm, pH 7.3). Membrane potentials indicated in the text are corrected for a measured liquid junction potential of +10 mV. Recordings were obtained via an Axon Multiclamp 700B. Synaptic events were elicited by photo-activation of ChR2$^+$ axon terminals stimulation using a 470-nm LED (Xcite by Lumen Dynamics) illuminating the sample through the objective. IPSCs were recorded at Vc = 0 mV, unless otherwise stated. Rise times were measured between 10% and 90% of peak amplitude (Supplementary Table 1). Decay time constants were derived by fitting the sum of two exponentials and the first fast tau component was extracted (Supplementary Table 1). Data were acquired using Elphy software (Gerard Sadoc, Centre National de la Recherche Scientifique; Gif-sur-Yvette, France) and analyzed with Elphy and IgorPro (Neuromatic by Jason Rothman, www.neuromatic.thinkrandom.com). In acute slices, none of the recorded GC exhibited ChR2-mediated inward currents (0/64) and IPSC kinetics in MCs and TCs were not consistent with GC-mediated inhibition[49].

### In vivo electrophysiology

Following stereotaxic viral injection, a L-shaped metal bar and a silver reference electrode were fixed to the caudal part of the skull. Mice were allowed one week to recover and were subsequently slowly and progressively trained for head restraint habituation, a 5% sucrose solution was given as a reward. The craniotomy was performed the day before recording and protected with silicone sealant (KwikCast). An array of 4 tungsten electrodes (~3 MΩ; FHC) was glued together and was slowly lowered into the OB. An optic fiber (multimode, 430 μm diameter, numerical aperture (NA) 0.39, Thorlabs) was positioned on the surface of the OB. A drop of silicone sealant was applied to the brain surface to increase recording stability and avoid tissue desiccation. LFP signals were recorded in the MCL/GCL. Signals pre-amplified (HS-18; Neuralynx), amplified (1000×; Lynx8,

Neuralynx) and digitized at 20 kHz (Power 1401 A/D interface; CED). Light stimulation of cortical GABAergic axons was performed using an optic fiber coupled to a DPSS laser (473 nm, 150 mW; CNI Lasers; output fiber intensity, 20 mW) via a custom-built fiber launcher and controlled by a PS-H-LED laser driver connected to the CED interface. Light stimulation consisted in patterned light stimulation (10, 33 or 66 Hz) with 5-ms-long light pulses.

### Calcium imaging using fiber photometry

We used a fiber photometry system adapted from ref. 94. Immediately following GCaMP6f virus injection in the OB, AON or APC, optical fibers (multimode, 430 μm in diameter, NA 0.5, LC zirconia ferrule) were bilaterally implanted close to the virus injection site, in the ventral part of the OB for GCL recording (1 mm anterior and 1 mm lateral from junction of inferior cerebral vein and superior sagittal sinus, 2 mm deep from the brain surface) and in the lateral part for MC/TC recordings (1 mm anterior and 1.5 mm lateral from junction of inferior cerebral vein and superior sagittal sinus, 1.5 mm deep from the brain surface) and then secured to the skull with a liquid bonding resin (Superbond, Sun Medical) and dental acrylic (Unifast). Three weeks post-injection, GCaMP6f was continuously excited using a 473 nm DPSS laser (output fiber intensity <0.1 mW; Crystal Lasers) reflected on a dichroic mirror (452–490 nm/505–800 nm) and collimated into a 400 μm multimode optical fiber (NA, 0.48) with a convergent lens (*f* = 30 mm). The emitted fluorescence was collected in the same fiber and transmitted by the dichroic mirror, filtered (525 ± 19 nm) and focused on a NewFocus 2151-femtowatt photoreceptor (Newport; DC mode). Reflected blue light along the light path was also measured with another amplified photodetector (PDA36A, Thorlabs) for monitoring light excitation and fiber coupling. Red light (589 nm, 10 mW, pulse duration: 10–15 ms) was collimated in the recording optic fiber to selectively activate cortical ChRimsonR-expressing GABAergic axon terminals in the OB while GCaMP6f was independently excited with low blue light intensity (<0.1 mW), thereby avoiding cross-excitation of ChRimsonR[27]. Sessions with significant averaged changes in the reflected blue light (>1% ΔF/F) were discarded from the analysis. Signals from both photodetectors were digitized by a digital-to-analog converter (DAC; Power 1401, CED) at 5 kHz and recorded using Spike2 software.

Mice were placed in a small, ventilated cage (~0.5 L). Using a custom-built air-dilution olfactometer controlled by the CED card, pure monomolecular odorants were diluted in mineral oil and saturated odorized air was further mixed with the air stream (1/10 dilution) before being delivered into the ventilated cage (flow rate of 4 L/min), thanks to solenoid pinch valves. Odors were presented for 5 s every 60 s and dynamics of odor introduction and exhaust in the cage were constantly monitored using a mini photoionizer detector (miniPID, Aurora) positioned at the ceiling of the cage. Odors used were: Acetophenone 1%, Anisol 1%, Carvone+ 5%, Decanal 5%, Ethylbutyrate 0.5%, Geraniol 5%, Heptanal 1% Hexanone 0.5%, 2-methylbutyraldehyde 1%, Pentanol 1%, Valeraldehyde 0.2%, Methyl Salicylate 2%, 3-methyl-3-penten-2-one 1%. The 589 nm light stimulation was applied during 1 s, 3.5 s after odor onset when odor and light were simultaneously presented as well as 30 s after odor presentation. For a given stimulation frequency, cycles of odor, light, and odor +light presentations were repeated 10 times for each condition to obtain interleaving trials with and without light (recording session duration ~240 min). Signals were smoothened (0.02 s window) and downsampled to 500 Hz. For each trial, the signal was normalized to the baseline fluorescence of the trial using the ΔF/F ratio with $F_O$ being the average fluorescence 2 s before the beginning of the trial. After completion of the recordings, mice were deeply anesthetized and transcardially perfused with 4% paraformaldehyde. OB and AOC were cut into 60μm-thick slices and observed with light and epi-fluorescence microscopes to evaluate the correct position of the

optical fibers and the correct expression and diffusion of the virus. Animals in which post-hoc histological examination showed that viral injection or implanted optic fiber were not in the correct location were excluded from analysis. Selected sections were counterstained with DAPI and mounted for image acquisition (Axiovert 200 with Apotome system, Zeiss Zen 2.6 Blue, Zeiss).

### Calcium imaging using two-photon microscopy

**Acquisition parameters and imaging.** After viral injections, a cranial window (3.0 × 1.4 mm glass) was placed over both OB and a stainless-steel head bar (L-shaped) was cemented to the skull. Mice were then allowed to recover for a month. During this period, the animals were progressively habituated to the head fixed position while staying quiet in the 50-ml open-ended support tube. Calcium activity was imaged using a two-photon system (950 nm, Spectra Physics) with a Prairie Investigator microscope (Prairie View Software 5.4.64.100, Bruker) and equipped with GaAsP photomultiplier tubes (PMTs). $Ca^{2+}$ transients were imaged using a 16X, 1.05 NA microscope objective (Nikon) with a 2X digital zoom. The field of view was 512 ×512 pixels (423.7 ×423.7 µm), imaged at 15 Hz using a resonant galvanometer. Imaging planes (MCL, EPL or GL) were determined using anatomical landmarks and layers depth profiles as in refs [53,56]. Mean recording depth ± s.d (relative to the GL): MCL, 201.8 ± 29.9 µm; EPL, 60.1 ± 20.7 µm. MCs were further distinguished from TCs by their denser packing and larger soma size[32], less dense neuropil[55].

**Stimulation protocols.** Trials consisted in 8 s baseline, 2 s stimulation (odor, light, or odor+light) and 10 s inter-trial interval. Trials were grouped in blocks of 20 trials. 2–3 blocks were acquired per stimulus type. Data was acquired from 6 OBs of 4 animals.

**Light activation of GABAergic cortical axons.** The LED illumination for full-field photo-activation feature of the Investigator series (Bruker) was used to photo-stimulate the GABAergic axons in the OB. Blue light was directed to the field of view through the microscope objective. The PMT shutter remained closed during the photo-stimulation period and GCaMP6s fluorescence light was collected before and 50 ms after the stimulation for allowing bidirectional realignment of the scanning. This time-window was evaluated using control trials with the light shutter closed but in the absence of photo-stimulation. GCaMP6s photo-bleaching using our ChIEF$^+$ axon photo-activation paradigm was assessed by applying the same protocol to mice expressing GCaMP6s solely and was minimal.

**Odorant delivery.** The odor pairs were a natural odor pair (curry powder vs. cinnamon) or a pair of pure monomolecular odorants (ethyl butyrate, valeraldehyde, isoamyl acetate, ethyl tiglate, hexanone or cineole, Sigma-Aldrich). Pure odorants were diluted 1:10 in 10 mL mineral oil and natural odorants were presented in their native state. Saturated odor vapor was further diluted with humidified clean air (1:10) by means of computer-controlled solenoid pinch valves. Odor presentation was performed using a custom-built computer interface. Odor delivery dynamics were monitored and calibrated using a mini-PID (Aurora). Odors were delivered randomly within a block (10 trials of each odorant).

**Odor and light stimulation.** In "Odor + Light" trials, odorants and light were presented simultaneously utilizing the protocols mentioned above. After completion of the recordings, mice were transcardially perfused with 4% paraformaldehyde. OB and AOC were cut into 60µm-thick slices and observed with light and epifluorescence microscopes to evaluate the correct expression and diffusion of the virus. Selected sections were counterstained with DAPI and mounted for image acquisition (Axiovert 200 with Apotome system, Zeiss Zen 2.6 Blue, Zeiss).

### Image analysis

**Motion correction.** A full field of view motion correction was performed using a custom-made program in MATLAB 2018a. A two-dimensional cross-correlation of every frame with the average projection of the entire image set was used to identify the out of frame z-movements (Pearson's r > 0.65 in the 2D cross-correlation). For lateral motion correction, the established ImageJ plugin *MoCo*[95] was used. In brief, it uses a Fourier-transform to improve the efficacy for identifying translational motion.

**Principal component analysis (PCA) assisted reconstruction.** We employed PCA on the raw motion-corrected datasets, after concatenating all the trials for each experiment into a 3-dimensional matrix. It leads to the possibility to express the original data in a lower dimension, capturing the largest variability in the dataset. The original image set is reconstructed from the most variable eigenvectors (non-varying (i.e., inactive) pixels and shot noise in the dataset do not have high variability across time and can be excluded). The reconstruction was done by a linear combination of the PC scores and the PC coefficients for the first 10 PCs of the dataset (Supplementary Fig. 10a, b).

**Identification of regions-of-interest (ROIs).** The PCA-reconstructed images were used for the identification of ROIs. ROIs were manually drawn on the cell bodies using ImageJ 1.53t and were imported in MATLAB 2018a (Supplementary Fig. 10c). As explained above, in order to remove the contribution of neuropil and background fluctuation, we performed a second PCA inside each ROI (Supplementary Fig. 10d–f). We used PC1-3 to redefine the outer bounds of each ROI for activity quantification. Note that PCA reconstructions did not modify the fluorescence data inside the ROIs. This process eliminated noisy signals (Supplementary Fig. 10g–m) and improved signal-to-noise ratio by 25% in a similar two-photon calcium imaging dataset[96].

### Data analysis

**Z-score calculation.** For each ROI, the pixel intensities were smoothed across 5 frames and *z-score* was calculated for each cell as follow:

$$z = \frac{\mu resp - \mu baseline}{\sqrt{\sigma resp^2/n - \sigma baseline^2/n}} \tag{1}$$

With µ and σ being the mean and standard deviation; resp, response (1 s after shutter reopening) and baseline is 1 s before shutter closes. n is the number of trials. For comparison, we show ΔF/F values in Supplementary Fig. 8, with $F_0$ being the baseline determined for each trial.

Individual cell response to ChIEF light stimulation was considered significant if it passed a two-tailed paired t-test based on single trials, with an alpha threshold of 0.01. Response and baseline values were the mean values 1 s after and 1 s before the shutter closed and reopened, respectively.

Odor responsive cells were identified using a two-sided paired t-test (α = 0.05) on all odor trials, regardless of whether GABAergic axons were stimulated or not to avoid a selection bias.

**Euclidean distance.** For each recording session (7 for MCs, 9 for TCs), pairs of population vectors were constructed from the averaged z-score responses to either the two odors presented on that day (Between odors design), or to the same odor (Within odor design). Only the cells responding to both odors (Between odors) or to the given odor (Within odor) were selected. Sessions were kept if a minimum number of 5 cells were responding to an odor. Pairwise Euclidean distance was calculated on the population vectors in the remaining sessions.

The Euclidean distance between two vectors $p$ and $q$ is given by the formula:

$$d(p,q) = \sqrt{(p-q)(p-q)\prime}  \qquad (2)$$

## Population network model

The firing rate of the excitatory (MCs or TCs, MC/TCs) and inhibitory subnetworks (GCs) were described by the following differential equations (MATLAB 2018a):

$$d\overline{MC} = -MC + I - w_{mc \to gc}*GC - w_{fb \to mc}*FB \qquad (3)$$

$$d\overline{GC} = -GC + w_{mc \to gc}*MC - w_{sac \to gc}*SAC - 2*w_{fb \to mc}*FB \qquad (4)$$

where $I$ is the odor input, $w_{mc \to gc}$ is the strength of the connection between MCs (or TCs) and GCs and $-w_{mc \to gc}$ is the reciprocal connection from GCs to MCs (or TCs). $w_{sac \to gc}$ is the strength of the inhibitory connection from dSACs to GCs. $w_{fb \to mc}$ is the strength of the cortical GABAergic feedback (FB) connection on MCs (or TCs). According to our slice electrophysiology results, the strength of the GABAergic feedback is roughly twice stronger on GCs, therefore it was set as $2*w_{fb \to mc}$. The strength of the feedback on dSACs is even stronger and because, for the sake of simplicity, dSACs are not modeled, we modeled the inhibitory input strength to GCs in presence of feedback decreasing with the inverse of the strength of the feedback to MCs (or TCs).

The steady state (i.e., nullclines) of the MCs/TCs and GCs subnetworks are as following:

$$MC_{nullcline} = \frac{1}{w_{mc \to gc}}*(I - MC - w_{fb \to mc}*FB) \qquad (5)$$

$$GC_{nullcline} = -w_{mc \to gc}*MC - w_{sac \to gc}*SAC - 2*w_{fb \to mc}*FB \qquad (6)$$

Where $FB = \{1, \text{if feedback } 0, \text{otherwise and}$

$SAC = \{\dfrac{0.1}{w_{fb \to mc}}, \text{if feedback } 1, \text{otherwise}$

We explored different strengths and relative strengths between the MC-GC connections and the feedback.

The other parameters, such as the strength of the odor input and that of dSACs were not changed but they affect only the basal firing rate of MCs/TCs and GCs, not the slope of the nullclines nor the fixed points. Therefore, results will be qualitatively the same with different odor and dSACs input strength.

The fixed point (i.e., the equilibrium of the system) was used to determine the population rates of the MCs/TCs and GCs.

## Behavior

Two-guide cannulas (26-gauge, 7 mm long) were bilaterally implanted over the dorsal surface of the OB on the same day as viral injections 1 mm anterior and 1 mm lateral from junction of inferior cerebral vein and superior sagittal sinus. Guide cannulas were stabilized with a liquid bonding resin (Superbond, Sun Medical) and dental acrylic (Unifast) and a dummy cannula was positioned in the guide cannula to prevent blocking. Mice were habituated to be handled and maintained still while manipulating the dummy cannulas. On the day of the experiment dummies were retrieved, cannulas (8.5 mm long, to inject at 1.5 mm below the surface of the brain, 33-gauge and connected to a 10 μL Hamilton syringe) were placed for injections into the GCL. Dummies were put back in place a few minutes after the end of the injection.

Behavior experiments were conducted using a go/no-go operant conditioning scheme as previously described. 2 weeks after the surgery, aged-matched adult male VGAT-Cre mice (10-12 weeks old) were partially water-deprived (maintained at 80-85% of their baseline body weight) and trained in custom-built computer-controlled eight-channel air-dilution olfactometers[71,97]. Solenoid pinch valves controlled purified air streams, passing over the surface of mineral oil-diluted odorants. The odorized air was diluted 1:40 in odor-free air before its introduction into an odor sampling tube in the mouse operant chamber. Standard operant conditioning methods were used to train mice to insert their snouts into the odor sampling port for at least 1 s and to respond by licking the water delivery tube located 5 cm left of the odor port to get a water reward (3 μL). An infrared detection system continuously monitored the presence of the animal in the odor port. After this training phase to learn the procedure (200 trials per day for 5 days), mice had to learn to lick in the presence of a positive odor stimulus S+ and to refrain from licking and retract their head from the sampling port in the presence of a negative odor stimulus S−. In each trial, a single stimulus was presented and S+ and S− trials were presented in a modified pseudo-random order. Inter-trial intervals were minimum 8s-long. Each mouse performed a maximum of 10 blocks (200 trials) per day. The percentage of correct responses was determined for each block of 20 trials. A score of 85% at the very least implied that mice had correctly learned to assign reward/non-reward values. Odor sampling time was the time between the opening of the final valve and head retraction out of the odor sampling port.

Initial odor-reward learning, without intrabulbar injection, was performed using Anisole (S+) and Heptanone (S−). All the mice learned the behavioral procedure and were able to discriminate the two odors (behavioral performance >85%) within three days. Three additional days of training were performed to ensure performance stabilization. Then mice were first trained with limonene enantiomers [S+, (+)-limonene; S−, (−)-limonene] and then carvone enantiomers [S+, carvone-(+); S−, carvone-(−)]. Detection threshold was assessed by diluting each day by a factor of 10 the two enantiomers to detect (from 1% to 0.0001% dilution). Discrimination threshold was assessed by utilizing binary mixtures of the enantiomers, with a progressive and symmetric increase of the proportion of one into the other each day (from pure enantiomers discrimination, i.e., 100:0 vs. 0:100, to discrimination of mixtures with 55:45 vs. 45:55 enantiomer ratios). To induce pharmacogenetic silencing before each different olfactory task, mice underwent bilateral intrabulbar injection of CNO or vehicle (saline) through the guide cannula (CNO final concentration: 0.1 mg/mL, injection speed: 0.33 μL/min for 3 min, 1 μL total/bulb) and were left in their home cage for 15–20 min to allow CNO or vehicle (saline) diffusion within the OB, before being placed in the olfactometer. The control group was composed of mice expressing mCherry in cortical GABAergic axons without the h4MDi receptor injected with CNO (controlling for CNO side-effects, $n = 6$) and mice expressing h4MDi in cortical GABAergic axons injected with saline (controlling for any non CNO-dependent side effect of expressing the exogenous h4MDi receptor, $n = 3$). For CNO injections, experimenters were blind relative to the viral constructs expressed in individual mice. For behavior, animals which did not perform the 200 trials in the 60 min time window following CNO injection were discarded from the analysis. After completion of the behavioral experiments, mice were transcardially perfused with 4% paraformaldehyde. OB and AOC were cut into 60-μm-thick slices and observed with light and epifluorescence microscopes to evaluate the correct position of the injection cannula and the correct expression/diffusion of the virus. Animals in which post-hoc histological examination showed that transgene expression were not restricted to the AON were excluded from analysis. Selected sections were counterstained with DAPI and mounted for image acquisition (Axiovert 200 with Apotome system, Zeiss).

## Statistical analysis

Sample sizes are indicated in the figure and/or in the legend of the corresponding figures. All statistics were performed using GraphPad Prism 8 or MATLAB 2018a.

## Reporting summary

Further information on research design is available in the Nature Portfolio Reporting Summary linked to this article.

## Data availability

Processed imaging data is available online at: https://zenodo.org/record/7050088#.YxXaSHbMJD8 (https://doi.org/10.5281/zenodo.7050088). Data generated in this study are available in the Source data file. Correspondence and requests for all other materials and information should be addressed to C.M. and G.L. Source data are provided with this paper.

## Code availability

Scripts used for analyses are available on GitHub: https://github.com/camille-lab/GABAergicOlfactoryFeedback.

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

## Acknowledgements

We thank Dr. William Podlaski for help with the model. We thank Sara Moberg for comments on the manuscript. We also thank Carine Moigneu for her help with some viral injections, Julien Grimaud and Lucie Dixsaut for early works. We also wish to thank Uwe Maskos, Christoph Schmidt-Hieber and David DiGregorio from the Institut Pasteur for the gift of SST-Cre, PV-Cre and VIP-Cre mice. We also thank the Genetically-Encoded Neuronal Indicator and Effector (GENIE) Project and the Janelia Farm Research Campus of the Howard Hughes Medical Institute for sharing GCaMP6f constructs. We are greatful to Alexandru A. Hennrich and Karl-Klaus Conzelmann (Max Von Pettenkofer Institute Virology and Gene Center, Medical Faculty, Ludwig-Maximilians-University Munich, Germany) for the generous gift of (EnvA)SAD-ΔG-mCherry virus. We also thank the Vector Core of the Laboratory for Translational Research in Gene Therapy (INSERM UMR 1089, Université de Nantes, France) for AAV vector production and the Viral Core Facility of the McGovern Institute (MIT) for HSV production. This work was supported by the life insurance company "AG2R-La-Mondiale", the Agence Nationale de la Recherche (ANR-15-CE37-0004 "SmellBrain", ANR-16-CE37-0010 "ORUPS", ANR-15-NEUC-0004 "Circuit-OPL") and the Laboratoire d'Excellence Revive (Investissement d'Avenir, ANR-10-LABX-73). Our laboratory is part of the Ecole des Neurosciences de Paris (ENP) Ile-de-France network and is affiliated with the *Bio-Psy* Laboratory of Excellence. C.M. is a recipient of a fellowship from the French Ministère de l'Education Supérieure et de la Recherche and was also supported by the Fondation de la Recherche Médicale (FDT20160435483).

## Author contributions

C.M., G.L., A.N., S.S. and P.-M.L. designed the experiments, C.M., G.L., A.N., S.S., and E.P. performed and analyzed the experiments. C.M., G.L. and P.-M.L. wrote the manuscript.

## Competing interests

The authors declare no competing interests.
