## [Peer Review File · Nature Communications]

Long-range GABAergic projections contribute to cortical feedback control of sensory processing.Reviewer #1:

Comments for the Author:

In this manuscript Lepousez, Lledo, Mazo and colleagues describe long-range GABAergic projections from (parts of) olfactory cortex to the olfactory bulb and analyze their functional role / consequences. Demonstrating the existence of significant long-range GABAergic feedback from olfactory cortex is indeed very interesting and potentially very relevant for a wider community of sensory / circuit neuroscientists. After the initial (viral and CTB) tracing-based description of GABAergic projections from anterior olfactory nucleus and anterior piriform cortex (collectively referred to as AOC), the authors describe the different chemotypes underlying projections. Subsequently they use ChR2 stimulation and pharmacology in brain slices to define targets of feedback. They follow this up with different investigations into the functional impact of GABAergic feedback (using different exciting opsins for physiology and suppressing chemogenetic tools for behavior). Using photometry they find inhibition of GCL neurons, using 2P imaging they find a (small) inhibition of MC and TC activity and an increase in separation of stimulus representation for TCs but not for MCs. Using field potential recordings they find selectively an increase in beta oscillations (if feedback is optogenetically driven at beta frequencies) but no change at other frequency combinations. Finally they show that while odor detection thresholds are largely unaffected by suppressing GABAergic feedback chemogenetically, discrimination thresholds for some discrimination tasks of highly similar enantiomer mixtures are somewhat reduced. Overall, the authors use an impressive diversity of techniques, many difficult and expertly employed.

I have, however, main concerns about the paper:

Firstly, the diversity of circuit analyses makes it difficult to build a coherent picture of what GABAergic feedback is actually doing. Seemingly, cortical GABAergic fibers impinge on all cell types in the OB making the interaction of feedback and odor stimulation on representation very difficult to comprehend. It seems as there is some effect on representation in TCs but not MCs - but how this effect comes about, why it is differential, and how it might or might not relate to direct inhibition or indirect effects through interneurons remains unclear. Even the effect on GCL interneurons on its own is difficult to form a coherent picture of, considering that dSACs – who in turn provide a strong inhibitory drive onto GCs – are seemingly a prominent target for GABAergic feedback (fastest and strongest according to Fig 3). Moreover, the stimulation experiments do not immediately inform us about what the normal role for these feedback projections might be – when are they normally activated, how are they engaged etc. The behavior employed could give some clue; however the effect observed is not particularly striking. Firstly, it is quite subtle. Moreover, a lot of other modifications (e.g. in many expert papers by the senior authors themselves as well as several other groups) have demonstrated that disturbing the olfactory circuitry subtly can result in impairment of odor fine discrimination, so the finding of such perturbation seems quite generic. Attributing a specific role in e.g. expectation, priming, enhancing OB processing for some stimuli at the expense of others etc remains unclear. I very much realize that answering all these mechanistic questions is way beyond a study that describes these projections for the very first time and should derive its impact from that initial description. As is, however, the large number of - superficially seemingly conflicting or at least not building upon each other - physiology and behavior results rather confuses than provides a coherent picture as to their role or impact on olfactory function. I was left wondering what I actually learned from all the physiology and behavior. It probably requires time and significantly more extensive and detailed investigations to build clear hypotheses about the role of these feedback projections and explicitly test these hypotheses. It might also require more consistent and systematic manipulations and investigations (same optogenetic and/or chemogenetic tools, bidirectionally modulating consistent populations of projections etc).

Having said that, the finding that there are GABAergic feedback projections from AOC in itself – if demonstrated beyond any doubt – is a very important one. Thus, providing a detailed investigation into the anatomy, delineating the circuitry and demonstrating that these connections are functional (as e.g. with CRACM in Fig 3) would be an important contribution without much of the in vivo analysis. Here, substantially more detail are needed, in particular to exclude beyond any doubt that any of the effects observed in the OB could be due to viral diffusion or off-target expression (eg. leaky expression of the flexed constructs). Moreover, more clarity about the different chemotypes of the projection neurons (which projections are e.g. due to SOM positive neurons compared to other chemotypes, which projections originate from the different parts of AOC, approximately how many neurons contribute to the extensive axonal network in the OB etc) would help to build clearer hypotheses about their functional role.

Specific points:

1) To ascertain that the fibers stimulated in the OB (throughout the paper) indeed originate from AOC, the authors need to perform more thorough controls.

Firstly, to control for leaky expression, the same flexed virus could be injected into AOC of a WT animal at high titer (maybe together with a virus expressing the same fluorophore unconditionally to control for injection site, spread and provide similar images of axonal arborization in the OB) – and the same CRACM experiments as in Fig 3 performed with the experimenter blind to the genotype. While the specific virus used by the authors is not known to have leaky expression for such a high profile claim this would be an important and strong control.

Our reply: We thank the reviewer for pointing out this control. We performed several experiments to validate the absence of leaky expression in our CRE-dependent tools.

First, to control for unconditional labeling using a Cre-dependent virus, we co-injected in 2 VGAT-Cre and 2 WT (C57Bl/6RJ) a mixture of Cre-dependent virus (AAV-FLEX-ChR2-eYFP, no dilution; titer $\sim 1.10E13$ VG/ml) and Cre-independent virus (AAV-CaMKII-ChR2-mCherry, no dilution; titer $\sim 1.10E13$ VG/ml). The injection was targeted in the APC as well as in the Nucleus Accumbens (NAc, above APC) in order to target a cortical as well as a GABAergic structure. NAc is composed of 95% GABAergic cells, whereas APC only contains 10-20% of GABAergic cells. Three weeks post-injection, we did not observe any conditional labeling (eYFP) in WT animals, whereas strong labeling in the APC and NAc was visible. In VGAT-Cre mice, an expected comparatively stronger eYFP expression in the NAc vs APC was observed. These expression control data are now presented in the **Supplemental figure 1a**.

Second, we tested for a potential leaky expression of the Cre-dependent tools into glutamatergic cells when using AAV-FLEX-ChR2-YFP in VGAT-Cre mice. For this, we injected AAV-FLEX-ChR2-YFP in a VGAT-Cre mouse and patched GCs three-weeks post-injection. Upon optogenetic stimulation of ChR2-expressing axons, no PSCs were evoked when GCs were held at -75 mV. In contrast, PSCs were recorded when holding GCs at 0 mV. These evoked PSCs were resistant to the AMPA receptor blocker NBQX but abolished in the GABA_A receptor blocker Gabazine. This observation also extended to PSCs recorded in Mitral/Tufted cells. We further generated a current-voltage (I-V) curve and found that the PSCs reversed at ~ -75 mV, consistent with chloride reversal potential. These new data are now presented in Figure 3b-d.

Third, we injected a conditional AAV expressing Synaptophysin-mRuby in the AOC of VGAT-Cre mice. Synaptophysin is a presynaptic protein and therefore mRuby is restricted to axonal presynaptic boutons. In OB slices, $95.8 \pm 0.8\%$ of the labeled mRuby-positive presynaptic boutons colocalized with the vesicular transporter VGAT (see answer to comment 19 for more details).

Lastly, we would like to emphasize that, if glutamatergic fibers were labeled, we would expect to see a strong activation of GCL interneurons with increasing light frequencies, a phenomenon was never observed with the GCL imaging approach.

Taken together, we are confident that our conditional viral expression specifically labels GABAergic neurons in the AOC and their projections in the OB.

2) Secondly, virus spread needs to be shown more thoroughly. Throughout the paper, there are hardly any instances where distribution of infected cell bodies is shown (for anterograde experiments). From the methods section it seems that slices were visually inspected for unwanted virus spread. However, this data needs to be shown and quantified.

Moreover, while I am not familiar with the SOM expression in the olfactory bulb, it might be that a SOM-Cre mouse is a much safer driver line in that e.g. newborn GCs or PGCs (that could conceivably be infected by virus spread from AOC injections) might not express SOM. If that's not possible, a thorough quantification of virus spread, excluding e.g. expression in migrating and newborn neurons is essential. Having said that, as the authors are clearly experts on adult neurogenesis they might have very strong arguments as to why spurious expression in newborn OB interneurons is not a confound but I could neither think of any nor find them in the manuscript.

Our reply:The reviewer rightfully mentions 2 possible issues with viral labeling.

The first one concerns the spread of the virus, from the injection site directly into the OB.

As mentioned in the Materials & Methods section, we have optimized the injection coordinate and the injection volume to prevent any direct leak to the OB. Accordingly the AON is not fully labeled. We now present some pictures of the injection sites in the AON and at the transition between OB and AON (Supplemental Fig. 1b-c for sagittal and horizontal sections). Different examples for different mouse lines are presented in the Supplemental Figure 1c. Another representative sagittal image is also available on Figure 7 for the behavioral data. Note that AAV injection in the AOC is routinely performed in our laboratory and by the authors themselves and similar data has already been published (Mazo et al., J. Neurosci 2016 Figure 1; Lepousez, Nissant et al., PNAS 2014 Figure 3).

The second potential issue concerns the labeling of new-born neurons.

Our lab and others have been using AAVs (notably AAV9) for their major tropism for neurons and their very weak tropism for astrocytes and progenitor-like cells (see for instance Markopoulos et al., 2012 Neuron; Boyd et al., 2015 Cell Report; Otazu et al., Neuron 2015). Nevertheless, we performed a careful quantification of the number of labeled adult-born neurons (either the presence of visible somas in the GCL or the presence of spiny apical dendrites in the EPL) in VGAT-Cre mice injected with a AAV-FLEX-ChR2-eYFP in the AOC, as requested by the reviewer. We observed the presence of 0.21 ± 0.11 labeled adult-born neurons/mm² (n = 6 animals, 8 sections per animal), that is to say less than a cell per 60 μ m-thick coronal slices. This number is similar to the published quantification in Lepousez, Nissant et al. PNAS 2014 (0.32 ± 0.18 labeled adult-born neurons/mm² in GCL following injection in the AOC of WT mice of AAV9-hSyn-ChR2-mCherry).

To further examine the tropism specifically on neuronal progenitors/immature neurons, we directly injected AAV specifically in the lateral wall of the subventricular zone (SVZ) to transduce a large surface of the neurogenic zone (both anterior and posterior SVZ spanning more than one millimeter, see Figure below, panels A, B1, B2). This large injection volume targeting the neurogenic niche of the SVZ should dramatically increase the number of labeled adult-born

neurons, as compared to AOC injection which may only diffuse to some progenitors of the rostral migratory stream (RMS). Three weeks post-injection, we quantified the number of labeled adult-born neurons in the OB (panel B3) and observed a density of 1.74 ± 0.35 cell/mm² (panel C). As a comparison, retroviral/lentiviral injection in the neurogenic niche (SVZ or RMS) led to a density of 200-500 cells/mm² (for instance: Alonso et al., 2012 Nat Neuro: 469 ± 25 cells/mm² in Supplemental Fig. 1). This lack of tropism of AAV is also supported by data from the Allen Brain Connectivity Atlas. Similar AAV1-eGFP injection in the striatum lying next to the lateral ventricle in a WT mouse led to very limited infection and labeling of adult-born neurons (Experiment 146553266 – CP; Experiment 124059700 – CP; Experiment 127762867 – CP; Experiment 146553266 – CP).

On the functional side, none of the patched GCs exhibited intrinsic ChR2-mediated inward current (0/64). To further discard any contribution of potentially-labeled adult-born GC cells, our experiments were performed at three weeks post AAV injection — a time window during which adult-born neurons do not have yet mature GABA synaptic outputs and have limited impact on OB output neurons (see Bardy et al., 2010 J Neurosci, Fig. 3A). Moreover, light-evoked IPSCs from adult-born GC onto MC display rapid kinetics (compatible with MC soma targeting, see Bardy et al., 2010, Figure 1) whereas optogenetic stimulation of the GABAergic cortical fibers produced IPSCs in MCs with a relatively larger latency (see Figure 3 and Supplemental Table 1) consistent with more distal synaptic inputs onto MC dendrites (Bardy et al., J Neurosci 2010 Supplemental Fig. 9). Finally, one would expect much higher connectivity if a significant proportion of adult-born neurons were labeled (Bardy et al., J Neurosci 2010). All these details are now mentioned in the results, discussion and Materials and Methods sections when relevant.

Regarding the use of SOM-Cre mice, we now provide new quantifications demonstrating that SOM⁺ neurons projecting to the OB represent only a subpopulation of all GABAergic neurons projecting to the OB, therefore the use of a SOM-Cre line does not seem judicious. The total number of retrogradely-labeled SOM-

Cre neurons corresponds only to 21.7% of the GABAergic (VGAT-Cre) retrogradely-labeled neurons (Figure 2c versus Figure 2e, note the scale difference). Notably retrograde labeling in SOM-Cre mice failed to capture the dense labeling in the AONv and AONp (“the density of retrogradely-labeled cells was substantially lower in SOM-Cre compared to VGAT-Cre mice in the AONv (VGAT-Cre, 37.5 ± 6.4 ; SOM-Cre, 9.7 ± 0.2) and AONp (VGAT-Cre, 183.8 ± 15.4 ; SOM-Cre, 16.5 ± 0.6 ; Fig. 2c,e”). To substantiate this data, we quantified the proportion of GABAergic (VGAT-Cre) retrogradely labeled cells co-expressing the protein Somatostatin in all the different sub-regions of the olfactory cortex using immunohistochemistry. We reached a similar observation: overall, only 14.8 ± 1.4 % of all retrogradely-labeled GABAergic neurons co-expressed Somatostatin (new Figure 2g). This proportion falls to 2.8 ± 0.7 % in the AONp. To optimize our SOM labeling protocol, slices were co-incubated for 72h with two primary antibodies against Somatostatin (goat anti-Somatostatin, D20 Sc-7819 from Santa Cruz and rabbit anti-Somatostatin, #20067 from Immunostar) and both revealed in red with a biotin-streptavidin amplification (see also response to comment 6). This protocol resulted in ~95% of virally-labeled SOM neurons being colabeled by SOM IHC (Supplemental Figure 1c). Therefore, we conclude that using a SOM-Cre line would dramatically reduce the number of projections studied and will exclude the neurons present in the AONp.

3) As OT seems to have no GABAergic feedback projections to the OB, a control for precision of injection would be to inject conditional AAV into OT.

Our reply: Following the reviewer’s suggestions, we have performed a targeted injection in the ventral striatum/OT in the posterior region (relative to bregma: +1.5 mm anterior). The injection was optimized to avoid both the piriform cortex as well as the HDB, which both project to the OB. Following this injection, we did not observe any fiber in the OB (see Figure below).

4) Cell counting of retrogradely labeled cells is a key part in their argument for GABAergic projections from AOC. The methodology needs to be clearer – how were cells counted? Blind?

Our reply: For cell counting of retrogradely-labeled cells in PV-Cre, SOM-Cre and VIP-Cre mice, counting was performed blind to the animal genotype. For cell counting of retrogradely-labeled cells in VGAT-Cre mice, there was only one condition (i.e., no control versus experimental comparison), so no possibility to be blind to the condition. We’ve now elaborated on the methodology employed for the quantification of retrogradely-labeled cells in the Materials & Methods section (see “Cell counting of retrogradely-labeled cells” section).

5) Also, considering that the area of e.g. AONd in a slice such as the one in Fig 2D looks like it is $<0.4 \text{ mm}^2$ that implies that the 1.8 cells/mm^2 that are reported as retrogradely labeled cells in AONd translate into on average of less than one labeled cell per slice. Thus, can the authors give actual cell numbers counted overall and estimate the total number of GABAergic projection neurons in the different brain regions? Unless I misunderstand, such small numbers seem to make the extensive arborization in the OB even more surprising (as well as the broad and reliable – 50% of cells responding to ChR2 stimulation - physiological effects). Can the authors provide an explanation for this apparent discrepancy? Is labeling simply not efficient enough?

Our reply: In VGAT-Cre mice, AONd virtually does not project to OB: as pointed out by the reviewer, retrogradely-labeled neurons represent less than 1% of all counted cells (see Figure 2c). One could rather consider the density of cells in the AONp for back-of-the-envelope calculation, which show a 100-times higher cell density ($183.8 \pm 15.4 \text{ cells/mm}^2$; Figure 2c). Regardless, when performing OB light stimulation, we stimulate the axons from all the ChR2-expressing AON/APC GABAergic cells projecting to the OB, and not only the ones from AONd. Thus, the total number of AON/APC projecting cells should be considered instead. As asked by the reviewer, the total number of counted cells (2134 cells counted from $n = 5$ animals) is now provided in the legend of Figure 2c. The relative proportions per region can be found in Figure 2c (right histogram). From these quantification on one-out-of-four serial sections, we estimated that our retrograde labeling revealed the existence in the brain of at least $\sim 1,700$ cortical GABAergic neurons projecting to the OB. The connectivity ratio might therefore not be too considering this estimation.

The reviewer also asked about the arborization of the identified long-range GABAergic cells of the AOC. Indeed, describing the arborization pattern as well as the presence of diverging axon collaterals innervating multiple regions is a relevant question to better understand the function of these neurons. In other words, does OB-projecting GABAergic cells of the AOC also project to other brain areas? To unambiguously label OB-projecting GABAergic cells of the AOC without contamination from HDB, we used an intersectional dual-conditional strategy (capitalizing on the combinatorial conditional expression from CRE and FLP recombinases). OB-projecting GABAergic neurons were first targeted with an OB injection of retrograde virus expressing the FLP recombinase in a Cre-dependent manner (AAVretro-DIO-FLP). To label only AONp neurons, we then injected in the AONp a virus expressing ChR2-eYFP in a FLP-dependent manner (AAV-fDIO-ChR2-eYFP). As a control of potential expression leak of the AAV-fDIO-ChR2-eYFP, we also injected 2 mm above in the NAc — a region which does not project to the OB and thus should not express the FLP recombinase. We observed eYFP-expressing neurons specifically in the AONp ($85.1 \pm 2.5\%$ of labeled neurons in the AONp, 9.9 ± 1.4 in the APC, $3.2 \pm 0.4\%$ in AONv and 1.6 ± 0.6 in AONI, $n = 212$ cells from $n = 4$ individuals) and not in the NAc (new Figure 2h). This labeling was also nearly absent when injecting both AAVs in WT mice (see Supplemental Figure 1d). We then analyzed the relative density of fibers (measuring the mean immunoreactivity per surface area, normalized to the immunoreactivity in the AONp), and observed that OB-projecting GABAergic neurons of the AONp innervate preferentially the OB, the AONI, the AONv and the APC. They also send minor projections to other areas of the olfactory cortex. A recent paper mentioned that GABAergic neurons in the APC to OT transition — possibly the AONp — project to the lateral hypothalamus (Murata et al., 2019 Scientific Reports). We unambiguously show here that OB-projecting GABAergic neurons of the AONp nearly do not innervate the lateral hypothalamus. These new data are now presented in Figure 2h and discussed.

6) Fig 2 - SOM- & VGAT-Cre for retrograde labeling in APV and core of AON: While cell density seems to be quite comparable between VGAT-Cre and SOM-Cre retrograde labelling in APC and AONd, actually only around 38% of (retrograde injection in VGAT-Cre animals) labeled cells in those areas were SOM positive. This might be antibody effectiveness – but in order to assess

that, a simultaneous control on e.g. injected SOM-Cre animals would be useful. Otherwise the 37.9 or 38.2% remain without any context and seem inconsistent.

Our reply: We thank the reviewer for their thorough reading and picking up on these discrepancies.

To optimize our SOM immunolabeling protocol, we co-incubated the brain slices for a prolonged period of 72h with two primary antibodies against SOM (goat anti-Somatostatin, D20 Sc-7819 from Santa Cruz and rabbit anti-Somatostatin, #20067 from Immunostar) and revealed both in red with Biotin-Streptavidin amplification system. To quantify the antibody efficiency and specificity, we performed SOM immunostaining in the olfactory cortex of SOM-Cre mice injected with AAV-Flex-ChRimson-mCherry. We observed that $95.5 \pm 0.5\%$ ($n = 2$ mice) of genetically labeled SOM+ neurons were co-labeled with immunostaining against SOM (Supplemental Fig. 1b). We performed the same control for PV immunolabeling efficiency and found that $96.9 \pm 0.5\%$ ($n = 2$ mice) of genetically labeled PV+ neurons were co-labeled with immunostaining against PV (Supplemental Fig. 1b). These numbers are very similar to the published efficiency in SOM-Cre mice (Ssttm2.1(cre)Zjh: 95% of efficiency, see Tanagushi et al., 2011 Neuron).

Using this optimized and validated staining method and careful examination in all the subdivisions of the olfactory cortex, we found a higher colabeling with SOM immunostaining in VGAT-Cre mice in the AONd, AONI, AONm and APC (~50 %). SOM colabeling remained low in the AONv and AONp (see new Figure 2g). Both anterograde and retrograde labeling in SOM-Cre recapitulated our observations using IHC in VGAT-Cre mice: substantial anterograde innervation in the OB but retrograde labeling does not capture the particularly high density of GABAergic projecting cells in the AONp (same order of magnitude than AONv or AONm, compared to an order of magnitude higher in VGAT-cre mice). The relative comparisons within mouse lines highlight the diversity of cortical GABAergic neurons projecting back to the OB — SOM+ neurons projecting to the OB represent only a subpopulation of all GABAergic neurons projecting to the OB (see also response to comment #2). We have rephrased the sentence to highlight this result rather than comparing the absolute densities across mouse lines and brain regions, which might not be trivial.

7) On a related note, I couldn't find a description how subdivisions between brain regions in olfactory cortical areas were made – as this is a very important point for establishing these projections, this needs to be clear, objective and reproducible.

Our reply: We thank the reviewer for pointing out the insufficient description of the subregions classification. We now provide a detailed description on how divisions and subdivisions of the olfactory cortical regions were made.

In all cases, we used well-established anatomical landmarks, previous descriptions from the literature, comparison with standard atlases of the mouse brain and personal lab expertise in this anatomical characterization (see Mazo et al., Sci Rep 2017).

The Material and Methods section now reads:

“Brain regions were manually delineated using morphological parameters, DAPI staining, immunohistochemistry labeling and the Allen Mouse Brain Reference Atlas (<https://mouse.brain-map.org/static/atlas>).

The piriform cortex is located in the ventrolateral forebrain, with a typical three-layered cortex and a layer 2 containing densely packed neurons. The piriform cortex was subdivided into anterior (APC) and posterior (PPC) regions, with the boundary at the caudal end of the lateral olfactory tract (LOT), as in 101.

The OT is “readily identifiable as a large, pronounced, elliptical bulge nested between the LOT, the optic chiasm and the hemispheric midline ridge”. It is a “trilaminar region which contains a peculiar gyrating structure with anatomically defined ‘hills’ (gyri and sulci) and

“islands” (Ref. 102). The OT stains heavily for choline acetyltransferase and this staining was used in some slices (Supplemental Fig. 2b).

The AON is mainly located in the olfactory peduncle and consists of most of it. Yet, as detailed below, it extends caudally to the piriform cortex. The AON is a bilaminar region. Subdivisions of the AON were defined according to well-documented anatomical and cytoarchitectural landmarks (Ref. 31,103,104). The AON can be divided into two basic zones, the pars externa which is a “thin ring of cells that encircles the rostral end of the olfactory peduncle”¹⁰⁴ and the pars principalis. The latter is further subdivided in five regions, four of which are defined as a quadrant emerging from the anterior commissure: pars dorsalis (AONd), pars ventralis (AONv), pars lateralis (AONI) and pars medialis (AONm). A fifth region extends caudally to the piriform cortex (pars posterioralis, AONp). AONI: area that lies directly under the LOT. AONI has the highest density of cells in the pars principalis, forming almost a visible layer. AONm, anterior section: AONm is lying below the OB. Posterior: AONm is delimited by the dorsal and ventral Tenia tecta. The ventral part of AONm also exhibits a cell-free gap which marks the border with AONv. AONd: facing orbito-frontal cortex, with no contact with LOT. The AONd is delimited on the medial border by the dTT. AONv: diffuse layer 2, with lower density of cells and no visible layer-like compared to AONI, which marks the border. AONv has limited contact with the LOT. The OT appears over the AONv in posterior sections.

AONp is caudal to AONv, buried between OT and APC, outside the olfactory peduncle per se. AONp is in contact with the anterior commissure, but with limited contact with LOT. The AONp starts when the dorsal and ventral TT fused and when the OT emerges clearly with 3 layers visible, isolating the AONp from the LOT. AONp has a group of large, loosely aggregated neurons (further references: 32,79,105)

The NDB/MCPO is a more caudal structure, located in the basal forebrain that runs rostro-caudally from the septum to the anterior amygdala area (Ref. 50,80)”.

8) Second sentence in abstract states the discovery of inhibitory cortical projections to subcortical areas. This is only 5% of the data in the paper and should be less prominent. The authors might want to consider dropping it or relegating it to supplement. S1 anterograde labelling: Rephrase as being another example of cortical feedback to subcortical structures. Phrased in this way, it determines the OB as a subcortical structure which – while used by some – seems making the term “subcortical” very generic.

Our reply: We took the reviewer's point into consideration and relegated the S1 cortex data in the Supplemental Figure 4.

We rephrased the sentence to “Here, we uncover for the first time the existence of a parallel **corticofugal** inhibitory feedback.” and do not mention S1 data in the abstract anymore.

9) Include Boehm et al., 2020, showing modulation of odor responses in MTCs by GABAergic (and cholinergic) projections from subcortical basal forebrain (HDB) to bulb, as well as the distribution of axonal projections across OB layers.

Our reply: The reference Böhm et al., 2020 (Böhm, E., Brunert, D. & Rothermel, M. Input dependent modulation of olfactory bulb activity by GABAergic basal forebrain projections. doi:10.1101/2020.03.29.014191) was already present in the reference list (ref. 88) and cited page 16 and page 17 in the initial manuscript. It is now cited in the introduction as well, as we now mention the GABAergic projections from the basal forebrain, as suggested by reviewer 3. This preprint paper is now published in Scientific Reports, and we have updated the reference accordingly.

10) *In the optogenetic silencing experiments of the paper, the authors ignore projections from PPC by targeting injections to AON and APC. The differential projection patterns of MCs and TCs are well-established and get acknowledged by the authors. Omitting a substantial part of piriform, however, poses a bias that needs to be acknowledged more explicitly.*

Our reply: PPC does not contribute substantially to the OB-projecting GABAergic cell population, they represent less than 5% of all OB-projecting GABAergic neurons (see Figure 2). In this first study aiming at investigating cortical GABAergic feedback to the OB, we focused on the two main sources of projections, i.e the AON and APC and referred to them collectively as the anterior olfactory cortex (AOC), similar to the nomenclature in our previous article (Mazo et al., J Neurosci 2016). Following the reviewer's comment, we now clearly mention in the discussion section:

*“As a first step to decipher the functions of cortical GABAergic feedback in sensory systems, we investigated here these inputs as a homogeneous functional unit **by injecting collectively in the AON and APC, the 2 regions consisting of in ~95% of the OB-projecting cortical neurons**”*

11) *In Figure 2f neurochemical identity of APC and PPC should be plotted separately.*

Our reply: We now provide the proportion of GABAergic (VGAT-Cre) retrogradely labeled cells co-expressing the protein Somatostatin in all the subregions of the olfactory cortex (new Figure 2g).

12) *Immunostaining of synaptic components does not provide conclusive evidence of direct synaptic connections. The authors therefore rightfully state “putative GABAergic” synapses”. The CRACM approach with patch recordings in tissue slices provides a more direct line of evidence for synaptic connections. Post-hoc immunostainings of slices containing dye-filled patched cells would strengthen the argument for GABAergic synapses and additionally have the potential to elucidate the neurochemical identity of the projecting cells as well as allow unambiguous confirmation of the morphological identity of the postsynaptic targets. It is, for example, unclear how cell types in deeper layers were discriminated, e.g. dSACs from GCs.*

Our reply: Following the reviewer's comment, we removed our immunostaining data from the manuscript and strengthened our electrophysiological characterization.

We now provide representative pictures of biocytin-filled patched cells as well as more precise details about morphological/electrophysiological properties used to discriminate the different post-synaptic cell types. Identity of the postsynaptic targets was determined using their laminar location in the tissue, morphology, and intrinsic properties, similarly to previous work (Boyd et al., Neuron 2012; Bardy et al., J Neurosci 2010). Unfortunately, we could not obtain sufficiently good quality images of the recorded GCs, because pipette withdrawal inevitably damaged the cells (same issue was encountered for PG cells in Sanz Diez et al., J Physiol 2019). Nevertheless, cells were filled with a fluorescent dye (Alexa 594) and systematically scrutinized after the recording. In the specific case of dSACs and GCs, morphology, input resistance and soma size are drastically different between these cell types. For GCs: small soma of ~8-10 μm in diameter in the superficial GCL with one apical dendrite arborizing in the EPL and small basal dendrites, high input resistance of $875.90 \pm 117.30 \text{ M}\Omega$ (Patched GCs were preferentially located in the superficial GCL, 100-150 μm below the IPL). For dSAC: soma size of ~10-20 μm in diameter in the IPL or immediate surrounding GCL, with unique multipolar dendritic morphology (compared to GCs or MCs) and multiple neurites in the IPL, low input resistance of $268.60 \pm 55.71 \text{ M}\Omega$. We now mentioned these details in the Material and Methods section (see “*Slice Electrophysiology, Identification of neuronal subtypes*”).

We performed additional experiments to confirm the GABAergic nature of the stimulated axons in the OB (see also response to comment 1). 1) PSCs were insensitive to AMPA receptor blockers but were abolished by gabazine (see response to comment 14) and 2) PSCs reversed at ~ -75 mV, consistent with the reversal potential of GABAergic receptors.

Moreover, we confirmed the monosynaptic pathway of GABAergic inputs from olfactory cortex onto OB cells using rabies-based monosynaptic retrograde tracing from genetically identified populations of neurons in the OB (Supplemental Figure 5). The pseudotyped G-deleted rabies virus (EnvA)SAD- Δ G-mCherry was generously provided by Karl-Klaus Conzelmann (Max Von Pettenkofer Institute Virology and Gene Center, Medical Faculty, Ludwig-Maximilians-University Munich, Germany). For this tracing, we injected in the OB of either Tbet-Cre mice (expressing Cre only in M/T cells, ref: Haddad et al., 2013 Nat Neurosci) or in the GCL of VGAT-CRE mice, a Cre-dependent AAV virus expressing the TVA receptor for the Rabies envelope protein EnvA and the Rabies G glycoprotein. Two-weeks post-injection, we injected a modified pseudotyped Rabies virus expressing mCherry in the OB, which infected genetically-identified ‘starter cells’, and then infected pre-synaptic neurons in the olfactory cortex. Using double-immunostaining for GAD67 (Mouse anti-GAD67, Millipore MAB5406), a marker of a subpopulation of GABAergic neurons with detectable somatic labeling (the GAD65 protein being not clearly expressed in the soma), we could observe a concentration of double-labeled cells in the AONp, at the transition between APC, OT, and anterior commissure. These data, presented in Supplemental Figure 5, provide an additional proof of direct monosynaptic connection from GABAergic cells of the AOC onto both OB principal neurons and GCL GABAergic cells.

13) In light of the later in vivo imaging experiments that show differential TC/MC effect why were no m/dTCs recorded from?

Our reply: We agree that differential connectivity to MC vs TC could explain the differences seen in vivo and we thank the reviewer for this suggestion. We thus performed recordings from m/dTCs (new data presented in Figure 3). MCs and m/dTCs appear to display similar input strength and connectivity ratio with cortical GABAergic feedback. We now further discuss these data in the Discussion section. The differential MC vs TC effect observed in vivo may therefore emerge either from differential connectivity with GABAergic projection neurons (different subtypes of different regions), from the differential connectivity with other neuronal populations within the OB, or from differential intrinsic properties.

14) Include population data for pharmacology experiments in Fig 3, not just example traces.

Our reply: We performed additional recordings and now show that input to GCs as well as MCs and TCs are resistant to the AMPA receptor blocker NBQX but are fully abolished by the GABA type A receptor blocker SR95531 (GCs: One-way ANOVA, $F(2,33) = 9.82$, $p = 10^{-4}$; Tukey’s post-hoc test: ACSF vs NBQX: $p > 0.05$, ACSF vs Gabazine, $p = 0.0008$, NBQX vs Gabazine, $p = 0.003$; $n = 11$ GCs; MCs/TCs: One-way ANOVA, $F(2,21) = 12.56$, $p = 0.0003$, Tukey’s post-hoc test: ACSF vs NBQX, $p = 0.99$; ACSF vs Gabazine, $p = 0.0008$; NBQX vs Gabazine, $p = 0.0008$; $n = 8$ MCs/TCs; Fig. 3).

15) The authors show that GABAergic fibers drive inhibition of odor-evoked responses in the GCL with fiber photometry. However, the odor concentration rise time of ~ 3 s and decay of ~ 2 s makes a reliable quantification of the effect of light stimulation during the odor presentation period impossible. Moreover, lingering odor in the “post odor” period impedes any interpretation of the data. More careful analysis is required and the authors need to provide more data on the kinetics of the odor stimulus (if this data is to be retained, see above).

Our reply: The fiber photometry recordings were performed in freely moving mice placed in an odor-delivery chamber with a volume of ~0.5 L, and with an olfactometer output flow of 4 L/min. Those settings are indeed not compatible with sharp rise and decay of odor stimulation kinetics, such as the ones measured directly at the output of an olfactometer. Moreover, the tip of the PID sensor head was positioned on the ceiling of the chamber and not next to the nose of the freely moving mouse, so the detection pattern of the PID is not fully informative of the odor concentration in front of the nose of the animal. We thus removed the misleading PID trace from the figure, which was for an illustrative purpose only and replaced it with the timing of odor valve opening (new figure 4). Yet, the recorded calcium odor responses were robust and reliable, with limited trial-by-trial variability (see the small standard error on odor response), sharp rise kinetics and a stable plateau-like activity. The photostimulation period was positioned during the odor response plateau — and not according to the PID signal — and the impact of light stimulation was quantified as the change in fluorescence with the odor response immediately preceding light stimulation, in the same trial (intra-trial normalization, “net impact of light”). The “net impact of light” is further compared against the same quantification in trials without light stimulation (“odor only”, i.e., change in the magnitude of the odor response between time 1-2 s and 2-3 s after odor onset, ~0% dF/F on average confirming that odor responses had plateaued). Importantly, light stimulation and “odor only” trials were interleaved for each odor-recording pair.

The same rationale applied for the post-light period (we believe the reviewer meant “post-light” period and not “post-odor” period). Yet, this quantification is of secondary importance, and we moved this data to Supplemental Figure 7 in an effort to streamline the manuscript.

16) How does the direct / indirect disynaptic inhibition experiment fit in with the rest of the paper? A direct comparison between GABAergic feedback by genetically targeted expression of ChRimson in AOC and unconditional expression in APC would at least require histological quantification of the injection sites to compare similarity. The data indeed shows that higher frequency and CL stimuli reduce inhibition as measured by photometry. However, the experimental approach is too unspecific to further build on that claim and none of the other experiments follow up on the finding. On a technical note, the authors use a CamKIIa promoter in order to restrict expression to principal neurons. Can they assess specificity histologically?

Our reply: The AOC conveys inhibition to MCs and TCs through direct GABAergic projections as well as through disynaptic inhibition mediated by GCs/PG cells. Introduction of a synaptic relay in extrinsic inhibition in the thalamus is known to induce band-pass filtering of the inhibition. We thus aimed at characterizing the differential frequency recruitment of cortical monosynaptic versus disynaptic inhibition to MCs and TCs. To clarify the rationale of this experiments, we now present these experiments as follows:

“In addition to GABAergic feedback-mediated inhibition, the AOC inhibits MCs and TCs through a disynaptic pathway: glutamatergic feedback drives GCs and PG cells which in turn inhibit MCs and TCs (Boyd et al., 2012; Markopoulos et al., 2012; Mazo et al., 2016) (Supplemental Fig. 7b,c). We thus wished to compare the differential frequency recruitment of inhibition resulting from stimulation of the monosynaptic cortical GABAergic vs. disynaptic cortical glutamatergic pathway. While GABAergic projection stimulation drove increasing MC/TC inhibition with increasing stimulation frequency, the MC/TC inhibition evoked by light stimulation of glutamatergic projection peaked at 33 Hz, implementing a bell-shaped, low-pass filtering of the excitatory drive (Supplemental Fig. 7c).”

OB-projecting GABAergic neurons are clearly outnumbered by OB-projecting pyramidal neurons in the AOC (see for instance rabies experiments in Supplemental Figure 5), resulting in sparser GABAergic projections relative to the glutamatergic ones. This is also true at the level of

postsynaptic cells: only half of GCs received direct inhibitory inputs (Figure 3f) whereas nearly all the GCs received glutamatergic inputs according to previous studies (Lepousez et al., 2014 PNAS; Boyd et al., 2012 Neuron; Markopoulos et al., 2012 Neuron). For these reasons, we agree with the reviewer that the respective strength of each circuit stimulation cannot be directly compared — and we did not wish to do so. Rather, we aimed at showing that the relationship between the inhibition magnitude and the light stimulation frequency are different for the 2 pathways: inhibition increases monotonically with frequency when stimulating GABAergic axons, while inhibition peaked at 33 Hz before reducing again when stimulating glutamatergic feedback. However, to streamline the manuscript, we now present the data related to the monosynaptic inhibition of the MC/TC population in main Figure 4 and the comparison is mentioned in Supplemental Figure 7. We present the direct light-driven cortical inhibition of MC/TC after our GCL interneuron data because of the continuity of the technique and because it provides a simple understanding of the impact of cortical GABAergic axon stimulation on MCs and TCs that raises the question about the “paradoxical” effect mentioned by Reviewer #3 (inhibition of both GCL interneurons and MCs/TCs). It therefore primes the reader to both the model (Figure 5) and 2-photon recording that digs into the different cell types, MC vs TC, at the cellular level (Figure 6). Note that this observation justifies the light protocol used for the subsequent 2-photon imaging figure (33 Hz vs CL), where we reproduced the same finding in MC: CL produces greater inhibition than 33 Hz. A similar trend was observed in TCs, but this was not statistically significant. We kept the comparison between mono- and di-synaptic cortically-driven inhibition as we thought it brings up an interesting distinction and further shows that our manipulation is selective for GABAergic neurons of the AOC (see comments from reviewer 3).

Regarding the CaMKII promoter, this promoter has been widely used to target excitatory pyramidal neurons in the AON and piriform cortex (Mazo et al., J Neurosci 2016; Boyd et al., Cell Rep 2015; Otazu et al., Neuron 2015; Libbrecht et al., 2018 J Neurophysiol; Esquivelzeta Rabell et al., 2017 Curr Biol; Oetl et al., 2016 Neuron; Russo et al., 2020 J Neurosci). Using CaMKII promoter for driving the expression of ChR2 in the AON, these previous works did not report direct light-evoked GABAergic inputs onto OB neurons, suggesting that the CamKII promoter target preferentially glutamatergic pyramidal cells. We now quantified axonal fluorescence across OB layers from GABAergic neurons (VGAT-Cre x AAV9-Flex-ChR2-eYFP) versus CaMKII-expressing neurons (AAV9-CaMKIIa-ChR2-mCherry) labeled with the same injection and found a different innervation pattern, notably in the deep GCL. Regardless, glutamatergic feedback neurons outnumber GABAergic feedback neurons (see Supplemental Figure 5 and Figure 1) and non-specific labeling will likely result in a feedback constituted overwhelmingly of glutamatergic axons. The fact that we do observe a clear frequency-dependent difference between the 2 pathways corroborates this expectation.

17) Unconditional GCaMP6 expression in the OB for somatic MC and TC 2p imaging leads to strong neuropil signal and many unspecific signals. The authors perform extensive computation both to the fluorescence images as well as the extracted traces. Cell types need then be defined only on their depth from the glomerular layer. Despite the established difference in soma size for MCs and TCs, average ROI size for both cell types was identical. Please comment (again – if that part of the paper is retained).

Our reply: PCAs are used to denoise the image/ROIs to help restrict ROI boundaries tightly around neuron somatas, precisely to avoid noise and neuropil contribution to the somata data. Note that they did not modify the pixel's fluorescence values inside the ROIs. Image and trace processing steps were similar as described in Saha et al., bioRxiv 2021.

We now illustrate the processing steps in Supplemental Fig. 11. A first PCA was performed on the registered time-series image in order to remove shot noise and background/tissue

fluorescence since they have low variability across time (Supplemental Fig 10a, b). ROIs were then manually drawn using ImageJ, exported in Matlab (Supplemental Fig. 10c). To minimize neuropil contamination and activity-independent noise, ROIs were denoised using another PCA, performed on all the pixels of each ROI. For illustrative purposes, we used PC1 to reconstruct the shape of the ROIs (Supplemental Fig. 10d-f). Panel g shows that ~90% of the variance in the data is explained by PCs 1-3. Panel h and i show the fluorescence traces for the 11 example ROIs in c-f using either the first PCs 1-3 (h) or PCs 4-6 (i). Panels j-m quantify the data using the signal-rich (PCs 1-3) vs. noisy (PCs 4-6) pixels. In Saha et al. (bioRxiv 2021), the PCA-denoising step increases the signal-to-noise ratio by 25%.

Sizes reported are the sizes of the elliptical ROIs manually drawn in Fiji before the PCA-based denoising and reconstruction step. Therefore, the ROI sizes reported are not the final ones and might not directly correlate with the somata sizes. We apologize for the confusion and thank the reviewer for pointing it out. We removed the reference to ROI sizes in order to avoid confusion in the new version of the manuscript.

Cell types were indeed identified using absolute distance to the glomerular layer, but also relative distance to other cell types in the same tissue (as in Sailor et al., Neuron 2016, Figure 5D), see Method section: “*Mean recording depth (relative to the GL) \pm s.d: MCL, $201.8 \pm 29.9\mu\text{m}$; EPL, $60.1 \pm 20.7\mu\text{m}$.*” Note that these values are in strong accordance with previously reported ones: ~120 μm for TCs and 220 μm for MCs in Yamada et al., Neuron 2017, Figure 1B; 120 μm for TCs, 280 μm for MCs and 70 μm for GL cells in Adam et al., Front Neural Circuits 2014, Figure 3A; 140 μm for TCs, 220 μm for MCs in Otazu et al., Neuron 2015 Figures 5A and 6A.

In addition, visualization of the cell morphology while focusing down in the tissue (along the apical dendrites of MCs and TCs), somata size and density, and neuropil density aided cell type identification. Both depth and cellular morphology have been used extensively previously: Adam et al., Front Neural Circuits 2014; Otazu et al., Neuron 2015; Yamada et al., Neuron 2017; Sailor et al., Neuron 2016.

Finally, we reasoned that neuropil contamination and mixing MCs and TCs would result in the data from the two cell types to be more similar than they actually are and cannot explain the difference observed.

18) The statement “Surprisingly, the magnitude of the light-evoked inhibition on spontaneous and odor-evoked activity was weakly correlated, indicating possible non-linear interactions between top-down GABAergic and bottom-up sensory inputs” is unclear to me – wouldn’t that correlation be expected for simple subtractive inhibition (or divisive inhibition as well)? Why does that imply interactions?

Our reply: We agree with the reviewer that this statement was unclear. At the population level, light-activation of cortical GABAergic feedback results in a subtractive inhibition of both spontaneous and odor-evoked responses. However, at the single cell level, the magnitude of the inhibition during spontaneous vs odor-evoked activity only weakly correlates, i.e. the inhibition received during spontaneous activity only loosely predicts the inhibition received during odor response. A first simple explanation is that the magnitude of the evoked inhibition depends on the neuron firing rate or baseline membrane potential. Another possible explanation is that the net effect of activating GABAergic feedback during spontaneous and odor-evoked activity is mediated by different OB circuits. For instance, GL circuits are more engaged during odor presentation than during spontaneous activity and thus, GABAergic feedback to that layer might contribute more to the overall inhibitory effect on mitral and tufted cells. To clarify our statements, we have reformulated the sentence as followed:

“We also compared the impact of cortical GABAergic stimulation on spontaneous and odor-evoked responses at the individual neuron level. We found little correlation between the magnitude of the light-driven responses in spontaneous versus odor-evoked activity in both MCs and TCs (Supplemental Fig. 9c). For both cell types, light-driven inhibition was slightly stronger in spontaneous versus odor-evoked activity (Supplemental Fig. 8c).”

19) I find the merge in Ext Fig 1b (bottom) not particularly convincing – at least in the resolution in the document. This needs to be quantified as well.

Our reply: We took the reviewer’s comment in consideration and performed a new experiment to better quantify this co-expression. To selectively label the presynaptic boutons of OB-projecting GABAergic neurons (and not the axon shaft + the putative boutons as before), we injected in the AOC of VGAT-Cre mice a conditional viral construct expressing GFP as well as the fusion protein synaptophysin-mRuby. In this condition presynaptic boutons appear in red and the axon shaft in green. We then performed immunostaining against the GABAergic vesicular transporter VGAT, using protocols optimized for the detection of synaptic proteins (Schneider-Gasser et al., 2006 Nat Protocols). A representative confocal picture is presented in Figure 1c. Quantification of colocalization punctas using IMARIS revealed that $95.8 \pm 0.8\%$ of the labeled mRuby-positive presynaptic boutons colocalized with the vesicular transporter VGAT. Considering the tissue penetration issue of the antibody in such a GABAergic-rich brain region — and despite restricting our analysis to the first 4 micrometers from the surface of the slice — we believe this high percentage of co-labeling indicates that virtually all labeled axons are GABAergic. The purely GABAergic nature of the projections was then confirmed with patch-clamp recordings (Fig. 3).

Minor comments

1) Figure 1 legend: change individuals to individual animals/mice

Our reply: This has been changed accordingly.

2) Figure 1 Labels: be clear and more precise which opsin is used

Our reply: “Chr2” has been changed to ChRimson accordingly. This data is now displayed in Supplemental Fig. 3.

3) Figure 2: Statistical comparison of cell densities / co-labelled cells is missing

Our reply: We performed statistical comparison between the 13 different brain regions containing OB-projecting cells and showed that AONp and HDB contained statistically more and denser retrogradely-labeled cells than any other individual regions. This statistical analysis is now provided in the legend of Figure 2, to strengthen the descriptive picture of the relative cell repartition across brain regions.

“AONp and hNDB/MCPO contain the largest density and proportion of cells (density: one-way ANOVA, $F(12,52) = 75.45$, $p = 10^{-28}$, Tukey multiple comparison test, AON vs each individual areas, $p < 10^{-7}$, hNDB/CMPO vs all the other individual areas, $p < 10^{-7}$; proportions: one-way ANOVA, $F(12,52) = 40.86$, $p = 10^{-22}$, Tukey multiple comparison test, AON vs hNDB/MCPO, $p = 0.86$, AON vs all the other individual areas, $p < 10^{-7}$, hNDB/CMPO vs all the other individual areas, $p < 10^{-7}$).”

4) Figure 3 results: Replace “fast latencies” with “short latencies”

Our reply: This has been modified accordingly.

5) Figure 5 results: End of first paragraph should refer to Fig. 5b

Our reply: This has been modified accordingly.

6) ExtDataFig 4: Panel e missing in figure and legend. Panel f: change 2.6 to 0.026

Our reply: This has been corrected accordingly.

7) Table 1: Omit first “tau” for rise time column

Our reply: This has been corrected accordingly. The table has been updated with new TC data.

8) Methods: Please provide age of experimental animals; if possible, indicate which experiments were done in male and female mice

Our reply: Experiments were performed on adult mice (8-10 weeks old). For all our histological experiences, we used indiscriminately male and female. The material and methods section has been updated accordingly.

9) P 8 “in parallel to the establish glutamatergic” \diamond “established”

“were scatter” \diamond “were scattered”

Our reply: This has been corrected accordingly.

10) Fig 4d – the label says light control, I assume that is the shutter control?

Our reply: In Extended Fig 4d and Fig 5d, both labels should read “light control” - light stimulation without expression of ChIEF. We now refer to this nomenclature in the main text:

“We additionally controlled for an effect of blue light illumination per se in control animals who did not express ChIEF (‘light control’)”

11) p.6 “in contrast to glutamatergic projections, GABAergic projections were strictly ipsilateral “ - is this shown anywhere ?

Our reply: A picture of a VGAT-Cre animal unilaterally injected with a retrograde virus is now provided in Supplemental Fig. 2 to illustrate this statement.

12) “Roughly two-third of the labeled cells being concentrated” \diamond verb missing

Our reply: This has been corrected accordingly. The text now says: “Roughly two-third of the labeled cells were concentrated”.

We wish to thank the reviewer for their critical comments. We addressed all the points raised and we believe these modifications greatly increased the quality of the manuscript.

Reviewer #2:

Comments for the Author:

Review of manuscript NN-A74256, “Long-range GABAergic projections contribute to cortical feedback control of sensory processing.”

Mazo and colleagues have provided a well-reasoned and thorough study that characterizes a new circuit motif in the olfactory system and its role in odor discrimination. This is an important topic and significant because it is the first report of long-range GABAergic projections from the olfactory cortex to the olfactory bulb. Beyond identifying these projections, the authors characterize both their functional connectivity and role in shaping odor responses in olfactory bulb output neurons. Throughout the study, a wide range of technical approaches are used, including electrophysiology, *in vivo* imaging, and behavior, to provide a careful and rigorous characterization of these newly identified projections. The manuscript is clearly written, and the data are well-considered and discussed. However, before the manuscript is suitable for publication, I have a few comments below that should be addressed.

1) The authors use a genetic and immunohistochemical approach to demonstrate that the projections from the AOC are indeed GABAergic. VGAT-expressing fibers express eYFP, and IHC was performed against GAD65/67 (extended data figure 1). It looks like there is decent overlap for a number of eYFP fibers with GAD, but for others, perhaps not. Some quantification of the overlap between the two makers should be provided. This will also lend confidence for experiments that follow.

Our reply: We took the reviewer’s comment in consideration and performed a more specific experiment to better quantify this co-expression. To unambiguously label the presynaptic boutons of OB-projection GABAergic neurons (and not the whole axon shaft as before), we injected in the AOC of VGAT-Cre mice a conditional viral construct expressing GFP as well as the fusion protein synaptophysin-mRuby. Functional presynaptic boutons are labeled in red whereas the axonal shaft is labeled in green. We then performed immunostaining against the GABAergic vesicular transporter VGAT, using protocols optimized for the detection of synaptic proteins (Schneider-Gasser et al., 2006 Nat Protocols). A representative confocal picture is presented in Figure 1c. Quantification of colocalization punctas using IMARIS revealed that $95.8 \pm 0.8\%$ of the labeled mRuby-positive presynaptic boutons colocalized with the vesicular transporter VGAT. The GABAergic nature of the axon terminal is then further confirmed with patch-clamp recordings (Figure 3).

2) Figure 3 - The long latency to the IPSC in MCs is interesting and a bit concerning. Given that the recordings are in NBQX it’s unlikely a disynaptic excitatory effect, but this does not rule out disinhibitory, or rebound, effects. For example, dSACs strongly inhibit GCs. In slices, if ChR2 activation inhibits dSACs, which then provides less inhibition to GCs, GCs may fire and release GABA onto MCs. The slow IPSC rise in MCs is also consistent with this possibility. I am not convinced that these recordings are strong evidence for direct connectivity between long-range GABA projections and MCs.

Our reply: We agree that the kinetics of the IPSCs on MCs (and now also TCs) might be a bit surprising at first sight. With our new dataset (including TCs), latencies recorded in MCs and TCs were significantly different from GCs and dSACs (One-Way ANOVA with post-hoc test, $p = 1.7E-4$; MC vs GC, $p = 0.0032$; MC vs dSAC: $p = 0.026$; TC vs GC, $p = 0.0029$; TC vs dSAC, $p = 0.021$).

Between GCs and MCs, latencies were higher by ~1.2 ms only. In contrast, following light stimulation of *glutamatergic* feedback, previous work using pair recordings found the delay between *disynaptic* IPSCs in MCs and monosynaptic EPSCs in GCs being 3.2 ± 0.4 ms (Boyd et al., Neuron 2012, Figure 3B2). Moreover, following glutamatergic feedback light stimulation, reported latencies of onset of *disynaptic* IPSCs evoked in MCs are ~8-10 ms with 10 ms-long light pulse (see Markopoulos et al., 2012 Neuron Figure S2; latencies evoked in MCs in our data: 4.26 ± 0.42 ms with 2 ms-long light pulse). Furthermore, in the original CRACM article, monosynaptic EPSCs latencies were 6.75 ± 1.92 ms, range 3.4–11.6 ms (2 ms laser pulse, Petreanu et al., Nature 2017; in our MC data: 4.26 ± 0.42 , 2.1–6.9 ms range, $n = 13$). One would expect considerably larger latencies with the *trisynaptic* pathway proposed by the reviewer. Also note that the latency to spike of ChR2-expressing neurons is dependent on the light pulse duration (Valley et al., J Neurosci 2013), light power (Petreanu et al., Nature 2017) and is variable between cells (Petreanu et al., Nature 2017), which makes interpretation of latencies dubious when using optogenetics. We therefore decided to remove this data from the main figure and report the latencies in Supplemental Table 1.

To further comment on the seemingly slower latencies, we showed that these slightly longer latencies in M/TC were also associated with slightly slower kinetics in rise time. This suggests filtered inputs, possibly because of their location on electrotonically distal dendrites, as characterized previously for other light-evoked local GABAergic inputs (see Bardy et al., J Neurosci 2010 Fig. S9 for a comparison of distal and proximal inputs onto MCs). This is consistent with our anatomical data showing significant labeling in the GL. It is noteworthy that similar latencies and kinetics were observed for *monosynaptic* EPSCs evoked by cortical *glutamatergic* activation (Markopoulos et al., 2012; Boyd et al., 2012; Mazo et al., 2016).

Finally, we employed monosynaptic retrograde tracing as an alternative technique to confirm the direct GABAergic inputs from the olfactory cortex onto MCs and TCs. We used rabies-based monosynaptic retrograde tracing to label presynaptic partners of genetically identified MCs and TCs (Supplemental Figure 5). The pseudotyped G-deleted rabies virus (EnvA)SAD- Δ G-mCherry was generously provided by Karl-Klaus Conzelmann (Max Von Pettenkofer Institute Virology and Gene Center, Medical Faculty, Ludwig-Maximilians-University Munich, Germany). For this tracing, we injected in the OB of either Tbet-Cre mice (expressing Cre only in M/T cells, ref: Haddad et al., 2013 Nat Neurosci) or in the GCL of VGAT-CRE mice, a Cre-dependent AAV virus expressing the TVA receptor for the Rabies envelope protein EnvA and the Rabies G glycoprotein. Two-weeks post-injection, we injected a modified pseudotyped Rabies virus expressing mCherry in the OB, which infected genetically-identified ‘starter cells’, and then infected pre-synaptic neurons in the olfactory cortex. Using double-immunostaining for GAD67 (Mouse anti-GAD67, Millipore MAB5406), a marker of a subpopulation of GABAergic neurons with detectable somatic labeling (the GAD65 protein being not clearly expressed in the soma), we could observe a concentration of double-labeled cells in the AONp, at the transition between APC, OT and anterior commissure. These data, presented in Supplemental Figure 5, provide an additional proof of direct monosynaptic connection from GABAergic cells of the AOC onto both OB principal neurons and GCL GABAergic cells

We now clarify this point and, extrapolating on the seemingly longer latency in MCs and TCs we posit that they reflect distal inputs onto their apical dendrites.

“The slightly longer IPSC latencies and slower kinetics in MCs and TCs (Fig. 3e, Supplemental Table 1) are consistent with input on electrotonically remote dendrites, presumably apical dendrites in the glomerular layer which are innervated by cortical GABAergic axons (Fig. 1b).”

3) Figure 3 - Were any TCs (not eTCs) recorded? This is important because these cells are a

major focus of subsequent figures along with the curious light-evoked IPSCs in MCs. It could be possible that TCs, but not MCs, are receiving direct inhibition from GABAergic projections, and this might help to explain the population vector separation result in the following experiments.

Our reply: We agree that differential connectivity to MC vs TC could explain the differences seen in vivo and we thank the reviewer for this suggestion. We now performed recordings from TCs (new data presented in Figure 3) and found that MCs and TCs display similar input strength and connectivity with cortical GABAergic feedback. This is now discussed:

“Since MC and TC showed similar connectivity with cortical GABAergic inputs (Fig 3), differential impact on MCs and TCs could arise from differential intrinsic properties, local connectivity (Nagayama et al., 2014), or odor response properties (Burton and Urban, 2014; Fukunaga et al., 2012). Alternatively, MCs and TCs, or the respective GC populations they connect to (superficial vs. deep GC), could be connected distinctly to GABAergic cortical neurons (different cortical regions preferentially targeting MCs versus TCs and/or different types of GCL GABAergic neurons).”

4) Figure 4C - In the PID trace it looks as if the odor concentration is falling right at the midpoint of the light epoch. This makes comparisons between the light and post light epochs somewhat less meaningful. Ideally, the odor concentration should be at a steady-state for each of the before, during, and post epochs.

Our reply: The fiber photometry recordings were performed in *freely moving mice* in a odor stimulation chamber with a volume of ~0.5 L, and with an olfactometer output flow of 4 L/min. Those settings are indeed not compatible with sharp rise and decay of odor stimulation kinetics, such as the ones measured directly at the output of an olfactometer. Moreover, the tip of the PID sensor head was positioned on the ceiling of the chamber and not next to the nose of the freely moving mouse, so the detection pattern of the PID is not fully informative of the odor concentration in front of the nose of the animal. We thus removed the misleading PID trace from the figure, which was used for an illustrative purpose only and replaced it with the timing of odor valve opening (new figure 4). Yet, the recorded calcium odor responses were robust and reliable, with limited trial-by-trial variability (see the small standard error on odor response), sharp rise kinetics and a stable plateau-like activity. The photostimulation period was positioned during the odor response plateau — and not according to the PID signal — and the impact of light stimulation was quantified as the change in fluorescence with the odor response immediately preceding light stimulation, in the same trial (intra-trial normalization, “net impact of light”). The “net impact of light” is further compared against the same quantification in trials without light stimulation (“odor only”, i.e., change in the magnitude of the odor response between time 1-2 s and 2-3 s after odor onset, ~0 % dF/F on average confirming that odor responses had plateaued). Importantly, light stimulation and “odor only” trials were interleaved for each odor-recording pair.

The same rationale applied for the post-light period. Yet, this quantification is of secondary importance, and we moved this data to Supplemental Figure 7 in an effort to streamline the manuscript.

5.1. Figure 4E - Why might 10 Hz stimulation lead to small but significant increases in GC odor-evoked responses - what are the circuit mechanisms involved? Could this be net disinhibition of GCs, since dSACs receive much stronger inhibition than GCs? Some discussion is warranted here.

5.2. It is difficult to square outcomes of Figure 4 with Figure 5 - if the GABAergic projections can be reasonably expected to inhibit granule cells in response to odors (Figure 4), the net effect at

MCs or TCs might be expected to be excitatory. It is puzzling to me that while GCs receive strong inhibitory input from AOC, as measured by GABAergic IPSCs, the net effect of activating these fibers in vivo is suppression of MC odor responses.

Our reply: We thank the reviewer for raising these two points. To investigate these two questions, we built a population model of the OB. We modeled a network of MCs/TCs (excitatory population) and GCs (inhibitory population), with GABAergic inputs from dSACs (model adapted from Chow et al., PLoS Comput Biol 2012).

A single MC/TC was reciprocally connected to a GC, while the GC received additional GABAergic inputs from a dSAC. The dSAC was not modeled per se and was simply providing inhibitory inputs to the GC. The activity of the MC/TC and GC was defined by a system of differential equations. Each population rate is represented by the nullcline (dimensions along which the activity of excitatory and inhibitory subnetworks remains constant, respectively; red, MC/TC; blue, GC) and steady state of the network is achieved at their intersection (fixed point of the system). We computed the fixed point without (solid lines) and with GABAergic feedback stimulation (dashed lines). GABAergic feedback was modeled with varying strength to the MC/TC, GC and inhibited inhibitory dSAC-to-GC inputs (relative strength ratio fixed, derived from our slice data: MC-GC: 1:2). This fixed-point analysis allowed us to explore a wide range of MC/TC-GC connectivity and feedback weight (see new **Fig. 5**). We identified regimes of MC/TC-GC connectivity and feedback weight for which both the excitatory subnetwork (MC/TC) and inhibitory network (GC) were inhibited upon GABAergic feedback stimulation. A phenomenon akin to a “paradoxical” effect when stimulating the inhibitory network elsewhere in the cortex (resulting in inhibition in both the inhibitory and excitatory subnetwork; Sadeh and Clopath, 2021). Interestingly, a weaker GABAergic feedback stimulation strength led to excitation in the GC subnetwork, as suggested by the reviewer and observed in Figure 4c. We reasoned that upon weak cortical feedback stimulation the reduction of the inhibitory drive from dSACs might counteract the direct inhibition from cortical feedback to explain the slight increase in GC firing observed in our experimental GC recordings (also this increase was *not* significant, see Figure 4c). This interpretation is corroborated by work from Labarerra et al (2013) showing that GCs are under tonic GABAergic inhibition in the awake state. Importantly, the values of MC/TC-GC connectivity weight where inhibition of both GC MC/TC populations is observed upon GABAergic feedback stimulation produces physiological firing rates in both subnetworks (low firing rate in GCs: Cang & Isaacson, 2003, Margrie & Schaefer, 2003, Labarerra et al., 2013; 15 Hz firing of MCs/TCs: Mazo et al., 2016; Lepousez and Lledo 2013, Rinberg et al., 2006; Figure 5d). Note that the weight of the dSAC inputs to GCs and the sensory inputs to MC/TC are impacting basal firing rate of GC and MC/TC, respectively, but not the slope of the nullclines. The slope of the nullclines, and thus the impact of GABAergic feedback stimulation, are solely determined by the GC-MC/TC connectivity.

In conclusion, we show that by simulating the impact of a GABAergic feedback on a minimal model of the OB we can recapitulate our experimental data. Critically, GABAergic feedback stimulation strength needs to be strong enough to inhibit both GCs (counteracting the release of inhibition from dSACs) and MCs/TCs (counteracting the release of inhibition from GCs). In order to observe a net disinhibition onto MCs/TCs upon GABAergic feedback stimulation, the model suggests that MCs/TCs would need to be under much stronger reciprocal connections with GCs and would be spiking far less (~6 Hz). The computational data are now presented in Figure 5. We also discuss further the interpretation of our in vivo data in light of our computational results.

6) I have a concern regarding the light vs. odor delivery timing in Figure 5. It perhaps makes a difference in the order in which stimuli were delivered to the animal. For example, if light inactivation of GABAergic projections precedes odor delivery, the effect may be very different

than if light follows odor delivery. Did the authors account for this in the context of animal sniffing dynamics, which control input to the bottom-up stream? Related to this point, was there any change in respiration measured during the light-alone trials?

Our reply: We thank the reviewer for raising this interesting point. Unfortunately, we did not record the respiration rhythm in our experiments. By averaging over 20 repetitions, we randomly sampled different sniff phases and our observation is therefore sniff phase-independent. It is indeed possible that light stimulation at different timing with light onset would have a different impact on MCs odor-evoked activity and it is an interesting avenue for future experiments. Whether light alone could change the respiration dynamics is also an interesting comment although we didn't observe any reactive sniffing triggered by light stimulation. These points are further discussed in the Discussion section:

“Precise spike timing of MCs and TCs relative to OB oscillations is critical for coding of odor intensity (Fukunaga et al., 2012; Shusterman et al., 2011; Smear et al., 2011, 2013), odor identity (Gschwend et al., 2012) and increases during olfactory learning (Li et al., 2015). Thus, altering the tightly regulated spike-field coherence could be a mechanism through which cortical GABAergic feedback could directly shape odor. In the future, it would be interesting to address whether stimulating cortical GABAergic feedback at different phases of the sniff cycle differently impacts MC/TC activity and behavior. “

7) Figure 6 - This is a bit of a chicken and the egg problem. If you are stimulating connectivity that naturally oscillates at beta range with beta-range stimulation, one should expect to see increased power.

Our reply: Network resonance is a classical feature of cortical circuits. When entrained using repetitive stimulation, cortical circuits appear to respond maximally, or resonate, at specific driving frequencies (Lea-Carnall et al., 2016 Plos Comput Biol). A prominent cortical resonance occurs in the gamma band (40-100Hz; Cardin et al., 2009 Nature). This gamma resonance is thought to emerge mainly from the interaction/connectivity between excitatory and inhibitory elements and determined by feedback inhibition (Buzsáki & Wang Annu. Rev. Neurosci., 2012; Tiesinga and Sejnowski, 2009 Neuron). Long-range cortical GABAergic neurons have been repeatedly proposed to play a role in synchronizing distant brain areas (Melzer and Monyer, Nat Rev Neurosci 2020), notably because they have been found to facilitate oscillations at resonance frequency, as revealed by optogenetic manipulations (Melzer et al., Science 2012; Takács et al., Nat Comm 2018; Vandecasteele, et al., PNAS 2014; Gangadharan et al., PNAS 2016; Kimet et al., PNAS 2015). In the olfactory system, the OB exhibits large network oscillations in different frequency bands: respiratory-driven theta (3-12 Hz) and local gamma (40-100 Hz). In addition, beta oscillations (15-40 Hz) are observed in some specific context. Interestingly, network resonance is also observed in the olfactory system — in the OB, network gamma oscillations respond maximally at 66 Hz driving stimulation of mitral/tufted cells (Lepousez & Lledo, Neuron 2013) whereas cortical feedback facilitate beta oscillations (Mazo et al., J Neurosci 2016). Here, we asked what OB oscillation regime would be specifically amplified by stimulation of the GABAergic feedback, if any. The experiment shows that activation of GABAergic feedback at a beta frequency range increases specifically beta oscillations in the OB, whereas stimulation at theta or gamma frequency fails to increase theta and gamma OB oscillations, respectively (now Supplemental Fig. 6). This cannot be explained by an increased inhibitory drive from cortical GABAergic feedbacks because light-evoked inhibition on GCL neurons and MCs and TCs increases with frequency (Figure 4). We conclude that cortical GABAergic feedbacks can specifically enhance beta oscillations when entrained at beta frequencies. This is now better

explained in the result more clearly (see “Cortical GABAergic inputs influences OB network oscillations”). These results have been moved to Supplemental Fig. 6 for strengthen the focus of the paper (as requested by Reviewer #3).

Minor comments

1) “In the olfactory system, sensory neurons project to the external layer of the olfactory bulb” - I think the authors can find another way to state this. Many would consider the external layer of the OB to be the olfactory nerve layer. It’s a minor point, but it distracts as written. “the signal is transmitted to mitral and tufted cells (MCs and TCs, respectively), the output projection neurons of the OB.” - It is not clear what the “signal” is in this context. Sensory information, perhaps?

Our reply: We edited the manuscript accordingly and thank the reviewer for pointing out some of our unclear statements. This sentence now reads:

“In the olfactory system, olfactory sensory neurons project to the olfactory bulb (OB), in the glomerular layer (GL) where they form synapses with apical dendrites of mitral and tufted cells (MCs and TCs, respectively), the output projection neurons of the OB.”

In summary, this submission identifies and characterizes a novel circuit motif in the olfactory system that may have analogs in other sensory systems. The experiments are in general carefully designed and the data are extensively analyzed with clear results. Many of the points raised above can be addressed by clarification of the methodological details or expanded discussion.

Our reply: We wish to thank the reviewer again for their thorough read of our manuscript and constructive comments. We think our manuscript greatly benefited from the reviewer’s input, notably with the addition of the model that should spark interest in the computational community.

Reviewer #3

Comments for the Author:

In this study, Mazo and colleagues explore the properties and function of a novel long-range GABAergic projection from olfactory cortical areas back to the main OB. Overall, the authors present a cogent set of experiments that support an important, and unexpected, role for long-range GABAergic modulation of OB function. The authors' data includes interesting anatomically, electrophysiological and behavioral studies and makes excellent use of molecular labeling approach as well as innovative optical methods for combining light stimulation with 2p cellular imaging studies. The main strengths of study are the completely novel topic and the interdisciplinary approach taken leading to a highly readable and interesting ms. However to this reader, there were three major issues/weaknesses that diminished enthusiasm for the work as current presented.

1) *First, and most easily addressed, was that at multiple points throughout the study the authors digress into secondary (or tertiary) issues that are not presented in enough depth to add much to the paper. These ancillary topics (e.g., long-range GABAergic projections in neocortex and optogenetic modulation of OB oscillations, and to some degree the fiber photometry work) could easily be eliminated from the ms, resulting a tighter, more focused study.*

Our reply: We thank the reviewer for this critical feedback on the manuscript readability. Following their suggestions and to improve the coherence and the focus of the manuscript, we have moved the cortico-thalamic data to Supplemental Fig. 4, removed mention of it in the abstract, and present this data in a separate paragraph to increase readability. LFP oscillation data has also been relegated to Supplemental Fig. 6. GCL fiber photometry data is also now more light-streamed: post-light analysis is relegated to Supplemental Figure 7, the correlation analysis between the impact of light and odor responses have been eliminated. Direct stimulation in the AOC data has also been eliminated. We also reorganized our manuscript: we present fiber photometry in MC/TC after GCL photometry. This primes the reader for the new computational approach as well as for more in-depth investigation of MC vs TC with cellular resolution using 2-photon recordings.

We have also strengthened our anatomical and functional connectivity data (see next point). We now present in Supplemental figures control for “leaky” viral expression, we improved the molecular characterization and provide new data relative to the projection patterns of OB-projecting GABAergic neurons. We have also reinforced the in vitro patch clamp data with additional controls and TC recordings (see next point).

2) Second, and to this reader, the primary issue with the study is whether the molecular labeling approach used for both anatomical and functional studies is completely selective or only partially selective. If there is some cross labeling onto long-range glu fibers, then many of the functional experiments (fiber photometry and behavioral effects following ChR2 stimulation) as suspect. Here, unfortunately, the text and figures related to the key expt on functional specificity seem to be at odds. The text claims that ChR2-evoked responses were “resistant to bath application of AMPA rec antagonists” (p 9). The text and figures only show one AMPAR blocker, NBQX, but do not show the key test--does NBQX attenuate the ChR2 response. Instead, example responses are shown already in NBQX and then following gabazine treatment. Few details are provided in the figure and legend (eg, what was the holding potential?) and no quantification was provided. Since this is, to this reader, the key functional expt in the study, it seems odd that the authors devoted relatively little attention to this analysis.

Our reply: We performed additional experiments to show the selectivity of our genetic approach aiming at specifically labeling GABAergic feedback and further confirm the functional connectivity between GABAergic axons and post-synaptic OB neurons.

1) *Leaky viral expression.* We validated the absence of leak expression of CRE-dependent viral tools. For this, we have co-injected a mix of two viruses — a non-conditional AAV-CaMKII-ChR2-mCherry and conditional Cre-dependent AAV-FLEX-ChR2-eYFP in WT mice (versus VGAT-ires-CRE mice) — and found no expression of the conditional virus in WT (see Supplemental Fig. 1a).

2) *Virus spread/diffusion to the OB.* As mentioned in the Materials & Methods section, we have optimized the injection coordinate and the injection volume to prevent any direct leak to the OB. Accordingly the AON is not fully labeled. We now present some pictures of the injection sites in the AON and at the transition between OB and AON. Different examples for different mouse lines are presented in the Supplemental Figure 1b (Sagittal and horizontal sections). Another representative image is also available on Figure 7. Note that AAV injection in the AOC is routinely performed in our laboratory and by the authors themselves and similar data has already been published (Mazo et al., J Neurosci 2016; Lepousez, Nissant et al., PNAS 2014).

3) *Infection of migrating adult-born neurons from the RMS.* Our lab and others have been using AAVs (notably AAV9) for their main tropism for neurons vs. astrocytes and progenitor-like cells (see for instance Markopoulos et al., 2012 Neuron; Boyd et al., 2015 Cell Report; Otazu et al.,

Neuron 2015). Note that tissue from all mice used in this study was systematically inspected to validate the absence of labeled soma/dendrites in the OB. Regardless, we performed a careful quantification of the number of labeled adult-born neurons (either the presence of visible somas in the GCL or the presence of spiny apical dendrites in the EPL) in VGAT-ires-Cre mice injected with a AAV-FLEX-ChR2-eYFP in the AOC. We observed the presence of 0.21 ± 0.11 labeled adult-born neurons/mm² ($n = 6$ animals, 8 sections per animal), that is to say less than a cell per 60 μm -thick coronal slices. This number is similar to the published quantification in Lepousez, Nissant et al. PNAS 2014 (0.32 ± 0.18 labeled adult-born neurons/mm² in GCL following injection in the AOC of WT mice of AAV9-hSyn-ChR2-mCherry).

To further examine the tropism specifically on neuronal progenitors/immature neurons, we directly injected AAV specifically in the lateral wall of the subventricular zone (SVZ) to transduce a large surface of the neurogenic zone (both anterior and posterior SVZ spanning more than one millimeter, see Figure below, panels A, B1, B2). This large injection volume targeting the neurogenic niche of the SVZ should dramatically increase the number of labeled adult-born neurons, as compared to AOC injection which may only diffuse to some progenitors of the rostral migratory stream (RMS). Three weeks post-injection, we quantified the number of labeled adult-born neurons in the OB (panel B3) and observed a density of 1.74 ± 0.35 cell/mm² (panel C). As a comparison, retroviral/lentiviral injection in the neurogenic niche (SVZ or RMS) led to a density of 200-500 cells/mm² (for instance: Alonso et al., 2012 Nat Neuro: 469 ± 25 cells/mm² in Supplemental Fig. 1). This lack of tropism of AAV is also supported by data from the Allen Brain Connectivity Atlas. Similar AAV1-eGFP injection in the striatum lying next to the lateral ventricle in a WT mouse led to very limited infection and labeling of adult-born neurons (Experiment 146553266 – CP; Experiment 124059700 – CP; Experiment 127762867 – CP; Experiment 146553266 - CP).

4) *Lack of direct infection of GCs (that would arise from virus diffusion or migrating neuron infection)*. None of the patched GCs exhibited intrinsic ChR2-mediated inward current (0/64). To further discard any contribution of potentially-labeled adult-born GC cells, our experiments were performed at three weeks post AAV injection — a time period at which adult-born neurons do not have yet mature GABA synaptic outputs and have limited impact on OB output neurons (see Bardy et al., 2010 J Neurosci, Fig. 3A).

5) *GABAergic nature of the labeled axons in the OB*. To unambiguously label the presynaptic boutons of OB-projection GABAergic neurons (and not the whole axon shaft as before), we injected in the AOC of VGAT-Cre mice a conditional viral construct expressing GFP as well as the fusion protein synaptophysin-mRuby. Functional presynaptic boutons are labeled in red whereas the axonal shaft is labeled in green. We then performed immunostaining against the GABAergic vesicular transporter VGAT, using protocols optimized for the detection of synaptic proteins (Schneider-Gasser et al., 2006 Nat Protocols). A representative confocal picture is presented in Figure 1c. Quantification of colocalization punctas using IMARIS revealed that $95.8 \pm 0.8\%$ of the labeled mRuby-positive presynaptic boutons colocalized with the vesicular transporter VGAT. Considering the tissue penetration issue of the antibody in such a GABAergic-rich brain region — and despite restricting our analysis to the first 4 micrometers from the surface of the slice — we believe this high percentage of co-labeling indicates that virtually all labeled axons are GABAergic. The GABAergic nature of the axon terminal is then further confirmed with patch-clamp recordings (Figure 3).

6) *GABA type A-mediated currents*. We performed the experiment suggested by the reviewer. As now shown in Fig 3, PSCs were insensitive to NBQX application but were abolished by gabazine in both GCs and MCs/TCs (GCs: One-way ANOVA, $F(2,33) = 9.82$, $p = 10^{-4}$; Tukey's post-hoc test: ACSF vs NBQX: $p > 0.05$, ACSF vs Gabazine, $p = 0.0008$, NBQX vs Gabazine, $p = 0.003$;

n = 11; MCs/TCs: One-way ANOVA, $F(2,21) = 12.56$, $p = 0.0003$, Tukey's post-hoc test: ACSF vs NBQX, $p = 0.99$; ACSF vs Gabazine, $p = 0.0008$; NBQX vs Gabazine, $p = 0.0008$; n = 8; Fig. 3).

7) *GABAergic receptor-compatible electrochemical properties of the functional connections*. We additionally performed an I/V curve to show that responses reversed at the expected reversal potential of chloride.

Note that the holding potential was specified in the Method section:

"Inhibitory postsynaptic currents (IPSCs) were recorded at $V_c = 0$ mV"

Following the reviewer's comment, we now indicate it in the figure legend as well (Figure 3).

3) Furthermore, while the latencies to IPSC onset in dSACs look relatively fast and are likely monosynaptic, the same cannot be said for the 4+ ms IPSC onsets recorded in MCs. The difference in IPSC onsets for different targets of the same set of long-range GABAergic axons is unexpected but could be easily explained if there is some cross reactivity in the labeling--given how dense the exc feedback projection is from cortical areas to GABAergic neurons in the OB, the longer latency IPSCs in MC could easily be disynaptic if there is any appreciable cross reactivity. At a minimum, the authors need to provide more detailed both example traces and analysis about the effect of bath NBQX, then followed by gabazine. One also wonders about tests of graded intensity light stimulation and paired stimulation but these are secondary issues.

Our reply: We provide new elements to better qualify these observations and discard any potential doubt.

First, with the response from the preceding comment we believe we demonstrated beyond any doubt that there is no appreciable expression of ChR2 in glutamatergic axons in our preparation.

Second, latencies were obtained from patch recordings performed in the presence of NBQX.

Third, in a subset of MCs and TCs, we performed the requested experiment, and we now show that IPSC magnitudes are not affected by NBQX but were abolished by gabazine (see response above).

For these three reasons, we can discard a disynaptic pathway – as mentioned by Reviewer #2 -- and the simplest polysynaptic pathway would involve three synapses (also raised by Reviewer #2, comment 2).

With our new dataset (including TCs), latencies recorded in MCs and TCs were significantly different from GCs and dSACs (One-Way ANOVA with post-hoc test, $p = 1.7E-4$; MC vs GC, $p = 0.0032$; MC vs dSAC: $p = 0.026$; TC vs GC, $p = 0.0029$; TC vs dSAC, $p = 0.021$).

However, between GCs and MCs, latencies were higher by ~1.2 ms only. In contrast, following light stimulation of *glutamatergic* feedback, previous work using pair recordings found the delay between IPSCs in MCs and EPSCs in GCs being 3.2 ± 0.4 ms (Boyd et al., Neuron 2012, Figure 3B2). Moreover, following glutamatergic feedback light stimulation, reported latencies of onset of *disynaptic* IPSCs evoked in MCs are ~8-10 ms with 10 ms-long light pulse (see Markopoulos et al., 2012 Neuron Figure S2, 10 ms long light pulse; latencies evoked in MCs in our data: 4.26 ± 0.42 ms with 2 ms-long light pulse). Furthermore, in the original CRACM article, monosynaptic EPSCs latencies were 6.75 ± 1.92 ms, range 3.4–11.6 ms (2 ms laser pulse, Petreanu et al., Nature 2017; in our MC data: 4.26 ± 0.42 , 2.1–6.9 ms range, n = 13). One would expect considerably larger latencies with plurisynaptic events (that would be at least trisynaptic given that NBQX was in the bath). Also note that the latency to spike of ChR2-expressing neurons is dependent on the light pulse duration (Valley et al., J Neurosci 2013), light power (Petreanu et al., Nature 2017) and is variable between cells (Petreanu et al., Nature 2017), which makes

interpretation of latencies dubious when using optogenetics. We therefore decided to remove this data from the main figure and report the latencies in Supplemental Table 1.

To further comment on the seemingly slower latencies, we showed that these slightly longer latencies in M/TC were also associated with slightly slower kinetics in rise time. This suggests filtered inputs, possibly because of their location on electrotonically distal dendrites, as characterized previously for other light-evoked local GABAergic inputs (see Bardy et al., J Neurosci 2010 Fig. S9 for a comparison of distal and proximal inputs onto MCs). This is consistent with our anatomical data showing significant labeling in the GL. It is noteworthy that similar latencies and kinetics were observed for EPSCs evoked by cortical glutamatergic activation (Markopoulos et al., 2012; Boyd et al., 2012; Mazo et al., 2016).

Finally, we employed monosynaptic retrograde tracing as an alternative technique to confirm the direct GABAergic inputs from the olfactory cortex onto MCs and TCs. We used rabies-based monosynaptic retrograde tracing to label presynaptic partners of genetically identified MCs and TCs (Supplemental Figure 5). The pseudotyped G-deleted rabies virus (EnvA)SAD- Δ G-mCherry was generously provided by Karl-Klaus Conzelmann (Max Von Pettenkofer Institute Virology and Gene Center, Medical Faculty, Ludwig-Maximilians-University Munich, Germany). For this tracing, we injected in the OB of either Tbet-Cre mice (expressing Cre only in M/T cells, ref: Haddad et al., 2013 Nat Neurosci) or in the GCL of VGAT-CRE mice, a Cre-dependent AAV virus expressing the TVA receptor for the Rabies envelope protein EnvA and the Rabies G glycoprotein. Two-weeks post-injection, we injected a modified pseudotyped Rabies virus expressing mCherry in the OB, which infected genetically-identified ‘starter cells’, and then infected pre-synaptic neurons in the olfactory cortex. Using double-immunostaining for GAD67 (Mouse anti-GAD67, Millipore MAB5406), a marker of a subpopulation of GABAergic neurons with detectable somatic labeling (the GAD65 protein being not clearly expressed in the soma), we could observe a concentration of double-labeled cells in the AONp, at the transition between APC, OT and anterior commissure. These data, presented in Supplemental Figure 5, provide an additional proof of direct monosynaptic connection from GABAergic cells of the AOC onto both OB principal neurons and GCL GABAergic cells

We now better discuss this point and, extrapolating on the seemingly longer latency in MCs and TCs we posit that they reflect distal inputs onto their apical dendrites.

“The slightly longer IPSC latencies and slower kinetics in MCs and TCs (Fig. 3e, Supplemental Table 1) are consistent with input on electrotonically remote dendrites, presumably apical dendrites in the glomerular layer which are innervated by cortical GABAergic axons (Fig. 1b) “.

3) The third weakness was the shift from well characterized optogenetic modulation of long-range GABAergic projects to less well development DREAD-based modulation for the behavioral olf discrimination expt in the final figure. Ideally, this new chemo-genetic approach would be paired with some replication using the same optogenetic light stimulation methods (eg, using a fiber

tether or in a head-fixed configuration) though obviously there would be some compromises from the ideal conditions to conduct the behavioral assay.

Our reply: The choice of a chemogenetic strategy to silence GABAergic cortical axons in the OB was driven by our previous expertise on optogenetic modulation of OB circuits.

The OB is quite large in volume (3 mm deep, 2 mm wide, 3 mm long). In this structure the effect of light-evoked inhibition is dramatically limited by light diffusion/scattering in the tissue, making robust optogenetic inhibition of both dorsal and ventral regions impossible. In addition, it is worth noting that the OB displays a higher degree of redundancy to the point that partial OB lesions have nearly no impact on the discrimination of even highly similar odorants (McBride & Slotnick, 2006 *J Neurosci*). This feature together with the limited light penetration up to ventral OB regions preclude any use of optogenetics tools for loss-of-functions experiments. To experimentally test this on cortico-bulbar circuits, we performed in vivo extracellular local field (LFP) recordings to characterize the effect of eNpHR3.0-mediated inhibition from the surface of OB on excitatory cortical inputs to the OB. For this experiment, we focused on excitatory cortico-bulbar circuits inputs because of their ability to generate a robust light-evoked fEPSP in the LFP, whereas optogenetic stimulation of GABAergic cortico-bulbar inputs had no detectable effect on LFP (see Supplemental Figure 6). Following dual expression of eNpHR3.0 and ChR2 in the AOC, we light-stimulated cortical axons in the olfactory peduncle (5 mW) via an implanted optic fiber and recorded in awake head-fixed mice the resulting evoked extracellular field EPSP (fEPSP) in the OB. Using dorsal OB 589 nm light stimulation (50 mW), we then monitored the impact of eNpHR3.0-mediated inhibition on this fEPSP. Close to the surface (-0.5 mm), eNpHR3.0 stimulation was able to efficiently reduce the amplitude of evoked fEPSP by ~80%. However, as the LFP recording depth increased, the magnitude of the eNpHR3.0-mediated inhibition decreased rapidly, indicating that the ventral part of the OB escaped from the optogenetic inhibition. As a comparison, we examined the silencing effect of the inhibitory DREADD on cortical-bulbar activity, given that hM4D(Gi) is an efficient

synaptic silencer (Stachniak et al., 2014 Neuron). Following the dual injection of AAV-CaMKII-hM4D-mcherry and AAV-CaMKII-ChR2-eYFP in the AOC, we performed LFP recordings coupled to dorsal light stimulation and local OB infusion (as in Soria-Gomez et al., 2014 Nat Neurosci). We observed a robust decrease in the magnitude of light-evoked fEPSP 15-20min following CNO local infusion (See panel e-f in the opposite figure; CNO, 0.1 mg/ml; injection speed, 0.5 μ l in 5 min, ~1mm above the recording site), without any effect on spontaneous theta and gamma oscillations. Altogether, these observations illustrate the stronger efficiency of the chemogenetic DREADD over the optogenetic halorhodopsin approach. We conclude that optogenetic loss-of-function manipulations are not compatible with the size and the circuit properties of the OB.

Other issues:

4) The Introduction should make more mention of long-range GABAergic projections from the septum/DB complex as these are well known in the literature. (And the authors go to describe the DB-to-OB projections in the Results section.)

Our reply: We took the reviewer's comment into consideration. The introduction has been edited as follows:

“The OB additionally receives external inputs from neuromodulatory systems. Specifically, the basal forebrain sends GABAergic axons to the OB (Gracia-Llanes et al., 2010; Zaborszky et al., 1986) and form synapses specifically onto inhibitory neurons (Hanson et al., 2020; Sanz Diez et al., 2019; Villar et al., 2021). Optogenetic stimulation of basal forebrain GABAergic axons in the OB results in an input-dependent impact on MCs, switching from an inhibitory to disinhibitory net effect in presence of odor input (Böhm et al., 2020; Villar et al., 2021). Basal forebrain GABAergic inputs seem important for oscillations and MCs spike synchronization (Villar et al., 2021) and have been found involved in olfactory discrimination (Nunez-Parra et al., 2013).”

5) Ideally, proof of the existence of a new long-range projection is done with EM analysis combined with immuno (eg, post-embedding to label GABA or GAD) as was done for GABAergic projections that arise from the septum/DB complex. In place of that approach, the authors use a variety of alternative methods such as comparing the labeling of pre- and postsynaptic markers in ext data Fig. 2. While not ideal, this approach requires a far more thorough analysis and quantification of apposition/colocalization rates. As it stands, the examples presented in the illustrations are not very compelling and do not add much to the study. The best approach would be to include an EM analysis but at a minimum this alternative, fluorescence microscopy-based approach needs to be presented more rigorously.

Our reply: We agree with the reviewer that our approach was by no means a proof and was purely suggestive of functional synapses. Therefore, we remove this data from the manuscript.

Instead, we now provide a rigorous quantification to substantiate the existence of long-range GABAergic projections. Presynaptic boutons of OB-projecting GABAergic neurons were specifically labeled (and not the axon whole axon shaft as before) by injecting in the AOC of VGAT-Cre mice a conditional viral construct expressing the fusion protein synaptophysin-mRuby (as well as GFP in the axon shaft). In this condition the functional presynaptic boutons appear in red whereas the axonal fiber appear in green. We then performed immunostaining against the GABAergic vesicular transporter VGAT, using protocols optimized for the detection of synaptic proteins (Schneider-Gasser et al., 2006 Nat Protocols). A representative confocal picture is presented in Figure 1c. Quantification of colocalization punctas using IMARIS revealed that 95.8 \pm 0.8% of the labeled mRuby-positive presynaptic boutons colocalized with the vesicular

transporter VGAT. Considering the tissue penetration issue of the antibody in a such GABAergic-rich brain region — and despite restricting our analysis to the first 4 micrometers from the surface of the slice — we believe this high percentage of co-labeling indicate that virtually all eYFP+ axons express VGAT.

In addition to long-range GABAergic projections, we demonstrate the existence of *functional GABAergic synapses* between cortical GABAergic axons and OB neurons using a well-established technique (ChR2-assisted circuit mapping techniques, CRACM; Petreanu et al., Nature 2007, >900 citations). The GABAergic nature of the projections was then confirmed with additional controls as asked by the reviewers (Fig. 3). We further performed rabies-based retrograde monosynaptic tracing experiments to corroborate the presence of monosynaptic GABAergic inputs from cortical neurons onto MCs/TCs and GCs. Similar techniques have been employed by numerous labs to prove the existence of long-range GABAergic projections (for reviews see Melzer and Monyer, Nat Rev Neurosci 2020, Urrutia-Piñones et al., Front In Sys Neurosci 2022).

6) The authors have developed an elegant approach for 2p cellular ca imaging combined with full field illumination to drive optogenetic activation interspersed with 2p monitoring of Ca signals in individual OB principal cells (Fig. 5). With clear results on odor evoked responses, not clear why the extensive focus on effects on the spont activity which seems harder to interpret. At a minimum, it would seem more logical to focus first on modulation of odor-evoked responses and then bring in effects on ongoing spont activity only as far as required to help illuminate the modulation of evoked responses.

Our reply: Looking at the impact of optogenetic stimulation of external input on both spontaneous and odor-evoked activity is a classical comparison in the olfactory system (for instance, see for cortical feedback: Markopoulos et al., Neuron 2012; Boyd et al., Neuron 2012; Mazo et al., J Neurosci 2016; for basal forebrain inputs: Bohm et al., Sci Rep 2020; Ma and Luo, J Neurosci 2012). It aims at characterizing the impact of optogenetic stimulation on two different network states dominated by internally generated versus sensory-driven activity, and which engage different circuits and different inputs. Spontaneous activity is a “resting” state which captures a specific functional network organization structured by internally generated activity. Importantly, MC/TC exhibit a relatively high spontaneous activity (15 Hz: Mazo et al., J Neurosci 2016; Lepousez and Lledo Neuron 2013, Rinberg et al., J Neurosci 2006). Odor stimulation engages additional circuits, notably characterized by the generation of gamma oscillations. In fact, we did find that the impact of light stimulation on MCs and TCs was weaker on odor-evoked versus spontaneous activity (Supplemental Figure 8c).

We rephrased the description of the results as follows:

“We found little correlation between the magnitude of the light-driven responses in spontaneous versus odor-evoked activity in both MCs and TCs (Supplemental Fig. 9c). For both cell types, light-driven inhibition was slightly stronger in spontaneous versus odor-evoked activity (Supplemental Fig. 8c).”

7) The comparison between optogenetically driving long-range glu and GABA axons in ext data fig 5 is exciting, showing differential freq recruitment. This should be extended to test odor responses as it provides another route to combat the criticism that much of the functional effects observed could reflect activation of a small number of long-range glu fibers. One wonders whether activation of long-range glu feedback fibers modulates euclidean distance in the same (TC-specific manner)? If not, then the authors have provided further support for the hypothesis that

their labeling is clean enough (low rates of cross reactivity) to interpret functional studies. But if similar effect, then cross specificity concerns remain

Our reply: Following Reviewer #1's comment and, in order to increase the focus of the manuscript on GABAergic cortical projections, we changed the presentation of these data. Impact of the stimulation of cortical GABAergic axons on the MC/TC populations is now presented after that on GCL interneurons. Data regarding the glutamatergic feedback stimulation is provided in Supplemental Figure 7 for comparison. According to the comments from Reviewer#1 and #2, bringing additional data on glutamatergic projections may not help to improve the focus and coherence of the paper.

However, we agree with Reviewer#3 that, given the relationship between the inhibition magnitude and the light stimulation frequency are different for the glutamatergic and GABAergic pathways, such functional difference provides another element to confirm the absence of cross-reactivity between glutamatergic and GABAergic feedbacks. This point is now mentioned in the Result section. Also, we would like to emphasize that, if glutamatergic fibers were labeled, we would expect to see a strong activation of GCL interneurons, which was never observed with the GCL photometry approach. Additionally, when GCs were patched and held at -75 mV we did not observe any PSCs, confirming the lack of glutamatergic inputs.

Regarding the comparison between the impact of glutamatergic versus GABAergic projections, we now discussed our results on euclidean distance of odor-responses in light of the data from Otazu et al. (Neuron 2015). In this paper, they showed that, in contrast to GABAergic feedback, manipulating glutamatergic feedback altered the similarity of mitral cell, but not tufted cell odor responses. Importantly, Otazu et al used a similar technique to record from MCs and TCs (depth, soma size and denser packing of MCs were used to distinguish MCs and TCs). Therefore, while manipulating glutamatergic feedback influenced MC, but not TC, odor responses similarity, manipulating GABAergic feedback conversely had an effect of TC, but not MC odor responses similarity. We thank the reviewer for raising this point.

Overall, this is interesting study that points out a novel and likely functionally important olfactory pathway. To this reader, there remains considerable caution about accepting the main conclusions until the electrophysiology work showing clear specificity is expanded. Assuming those new control experiments work (eg, no effect of bath NBQX on the optogenetically-driven postsynaptic response), the study will likely find a wide audience, esp within the olfactory community. The revelation of TC-specific modulation of fine olfactory discrimination by long-range GABAergic is especially exciting and will likely spark additional work. One wonders whether the same TC-specificity also applies when comparing different conc of the same odorant? Ideally, though, the fine odor discrimination find should be replicated using optogenic light stimulation, as noted above.

Our reply: We thank the reviewer for their constructive comments, which have greatly consolidate our findings. Using genetics, immunolabeling and pharmacological tools, we showed that AOC GABAergic neurons contribute to the cortico-bulbar pathway. Using anterograde tracing, we show that GABAergic cortico-bulbar axons terminals 1) express the VGAT marker, 2) have a distinct laminar OB innervation profile compared to cortical glutamatergic or basal forebrain GABAergic projections, 3) do not result from a 'leak' in conditional viral expression or from a direct viral transduction of OB interneurons. We confirmed the existence of OB-projecting GABAergic neurons using four retrograde tracing methods (conditional HSVs, a double-conditional AAV approach, monosynaptic rabies tracing and conventional CTB-based retrograde). Electrophysiological recordings coupled with pharmacology confirmed that GABAergic AOC

neurons form monosynaptic GABAergic synapse with OB neurons. Lastly, our functional results showed distinct characteristics compared to stimulation of glutamatergic cortical feedback (inhibition of GCL interneurons, monotonous increase of inhibition with increasing frequency, TC population odor response increase in dissimilarity). This study is the first report of cortico-bulbar GABAergic projections and it might spark a paradigm-shift in the way we think on the role of feedback in early sensory processing. This study opens a lot of questions that need to be addressed in the future.

REVIEWERS' COMMENTS

Reviewer #1 (Remarks to the Author):

Mazo and colleagues have made extensive revisions to their manuscript and have substantially strengthened their study with the inclusion of new data. The manuscript is also now more focused by moving some details to new supplementary figures. The responses to each of the reviewers are exceptionally detailed and carefully considered – it is evident that substantial work has been done to revise and improve this study.

In my previous review, many of my concerns were focused on the electrophysiology data, specifically the lack of TC recordings and questions about the latencies of ChR2-driven synaptic responses. Each of my concerns has been thoroughly addressed. The authors have made additional recordings, provided well-reasoned explanations, and provided a new computational model (Figure 5). The new figure helps reconcile data collected from slices and in vivo imaging experiments and I agree with the authors that the inclusion of this new figure will broaden the appeal of this work to the computational community.

I look forward to the publication of this important work.

Reviewer #2 (Remarks to the Author):

The authors have substantially improved the manuscript and addressed essentially all my major concerns (and virtually all the minor ones). I appreciate that the different aspects of the paper are now much better set up and the paper has become substantially streamlined. I personally still would think that "less is more" with respect to the numerous aspects of functional implications (and remove several of them or relegate them further) but this is probably personal choice. Certainly, the manuscript presents an important finding convincingly.

L 164 - should be "wondered"

L 195 - "in OB slices" or "in an OB slice"

L 201 - should be plural (GCs)

Reviewer #3 (Remarks to the Author):

The authors have done an excellent job in this revision and have addressed the concerns I had on reading the original NN manuscript. While the biophysical reasons why the opto-stimulated IPSC delays are longer onto principal cells than interneurons remain mysterious, the authors have done the obvious experiments to reveal any potential artifacts and other technical issues. To this reader, the new OB computational model doesn't add much to the manuscript especially since there doesn't appear to be a major new conclusion that comes from that work. That work isn't developed very far within this mostly experimental-focused manuscript. For these reasons, authors might consider moving it to its own publication.

RESPONSE TO REVIEWERS

Reviewer #1 (Remarks to the Author):

Mazo and colleagues have made extensive revisions to their manuscript and have substantially strengthened their study with the inclusion of new data. The manuscript is also now more focused by moving some details to new supplementary figures. The responses to each of the reviewers are exceptionally detailed and carefully considered – it is evident that substantial work has been done to revise and improve this study.

In my previous review, many of my concerns were focused on the electrophysiology data, specifically the lack of TC recordings and questions about the latencies of ChR2-driven synaptic responses. Each of my concerns has been thoroughly addressed. The authors have made additional recordings, provided well-reasoned explanations, and provided a new computational model (Figure 5). The new figure helps reconcile data collected from slices and in vivo imaging experiments and I agree with the authors that the inclusion of this new figure will broaden the appeal of this work to the computational community.

I look forward to the publication of this important work.

Our reply : We thank the reviewer #1 for his/her enthusiastic comments.

Reviewer #2 (Remarks to the Author):

The authors have substantially improved the manuscript and addressed essentially all my major concerns (and virtually all the minor ones). I appreciate that the different aspects of the paper are now much better set up and the paper has become substantially streamlined. I personally still would think that "less is more" with respect to the numerous aspects of functional implications (and remove several of them or relegate them further) but this is probably personal choice. Certainly, the manuscript presents an important finding convincingly.

L 164 - should be "wondered"

L 195 - "in OB slices" or "in an OB slice"

L 201 - should be plural (GCs)

Our reply : We thank the reviewer #2 for his/her comments. We have corrected the text accordingly.

Reviewer #3 (Remarks to the Author):

The authors have done an excellent job in this revision and have addressed the concerns I had on reading the original NN manuscript. While the biophysical reasons why the opto-stimulated IPSC delays are longer onto principal cells than interneurons remain mysterious, the authors have done the obvious experiments to reveal any potential artifacts and other technical issues. To this reader, the new OB computational model doesn't add much to the manuscript especially since there doesn't appear to be a major new conclusion that comes from that work. That work isn't developed very far within this mostly experimental-focused manuscript. For these reasons, authors might consider moving it to its own publication.

Our reply : We thank the reviewer #3 for his/her comments. Although we agree that our descriptive model does not offer new testable prediction on the function of cortical GABAergic projections, we believe that the model provides an additional evidence of the impact of these inhibitory projections on the OB circuit. The computational model also helps the reader to compile (and reconcile) all the different forms of inhibition described along the manuscript (inhibition of principal neurons, inhibition of GABAergic GCs, inhibition of GABAergic interneurons inhibiting GCs) and to explore the impact of the cortical projections at different activity regimes which were not all tested experimentally. Therefore the model reinforces the experimental observations and provides a global picture of the circuit properties. Lastly, as mentioned by the reviewer#1, we believe that the model will broaden the appeal of this work to the computational community and not only to neurophysiologists and neuroanatomists.